# ON THE NECESSARY CONDITIONS OF COMPOSITIONAL MODELS

## ABSTRACT

Compositional generalization, the ability to recognize familiar parts in novel contexts, is a defining property of intelligent systems. Modern models are trained on massive datasets, yet these are vanishingly small compared to the full combinatorial space of possible data, raising the question of whether models can reliably generalize to unseen combinations. To formalize what this requires, we propose a set of practically motivated desiderata that any compositionally generalizing system must satisfy, and analyze their implications under standard training with linear classification heads. We show that these desiderata necessitate *linear factorization*, where representations decompose additively into per-concept components, and further imply near-orthogonality across factors. We establish dimension bounds that link the number of concepts to the geometry of representations. Empirically, we survey CLIP and SigLIP families, finding strong evidence for linear factorization, approximate orthogonality, and a tight correlation between the quality of factorization and compositional generalization. Together, our results identify the structural conditions that embeddings must satisfy for compositional generalization, and provide both theoretical clarity and empirical diagnostics for developing foundation models that generalize compositionally.

Data space $\mathcal{X}$

Figure 1: **What enables compositional generalization in CLIP?** Training distributions contain common configurations (left: a cat on a person) but lack rare ones (right: a person on a cat). Yet the same text-based queries, e.g. "A photo of a person", must work on both, even when the latter was never seen during training. We investigate what properties encoder $f$ must satisfy for such transfer to succeed.

## 1 INTRODUCTION

Modern vision systems are trained on tiny, biased samples of a combinatorial space of visual concepts, like objects, attributes, relations in different contexts. Despite this, we expect them to perform well in the wild on novel recombinations of familiar concepts, an expectation tied to the view that systematic generalization, the ability to recombine learned constituents, is a hallmark of intelligence (Fodor & Pylyshyn, 1988). Yet a large body of empirical work shows that even high-performing neural models often struggle with systematicity when train/test combinations mismatch (Lake & Baroni, 2018; Keysers et al., 2020; Hupkes et al., 2022; Uselis et al., 2025). At the same time, large vision–language models such as CLIP (Radford et al., 2021) and its variants are trained on web-scale datasets (e.g., LAION-400M (Schuhmann et al., 2021a)) and achieve impressive zero-shot transfer on many tasks (Radford et al., 2021; Zhai et al., 2022).

However, they often fail when test images contain unusual combinations of familiar concepts (Xu et al., 2022; Bao et al., 2024; Thrush et al., 2022; Abbasi et al., 2024; Yuksekgonul et al., 2023; Ma et al., 2023). Figure 1 illustrates this tension for CLIP-like architectures: an image encoder $f$ produces embeddings on which linear classifiers predict concepts, but training data $\mathcal{X}_{train}$ cover only common compositions (such as a cat on a person) from the full data space $\mathcal{X}$, while models must answer queries like "Is there a person present?" correctly even on rare compositions (such as a cartoon of a person on a cat) from $\mathcal{X} \setminus \mathcal{X}_{train}$. Given how rarely, if at all, such compositions appear in training, we aim to identify which properties could enable generalization. To study this, we ask: *assuming that compositional generalization succeeds, what properties must the representations have to accommodate it?*

We argue for *non-negotiable, model-agnostic* properties that any neural-network-based system claiming compositional generalization must satisfy. We state three desiderata: *divisibility*, *transferability*, and *stability*. These desiderata formalize that (i) all parts of an input should be accessible to a simple readout; (ii) readouts trained on a tiny but diverse subset should transfer to unseen combinations; and (iii) training on any valid subset should yield robust generalization. Our scope is the common setting where predictions are linear in the embedding $f$: CLIP-style zero-shot classifiers, linear probing, and cases where a fixed non-linear head is folded into the encoder.

Our key finding is that these desiderata *necessitate* a specific geometry: *linear factorization* with *near-orthogonal* concept directions. This establishes what any model *must* achieve to compositionally generalize under standard training, providing a concrete target for future design. Moreover, it offers theoretical grounding for the *Linear Representation Hypothesis* – the linear structure widely observed in neural representations is a *necessary consequence* of compositional generalization.

Our contributions are: (1) **Defining desiderata.** We define three desiderata: *divisibility*, *transferability*, *stability*, and formalize compositional generalization in their terms. (2) **Structural necessity.** Under GD with CE/BCE, these desiderata imply *linear factorization*: embeddings decompose into per-concept sums with orthogonal difference directions. (3) **Empirical grounding.** Across CLIP and SigLIP families, we find strong evidence of factorization, near-orthogonality, low-rank per-concept geometry, and correlation with compositional generalization accuracy.

## 2 RELATED WORK

**Compositional generalization.** Research on compositional generalization investigates how models can systematically combine concepts. On the objective side, approaches such as Compositional Feature Alignment (Wang, 2025) and Compositional Risk Minimization (Mahajan et al., 2025) study how model training objectives, and model architecture Jarvis et al. (2024) affect compositional generalization. On the representational side, kernel analyses characterize when certain compositional structures in embeddings yield generalization theoretically (Lippl & Stachenfeld, 2025), and empirical work investigates the role of disentangled representations for compositional generalization (Montero et al., 2021; Dittadi et al., 2021; Liang et al., 2025). On the data side, recent work probes whether and how scaling and data coverage improve compositional behavior (Uselis et al., 2025; Schott et al., 2022; Kempf et al., 2025). Abbasi et al. (2024) investigate CLIP's ability to recognize unlikely attribute-object combinations, finding that CLIP models still fall short on such tasks.

Other works establish formal sufficient conditions for when particular model classes can achieve compositional generalization, e.g., generative models whose data are produced by a differentiable rendering process and whose training distribution provides compositional support over latent factors (Wiedemer et al., 2023), discriminative models whose inputs are drawn from an additive energy distribution (Mahajan et al., 2025), or linearly factorized representations (Uselis et al., 2025). In contrast, we do not impose specific structure on the data-generating process or on the learned representations. Instead, we ask what properties are implied *if* a model transfers from a restricted subset of the data space to the full space under our desiderata. Within this setting, our results can be interpreted as providing *necessary* conditions for compositional generalization for models that satisfy these desiderata.

**Geometry of learned representations.** A large literature studies the shape of learned features. In VLMs, Trager et al. (2023) report compositional linear subspaces, while in LLMs the *Linear Representation Hypothesis* (LRH) is examined mechanistically and statistically (Jiang et al., 2024; Park et al., 2023). Extending LRH, Engels et al. (2025) show that features can be multi-dimensional rather than rank-1, and Roeder et al. (2020) analyze identifiability constraints. Sparse-autoencoder probes provide evidence for monosemantic or selectively remapped features in VLMs (Pach et al., 2025; Zaigrajew et al., 2025; Lim et al., 2025). Beyond nominal labels, ordinal/ordered concepts motivate the rankability of embeddings (Sonthalia et al., 2025). More broadly, capacity limits for embedding-based retrieval emphasize geometric bottlenecks (Weller et al., 2025). Elhage et al. (2022) investigated empirically how neural networks can represent more features than there are dimensions in two-layer auto-encoder models. They found a tendency to encode features near-orthogonally with respect to neurons. Abbasi et al. (2024) find evidence of disentanglement in CLIP models. In contrast to these works, which document linear or near-orthogonal structure empirically, we show that under practice-driven desiderata and standard training, linearity and orthogonality are *necessary*.

**Data, objectives, and training effects on geometry.** Data distribution strongly shapes zero-shot behavior; concept frequency during pretraining predicts multimodal performance (Udandarao et al., 2024). On the objective side, BCE vs. CE can induce different feature geometries (Li et al., 2025), and contrastive/InfoNCE objectives exhibit characteristic similarity patterns (Lee et al., 2025). Convergence perspectives argue that the *objective* drives canonical representational forms (Huh et al., 2024), and objective choice has been tied to representational similarity across datasets (Ciernik et al., 2025).

**Binding, explicit structure injection, and concept identification.** Work on *binding* asks whether models maintain factored world states (Feng et al., 2025), and CLIP has been observed to show uni-modal binding (Koishigarina et al., 2025). Surveys and empirical studies examine binding limits and emergent symbolic mechanisms (Campbell et al., 2025; Assouel et al., 2025). Other approaches inject structure directly, e.g., hyperbolic image–text embeddings and entailment learning (Pal et al., 2024; Desai et al., 2024), or pursue concept identification at the causal/foundation interface and object-centric pipelines (Rajendran et al., 2024; Mamaghan et al., 2024).

**Relation to disentangled representation learning.** Work on disentangled representations largely focuses on specifying desiderata for internal codes (e.g., disentanglement, completeness, informativeness) and proposing metrics or training schemes to satisfy them, often with the informal motivation that such structure should help downstream generalization (Bengio et al., 2014; Eastwood & Williams, 2018; Higgins et al., 2018). Few recent studies directly probe how these properties relate to out-of-distribution or compositional generalization, with mixed or limited evidence (Watters et al., 2019; Dittadi et al., 2021; Montero et al., 2021; 2024). We instead ask a complementary question: if a discriminative model model *does* exhibit compositional generalization when learned from a subset of the data space, what must necessarily be true of its embeddings?

We provide a more detailed discussion in Appendix B

## 3 SETUP: A FRAMEWORK FOR COMPOSITIONALITY

We begin by detailing key desiderata for embedding models that contend to be compositional. We motivate them from a practical perspective: (1) models need to support distinguishing between any combination of concepts, (2) practical data collection is limited to a subset of the concept space, so a model needs to be able to transfer from a subset of the concept space to the full concept space, and (3) in practice apriori it is not known which subset needs to be chosen, so a model should be able to transfer robustly from any subset, matching in probability distribution to retraining over any other dataset.

### 3.1 SETUP: CONCEPT SPACES AND DATA COLLECTION PROCESS

We interpret the world as a product of concepts: any input $x_c \in \mathcal{X}$ (e.g., images) has an associated tuple of concepts $c \in \mathcal{C}$, describing its constituent parts and properties. This is a reasonable way to describe a large portion of the world. For example, current large-scale datasets (e.g., image–caption pairs) provide noisy natural-language descriptions that can be decomposed into *discrete* concept values. Clearly, a single concept tuple cannot capture all aspects of the world, e.g. how attributes bind to objects or how different objects relate spatially. Still, an intelligent system should at least be able to tell apart basic concepts (such as objects and their attributes), even without modeling their relations. In other words, concept spaces may not capture the full compositional structure of the world, but any model of the world must involve them in some form. Importantly, we do not assume *how* the concept values are distributed (e.g. being independent), only *what* they represent.

**Definition 1** (Concept space). Suppose we have $k$ concepts, and each concept can take $n$ possible values. For each concept $\mathcal{C}_i$ ($i = 1, \ldots, k$), let its set of possible values be $\mathcal{C}_i = \{1, \ldots, n\}$. The *concept space* is the Cartesian product

$$\mathcal{C} = \mathcal{C}_1 \times \mathcal{C}_2 \times \cdots \times \mathcal{C}_k = [n]^k, \tag{1}$$

that is, the set of all possible tuples $c$ with $|\mathcal{C}| = n^k$. We index inputs by concept tuples: for each $c \in \mathcal{C}$ we assume an associated $x_c \in \mathcal{X}$ (e.g., a natural image) realizing $c$.

Data-related components for compositional generalization involves three notions: (1) the total variation of the data, (2) the concepts we aim to learn and expect the model to capture, and (3) the data that is actually collectible. We capture (1) by the concept space $\mathcal{C}$ (Definition 1); (2), the targets that we aim to capture can be described by a label function $l : \mathcal{C} \to \mathcal{V} \subseteq \mathcal{C}$ that capture which concepts and

their values we want to learn. In this work we take the full target $\mathcal{V} = \mathcal{C}$, by noting that foundation models attempt to align with all present concepts. For (3), we formalize collectability constraints through a validity class that specfies which training supports are valid, indicating which concept combinations may appear in training. We formalize this below.

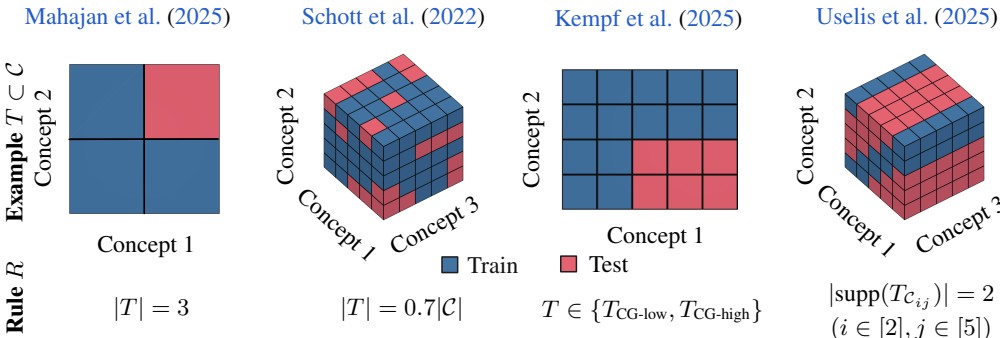

Figure 2: **Interpreting previous works' sampling designs $T$ and validity rules $R$.** Training sets $T$ specify which concept combinations are observed. Validity rules $R$ determine valid training configurations for generalization evaluation.

**Considering data collection.** We are interested in models that support efficient compositional generalization from a subset of the concept space. To formalize this notion, we specify a validity class $\mathcal{T} \subseteq 2^{\mathcal{C}}$ of valid training sets, where $2^{\mathcal{C}}$ denotes the power set of $\mathcal{C}$, and a validity rule $R : 2^{\mathcal{C}} \to \{0, 1\}$ that specifies whether a given training set is valid. This setup captures the natural question of which training sets we use and for which we expect generalization.

**Definition 2** (Training support, validity class, and training dataset). Let $\mathcal{C}$ be the concept space. A *training support* is any subset $T \subseteq \mathcal{C}$. *Validity class* is a collection $\mathcal{T} \subseteq 2^{\mathcal{C}}$ whose members are called *valid training sets*. The class $\mathcal{T}$ specifies which training sets are observable. Validity class $\mathcal{T}$ is specified by a *validity rule* $R : 2^{\mathcal{C}} \to \{0, 1\}$ through $\mathcal{T} = \{T \subseteq \mathcal{C} : R(T) = 1\}$. A *training dataset* for a training set $T$ is $D_T = \{(\boldsymbol{x_c}, \boldsymbol{c}) : \boldsymbol{c} \in T\}$.

We note that there are many validity rules used in practice. For example, if we can collect any subset of size $N < |\mathcal{C}|$, then $R(T) = 1$ whenever $|T| = N$. Figure 2 illustrates common choices: Mahajan et al. (2025) use training supports that cover every concept value; Schott et al. (2022) use random samples covering 70% of all combinations; Kempf et al. (2025) specify a small set of allowed supports; and Uselis et al. (2025) use supports whose joint marginals cover at least two values per concept. Note that these validity rules apply to concept supports rather than individual datapoints.

### 3.2 Compositional representations and models

Given the concept space and the training supports, we now make precise how we expect models to learn. We work with encoders $f$ that map an input to a vector representation (embedding).

**Scope of models.** We study embedding models: these cover modern foundation models like CLIP and SigLIP (Tschannen et al., 2025; Zhai et al., 2023), supervised-learning models, self-supervised models like DINO (Caron et al., 2021). At inference the models we study are *non-contextual*: the representation of an input depends only on that input (no dependence on other test examples, prompts, or the batch). Formally, the encoder is a map $f : \mathcal{X} \to \mathcal{Z}$, with $\boldsymbol{z} = f(\boldsymbol{x})$ (optionally $\ell_2$-normalized).

**Readout class (linear vs. non-linear).** Usually, encoders $f$ are associated with either a downstream or readout model $h$ that takes $\boldsymbol{z} = f(\boldsymbol{x})$ and outputs per-concept logits $h(\boldsymbol{z}) \in \mathbb{R}^{k \times n}$ using argmax classification rule (see Definition 3). This covers zero-shot use of text features as linear classifiers, standard linear probing, and the affine last layer in most neural classifiers. If $h$ is non-linear in a neural network, we absorb the layers preceding the linear layer $g$ into the encoder ($\tilde{f} = g \circ f$) and analyze the resulting affine layer. The definition below keeps the readout $h$ general to allow future extensions beyond linear heads, but all results in this paper consider the linear case, without such restrictions a high-capacity readout could make any injective encoder appear compositional by memorization.

**Definition 3** ((Linearly) compositional model). An encoder $f : \mathcal{X} \to \mathcal{Z}$ is *compositional w.r.t.* $\mathcal{C}$ if there exists $h : \mathcal{Z} \to \mathbb{R}^{k \times n}$ such that, for all $\boldsymbol{c} \in \mathcal{C}$ and all $i \in [k]$,

$$c_i = \arg\max_{j \in [n]} h(f(\boldsymbol{x}_{\boldsymbol{c}}))_{i,j}. \tag{2}$$

It is *linearly compositional* if $h$ can be taken affine $h(z) = Wz + b$. We refer to $h$ as the *readout*.

### 3.3 COMPOSITIONAL GENERALIZATION AND DESIDERATA

Given the ingredients (concept space $\mathcal{C}$, encoder $f$, and training-support family $\mathcal{T}$), we now define a learning rule $A$ and state three desiderata for compositional generalization: *divisibility*, *transferability*, and *stability*. We emphasize that this desiderata is on the NN-based models that exhibit generalization, as defined below, not on the representations, as studied in disentangled representation learning.

**Considering training.** We view a learning algorithm as a simple map

$$A : D_T \mapsto h_T, \qquad h_T \in \mathcal{H} \subseteq \{h : \mathcal{Z} \to \mathbb{R}^{k \times n}\},$$

from a dataset supported on $T \subseteq \mathcal{C}$ to a readout in a chosen hypothesis class. In practice, $A$ is typically (stochastic) gradient descent on a cross-entropy or contrastive objective, covering contrastive vision–language encoders (e.g., CLIP, SigLIP), standard supervised classifiers, and linear probes on self-supervised vision encoders like DINO.

$\exists\, h : h(f(\boldsymbol{x}_{\boldsymbol{c}}))$ correct $\forall \boldsymbol{c} \in \mathcal{C}$

Figure 3: **Relationship between (generalizing) compositional models.** The plot illustrates what requirements each definition imposes on classifiability (orange nodes), and transfer (purple nodes).

**Desiderata for compositional generalization.** Suppose we train a downstream readout $h_T = A(D_T)$ on some $T \in \mathcal{T}$. What should $h_T$ satisfy? We argue for three practically-motivated properties.

First, every combination of concept values should be *classifiable* by the readout: for any $\boldsymbol{c} \in \mathcal{C}$, the corresponding region of the representation space of $f$ is nonempty: there exists at least one $\boldsymbol{z}$ that $h_T$ assigns the concept values $\boldsymbol{c}$. Otherwise, generalization to the full grid is impossible. We refer to this property as *Divisibility*.

**Desideratum 1** (Divisibility). For a readout $h : \mathcal{Z} \to \mathbb{R}^{k \times n}$, every concept tuple must be classifiable:

$$\forall \boldsymbol{c} \in \mathcal{C} : \bigcap_{i=1}^{k} \mathcal{R}_{i,c_i}(h) \neq \varnothing, \quad \text{where } \mathcal{R}_{ij}(h) = \{\boldsymbol{z} \in \mathcal{Z} : \arg\max_{j' \in [n]} h(\boldsymbol{x}')_{i,j'} = j\}. \tag{3}$$

Divisibility is necessary but not sufficient: it guarantees that the space is divisible, but does not imply that the readout will be correct. We therefore ask that, for every training set, the learned readout transfers to the full grid; we refer to this as *Transferability*.

**Desideratum 2** (Transferability). For every $T \in \mathcal{T}$, the trained readout $h_T = A(D_T)$ correctly classifies all possible combinations of the concept space:

$$\forall \boldsymbol{c} \in \mathcal{C}, \, \forall i \in [k] : \quad \arg\max_{j \in [n]} h_T\big(f(\boldsymbol{x}_{\boldsymbol{c}})\big)_{i,j} = c_i. \tag{4}$$

Note that Transferability implies Divisibility. We state Divisibility explicitly because it highlights a capacity requirement: the embedding space must be able to represent all concept combinations.

Third, consider readouts learned from different valid supports $T \in \mathcal{T}$. Divisibility and Transferability ensure do not say anything about the behavior of the classification decisions. Intuitively: if an input depicts a "cat", retraining on another valid support should not flip the preference to "dog" or push the prediction toward near-indifference. We refer to this as *Stability*.

**Desideratum 3** (Stability). For any $T, T' \in \mathcal{T}$, any grid point $\boldsymbol{x}_{\boldsymbol{c}}$, and any $i \in [k]$, the per-concept posteriors agree across supports:

$$p_i^{(T)}(j \mid f(\boldsymbol{x}_{\boldsymbol{c}})) = \frac{\exp(h_T(f(\boldsymbol{x}_{\boldsymbol{c}}))_{i,j})}{\sum_{k=1}^{n} \exp(h_T(f(\boldsymbol{x}_{\boldsymbol{c}}))_{i,k})}, \qquad p_i^{(T)}(\cdot \mid f(\boldsymbol{x}_{\boldsymbol{c}})) = p_i^{(T')}(\cdot \mid f(\boldsymbol{x}_{\boldsymbol{c}})). \tag{5}$$

**Defining compositional generalization.** We now tie the ingredients into a single tuple $\Pi = (f, \mathcal{H}, A, \mathcal{T})$, which we use as the object that specifies the entire compositional-generalization setup: the encoder, the readout class, the learning rule, and the family of valid training supports. We specify compositional generalization as a process of learning readouts that generalize over *all* $T \in \mathcal{T}$ and satisfy Desiderata 1–3.

**Definition 4** (Compositional generalization). $\Pi = (f, \mathcal{H}, A, \mathcal{T})$ *exhibits compositional generalization if, for every* $T \in \mathcal{T}$ *with* $h_T = A(D_T)$, *Divisibility (Def.* 1*) and Transferability (Def.* 2*) hold on the full grid, and the posteriors are Stable across valid retrainings (Def.* 3*) for all pairs* $T, T' \in \mathcal{T}$. *We say that* $\Pi$ *exhibits* linear compositional generalization *when the readout hypothesis class is linear.*

We illustrate the relationship between (linear) models and their compositional counterparts in Figure 3. In practice one could consider relaxed or average-case variants; however, we here are interested in "ideal" representations that support compositional generalization under any data sample.

### 3.4 INSTANTIATING THE FRAMEWORK WITH CLIP

We instantiate the framework in the dual-encoder, vision–language setting in the style of CLIP models: images and texts are embedded into a shared space and trained to align, with captions acting as noisy descriptions of concept tuples.

**Encoders.** Let $f : \mathcal{X} \to \mathcal{Z}$ be the image encoder and $g : \mathcal{Y} \to \mathcal{Z}$ the text encoder. At inference both are typically $\ell_2$-normalized so that inner products are cosine similarities: $\|f(\boldsymbol{x})\| = \|g(\boldsymbol{y})\| = 1$.

**Prompts as linear probes.** Zero-shot classification uses text features as linear classifiers. For each concept $i \in [k]$ and value $j \in [n]$, we can choose a prompt $p_{i,j}$ (e.g., "a photo of a cat") and define a probe vector $\boldsymbol{w}_{i,j} := g(p_{i,j}) \in \mathcal{Z}$. Stacking these gives a readout

$$h(\boldsymbol{z}) = \left[ \boldsymbol{w}_{i,j}^\top \boldsymbol{z} \right]_{i,j} \in \mathbb{R}^{k \times n}.$$

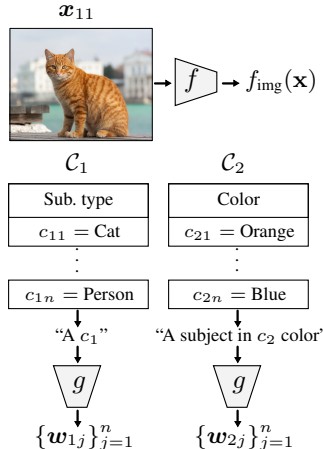

Figure 4: **Instantiating the framework with CLIP-like embedding models for analysis.**

Here $f$ is the representation model, while $h$ is a linear readout whose weights come from the text encoder. Training in CLIP-like models can be viewed as learning a readout model where the *same* set of text-derived probes serves across many images; prompts often mention only parts of an image, so the system is implicitly asked to recognize objects and attributes regardless of which other concepts co-occur. We illustrate this process in Figure 4.

**The question we study.** Given a concept space $\mathcal{C}$, what structure must $\boldsymbol{z} = f(\boldsymbol{x_c})$ have so that a single set of probes $\{\boldsymbol{w}_{i,j}\}$ (whether fixed by $g$ or learned as linear probes) satisfies our desiderata (Desiderata 1–3) on the full $\mathcal{C}$? In other words, what constraints does zero-shot, probe-based classification place on the geometry of image representations if we want compositional generalization?

## 4 IMPLICATIONS OF COMPOSITIONALITY ON REPRESENTATIONS

We now ask what our desiderata *force* on representations in common training regimes. Two questions guide the section:

**Q1** (§4.1) *Geometry under GD with CE/BCE and stable transfer.* If $A$ is gradient descent under binary cross-entropy, and $\Pi$ exhibits compositional generalization (Def. 4) across a family of supports $\mathcal{T}$, what structure is *necessary* for $f$ (and the linear readout $h$)? $\to$ We show additive (linear) factorization with orthogonal concept directions under natural $\mathcal{T}$.

**Q2** (§4.2) *Minimal dimension for linear readout.* Assuming separability/divisibility and a linear (affine) readout $h$, what is the smallest $d$ so that correct per-concept predictions are possible over all $n^k$ tuples? $\to$ With affine readouts, $d \geq k$ is necessary and tight.

### 4.1 GEOMETRY OF $f$ UNDER COMMON TRAINING SETTINGS

We instantiate $A$ as gradient descent on the binary cross-entropy (logistic) loss. As in §3.4, the readout $h$ is linear in the embedding $\boldsymbol{z} = f(\boldsymbol{x})$ (text-derived probes or learned linear heads). We illustrate the stable and unstable examples of feature representations in Figure 5.

**Proposition 1** (Binary case: compositional generalization implies linear factorization). Let $\Pi = (f, \mathcal{H}, A, \mathcal{T})$ be the tuple instantiated in Section 3.4, with linear heads $\mathcal{H}$ and $A$ given by GD+CE. Suppose that the training sets follow random sampling with validity rule $R(T) = 1$ if $|T| = 2^{k-1} + 1$. Assume Desiderata 1–3 are satisfied. Then under the binary grid $\mathcal{C}_i = \{0, 1\}$ with $\mathcal{X} = \{\boldsymbol{x_c} : \boldsymbol{c} \in [2]^k\} \subset \mathbb{R}^d$, there exist $\{\boldsymbol{u}_{i,0}, \boldsymbol{u}_{i,1} \in \mathbb{R}^d\}_{i=1}^k$ such that for every $\boldsymbol{c} \in [2]^k$ the following holds:

    1. *(Linearity)* $\boldsymbol{x_c} = \sum_{i=1}^k \boldsymbol{u}_{i,c_i}$.

2. *(Cross-concept orthogonality)* $(\boldsymbol{u}_{i,1} - \boldsymbol{u}_{i,0}) \perp (\boldsymbol{u}_{j,1} - \boldsymbol{u}_{j,0})$ for all $i, j \in [k]$ with $(i \neq j)$.

*Proof sketch.* GD+CE converges to a max-margin SVM in direction Soudry et al. (2024). Under the degree of freedom of CE, stability implies consistent weight differences across retrainings. The max-margin property with different training sets ensures each datapoint is a support vector for at least one dataset, implying prediction invariance when other concepts vary. Finally, since max-margin SVM weight vectors are parallel to the shortest segment between separable convex sets, appropriate pairing of datasets yields that flipping any concept results in an additive shift, with shift vectors orthogonal across concepts.

Intuitively, linear factorization means that a combination space of $n^k$ elements can be explained using only $n \cdot k$ factors. The orthogonality condition says that factors of concept values belonging to different concepts (e.g., "red" and "square") are orthogonal to each other, but no requirement is placed on the factors of concept values belonging to the same concept (e.g., "red" and "blue"). Additionally, we note that linear factorization in itself is not trivial - the fact that $n^k$ datapoints can be explained using $n \cdot k$ factors does not have to hold for any linearly compositional model. We illustrate this with examples in Appendix C.4.

The datapoint requirement can be interpreted as operating in either (i) a minimal-learning regime for extrapolating to the whole grid (as in Compositional Risk Minimzation framework Mahajan et al. (2025)), where $|T| = 1 + k(n - 1)$ suffices to extrapolate to the whole grid, or (ii) a large-sample regime in which random sampling yields near-complete coverage of the concept space. That is, the conclusions of Proposition 1 hold for $1 + c \leq |T| \leq 1 + 2^{c-1}$ for $c \geq 2$.

> **Takeaway §4.1.** Training under common GD+CE over embeddings to generalize compositionally and stably requires linear factorization and orthogonal of unrelated concept factors.

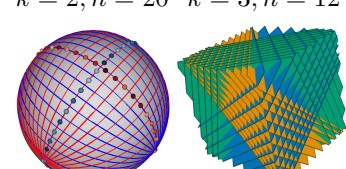

**Unstable Does not transfer**

$\boldsymbol{w}_1$   $\boldsymbol{w}_1$

**Stable Transfers**

$\boldsymbol{w}_1$   $\boldsymbol{w}_1$

Figure 5: **Stable and unstable examples of feature representations.** The top panel shows an unstable configuration, where depending on the sample, the readout either does not transfer or unstably. Bottom panel shows a stable configuration.

## 4.2 PACKING AND MINIMUM DIMENSION

Motivated by the separability axiom, we ask a basic capacity question: what is the minimum embedding dimension $d$ needed to support Divisibility (Desideratum 1), i.e. realize all possible $n^k$ combinations? The following result gives a tight lower bound. Proof and its sketch in Appendix F.

**Proposition 2** (Minimum dimension for linear probes). *For $k$ concepts, each with $n$ values, suppose there exist linear probes that correctly classify each concept value for all $n^k$ combinations from embeddings $f(\boldsymbol{x}) \in \mathbb{R}^d$. Then necessarily $d \geq k$.*

Importantly, the bound is independent of the number of values $n$ per concept, depending only on the number of concepts $k$. This holds whether each factor is discrete or continuous: the proof requires only that we can distinguish any two values per factor, which continuous factors can allow. We illustrate two examples of divisibility in Figure 6: on a sphere and in Euclidean space, though our formal results establish minimal dimensionality only for Euclidean space. Additional visualizations in Figure 14.

$k = 2, n = 20$   $k = 3, n = 12$

Figure 6: **Example geometries under linear compositionality. Left:** 2 concepts ($n = 20$ each) on a 2D sphere. Each colored stripe is the argmax boundary for one concept value; their intersections yield $20^2$ combination cells. **Right:** 3 concepts ($n = 12$ each) in 3D. Colored planes show argmax boundaries; their intersections carve out $12^3$ combination cells. Each boundary is colored according to the concept it belongs to.

> **Takeaway §4.2.** Minimum dimensionality scales with the number of concepts $k$, not values $n$.

## 5 SURVEYING NECESSARY CONDITIONS IN PRETRAINED MODELS

Here, we empirically evaluate the necessary conditions for compositional generalization in pretrained models. We aim to answer the following questions:

**Q3** (Section 5.1)  *Is linear factorization present in pre-trained models?*

**Q4** (Section 5.2)  *Does the degree of linear factorization correlate with compositional generalization?*

**Q5** (Section 5.3)  *Are per-concept difference vectors approximately orthogonal across concepts, as the theory predicts?*

**Q6** (Section 5.4)  *What geometric structure do factors exhibit?*

**Models and datasets.** We evaluate across diverse model families and training regimes: OpenAI CLIP (`ViT-B/32`, `ViT-L/14`), OpenCLIP (`ViT-L/14`), SigLIP (`ViT-L/14` or `L/16`), and SigLIP 2 (`ViT-L/14`). These span different architectures (ViT variants), training objectives (softmax vs. sigmoid), and data scales to assess generality of our findings. We evaluate on three compositional datasets: `PUG-Animal` (Bordes et al., 2023), `dSprites` (Matthey et al., 2017), and `MPI3D` (Gondal et al., 2019), which provide controlled concept variations across different visual domains. Additionally, we also evaluate on a compositional dataset with unnatural noun-adjective pairs (Abbasi et al., 2024) in Appendix D.3.2.

**Recovering the factors from representations.** Given that a linear factorization exists in the representations of a model $f$ as detailed in Section 4.1, we can recover the factors $\{u_{i,j}\}_{i\in[k],j\in[n]}$ by averaging over all the datapoints that share a particular concept value (Trager et al., 2023). For analysis purposes it is sufficient to recover the centered factors. That is, given all centered embeddings $\{f(x_c)\}_{c\in[n]^k}$, the factors can be recovered as $u_{i,j} = \frac{1}{|\{c\in[n]^k:c_i=j\}|} \sum_{c\in[n]^k:c_i=j} f(x_c)$.

### 5.1 LINEAR FACTORIZATION IN PRE-TRAINED MODELS

**Measuring linearity in pre-trained models.** To assess the extent of linearity present in the embeddings, we measure whitened $R^2$ score on the probe span. We (i) project on the probe span to remove information of additional information the embeddings may posess beyond the concepts each dataset exposes, and (2) whiten the embedding space to ensure that the $R^2$ score is not inflated by a few dominant directions. Concretely, given the recovered approximate factors $\{u_{i,j}\}_{i\in[k],j\in[n]}$, the $R^2$ score is computed as

$$R^2 = 1 - \frac{\sum_{x_c\in\mathcal{D}}\left\|f(x_c) - \sum_{i=1}^{k} u_{i,c_i}\right\|_2^2}{\sum_{x_c\in\mathcal{D}} \|f(x_c) - \bar{f}\|_2^2}, \tag{6}$$

where $\mathcal{D}$ is the dataset, and $\bar{f}$ is the mean embedding. Note that a score of $1.0$ indicates perfect linearity. We provide intuition of linear factorization and its relation to the $R^2$ in Appendix C.3, additional justification of whitening in Appendix C.2, and defer the details to Appendix C.1.

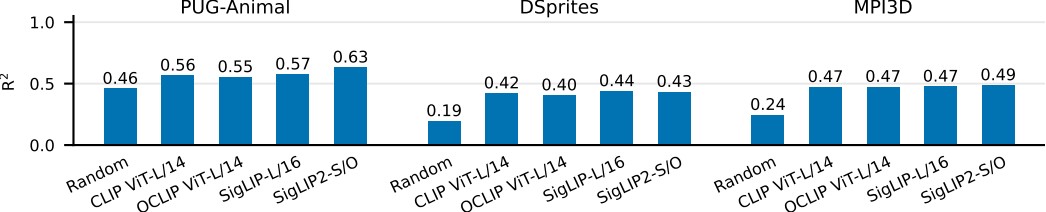

Figure 7: **Linear factorization partly explained current models' embedding spaces.** Bar plots of whitened $R^2$ on three datasets with varying concept/value counts.

**Results.** Figure 7 shows projected $R^2$ scores across models and datasets. Among all the datasets, each model's $R^2$ score is consistently above the random baseline (about $0.4$–$0.6$ vs. $0.19$–$0.46$, respectively). This suggests that embeddings are partially captured by a sum of per-concept components, while still leaving amount of information unexplained. Additionally, we observe that $R^2$ scores are similar across models in scale.

Importantly, we note that the $R^2$ scores, while consistently above random, are far from perfect, indicating that current models only partially satisfy the linear factorization predicted by our theory.

> **Takeaway §5.1.** Embeddings exhibit partial linear factorization ($R^2$ typically 0.4–0.6), explaining a moderate fraction of the variance via per-concept components. The gap from perfect scores highlights a divergence from the ideal compositional structure theory predicts.

## 5.2 COMPOSITIONAL GENERALIZATION AND LINEAR FACTORIZATION

We ask whether the *degree* of linear factorization predicts compositional generalization.

**Metrics and setup.** For each dataset/model, we train linear probes on 90% of all concept combinations and evaluate on the held-out 10% unseen compositions (cf. sampling discussion in Section 4.1). This corresponds to a validity rule $R(T) = 1$ if $|T| = 0.9\,n^k$. We compute *Projected $R^2$* on *whitened* $P_W x$ (Section 5.1) and pair it with a *compositional accuracy* score on the held-out compositions. All encoders from Section 5.1 are included; we use a randomly-initialized OpenCLIP ViT-L/14 model as a baseline by training linear probes on the embeddings. We use linear probing rather than zero-shot classification to avoid prompt-specification issues; nonetheless, the same conclusions hold in the zero-shot setting (discussion and results in Appendix D.3).

Compositional accuracy is computed by training one linear classifier per concept, then averaging each classifier's accuracy on the held-out combinations. For example, DSprites has 6 concepts (shape, orientation, $x$ position, $y$ position, size, and color); we train 6 classifiers and report their mean accuracy on unseen combinations.

**Results.** Across all datasaets *higher Projected $R^2$ coincides with higher compositional accuracy* (Fig. 8). Random encoders consistently occupy the low-$R^2$/low-accuracy corner, indicating the effect is not a dimensionality or scale artifact. This aligns with the linear factorization view: as per-concept components explain more variance, linear probes have cleaner axes to recombine, yielding better compositional transfer.

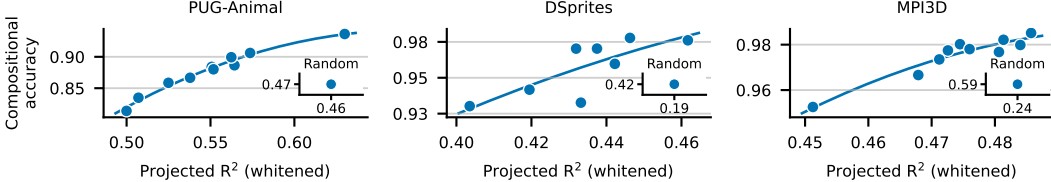

Figure 8: **Linearity in embeddings correlates with compositional generalization.** We show the correlation between projected $R^2$ (linear factorization) and compositional generalization performance across three datasets and multiple vision-language models.

> **Takeaway §5.2:** Linear factorization in pre-trained models correlates positively with compositional generalization performance.

## 5.3 ORTHOGONALITY OF FACTORS

Our theory (Proposition 1) predicts that per-concept difference vectors should be orthogonal *across* concepts under linear factorization, but not necessarily within-concept in generalizing linearly compositional models. We empirically test this prediction by testing orthogonality in two ways: (1) within-concept and (2) across-concept. We defer the details to Appendix D.1.

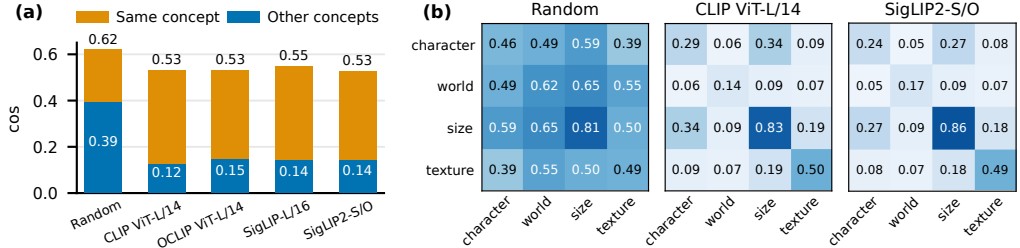

Figure 9: **Pre-trained models exhibit strong within-concept direction similarity and partial orthogonality across concepts. (a)** Aggregated within-concept direction similarity over datasets. **(b)** Pairwise average cosine across concepts. Lower values indicate greater orthogonality between factor vectors.

**Results.** Pretrained encoders exhibit consistently higher direction similarity within concepts than across concepts (Fig. 15): within-concept similarity (a) is around $\approx 0.53$–$0.55$, whereas cross-concept similarity (b) is $\approx 0.12$–$0.15$. The randomly-initialized encoder also exhibits this pattern; however, the across-concept similarity is higher (0.39 on average) compared to pre-trained models.

**Takeaway §5.3:** Pre-trained models exhibit higher direction similarity within concepts than across concepts, with difference vectors across concepts only partially orthogonal and thus deviating from the ideal of perfect cross-concept orthogonality.

## 5.4 DIMENSIONALITY OF FACTORS

Our theory predicts that generalizing liear compositional models require linear factorization of embeddings into per-concept components. When many concepts must coexist in a fixed embedding dimension, each concept's subspace should be low-rank to enable efficient packing (see Section 5.1). Here, we investigate to which extent concept factors in pretrained models are low-dimensional.

**Metrics and setup.** We study factor geometry after projection onto the probe span (as described in Section 5.1). For each concept $i \in [k]$ with value set $\mathcal{C}_i$ ($n_i = |\mathcal{C}_i|$), we aggregate the per-concept factors $\boldsymbol{u}_{i,j}$ for $j \in \mathcal{C}_i$ into a matrix $\boldsymbol{U}_i \in \mathbb{R}^{n_i \times d}$. We then analyze (1) the dimensionality of each concept and (2) how this dimensionality compares across models. To do so, we examine the spectrum of $\boldsymbol{U}_i$ (PCA on its rows) and report the number of principal components required to explain $95\%$ of the variance across values $j$.

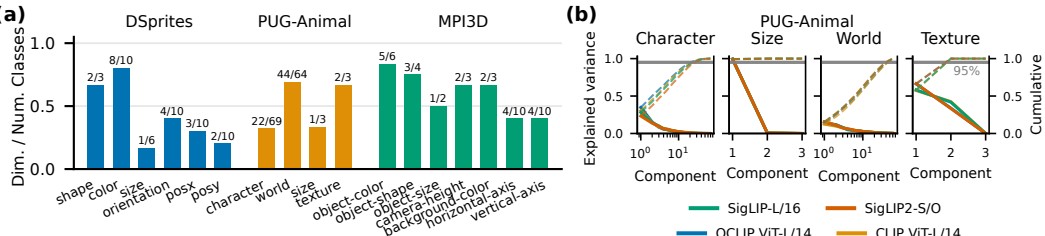

Figure 10: **Dimensionality of factors.** **(a)** Normalized ranks across datasets, and concepts under OpenCLIP L/14 (text above bars shows the effective dimension of the factor and the total number of values for that concept). **(b)** Variance explained in the recovered factors on PUG-Animal dataset over models exhibit high-similarity.

**Results.** Figure 10 shows that most semantic factors lie in low-dimensional subspaces relative to their cardinality (e.g., DSprites size $1/6$, MPI3D vertical-axis $2/5$). Across datasets and models, $\geq 95\%$ of variance is typically captured by one or two PCs, indicating that spectra align closely by concept. Discrete concepts show higher rank, potentially due to being composed of more atomic attributes. Overall, semantic factors are low-rank and geometrically similar across models, while discrete concepts are not strictly low-rank.

We also visualize DSprites factors (orientation, size, $y$-position) in Figure 11. Each subspace is effectively $< 3D$ ($\geq 95\%$ variance in $\leq 2$ PCs). Size and $y$-position trace near-1D path, while orientation forms a smooth 2D curve with small curvature, matching the effective dimensions in Fig. 10.

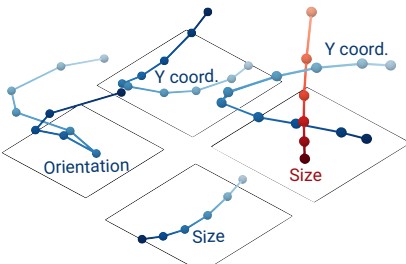

Figure 11: **Geometry of *factors* $\{\boldsymbol{u}_{i,j}\}$ in OpenCLIP ViT-L/14.** The factors are often low dimensional and near co-linear within a concept. Across concepts, the factors are near-orthogonal.

**Takeaway §5.4:** Ordinal and continuous factors are typically low-dimensional (typically $\leq 4D$), while discrete factors show higher rank, potentially because they encode multiple underlying attributes. All models exhibit similar factor geometry across encoders.

## 6 CONCLUSION

We showed that compositional generalization imposes strong structural requirements on neural representations. Under common training with linear heads, our desiderata of divisibility, transferability, and stability force embeddings to factorize additively into per-concept components with orthogonality across concepts, and require dimension at least equal to the number of concepts. Empirically, CLIP and SigLIP families partially exhibit this geometry, and the quality of factorization correlates with compositional generalization performance. These findings clarify when linear structure is not incidental but necessary, providing both theoretical guidance and practical diagnostics for building models that generalize compositionally.

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

# Appendix

## CONTENTS

# A  NOTATION AND SYMBOLS

This section fixes notation and collects basic identities used throughout the appendix.

Table 1: Key notation used in the analysis.

| Notation | Description |
|---|---|
| *Concepts and datasets* | |
| $\mathcal{C} = \mathcal{C}_1 \times \cdots \times \mathcal{C}_k$ | Concept space with $|\mathcal{C}_i| = n$ |
| $\mathcal{X} = \{\boldsymbol{x_c} \mid \boldsymbol{c} \in \mathcal{C}\}$ | Representation space |
| $\mathcal{D}^{\boldsymbol{c}}$ | Cross-dataset of size $1 + k(n-1)$ (see Definition 5) |
| $|S|$ | Dataset size $|S|$ |
| *Counts* | |
| $N_{i,j}(S)$ | Marginal count of concept $i$ taking value $j$ in dataset $S$ |
| *Interventions* | |
| $\boldsymbol{c}(i \to j)$ | Concept index with the $i$-th value set to $j$ |
| $\boldsymbol{x}_{\boldsymbol{c}(i \to j)}$ | Intervened representation with concept $i$ set to $j$ |
| $\bar{c}_i$ | Binary complement $1 - c_i$ (when $\mathcal{C}_i = \{0, 1\}$) |
| *Probes and parameters* | |
| $\boldsymbol{w}_{i,j}^{(\mathcal{D}^{\boldsymbol{c}})}$ | Weight vector for concept $i$, class $j$ |
| $b_{i,j}^{(\mathcal{D}^{\boldsymbol{c}})}$ | Bias term for concept $i$, class $j$ |
| *Factorization objects* | |
| $\boldsymbol{P} \in \mathbb{R}^{d \times d}$ | Projection matrix |
| $\boldsymbol{u}_{i,c_i} \in \mathbb{R}^d$ | Linear factor for concept $i$, value $c_i$ |

## B    EXTENDED DISCUSSION OF RELATED WORK

A large body of literature has studied the usefulness and implications of learning disentangled representations in an unsupervised way (Bengio et al., 2014; Lake et al., 2017). Most commonly, the goal is to learn a generative model, usually through a VAE (Kingma & Welling, 2014), that can compress the data in a disentangled manner, in a way that allows to reconstruct these representations. While shown to be impossible without additional assumptions (Locatello et al., 2019), under weak supervision learning is possible (Shu et al., 2020; Locatello et al., 2020). Measuring the degree of disentanglement in these models is in itself non-trivial and various metrics have been proposed, e.g. by measuring disentanglement by performing interventions on the representations (Higgins et al., 2017; Kim & Mnih, 2018). The DCI framework (Eastwood & Williams, 2018) proposes desiderata of properties disentangled representations should satisfy, namely disentanglement, completeness, and informativeness, and proposes a metric to measure them. Some works also consider what constitutes a good disentanglement (Higgins et al., 2018) and propose a conceptual framing of meaning behind disentangled representations with respect to the data generative process in terms of group actions of transformations.

Abbasi et al. (2024) investigate the role of representation disentanglement in compositional generalization in CLIP models. Using metrics such as DCI, they find that CLIP models with more disentangled text and image representations exhibit higher compositional OOD accuracy on their attribute-object dataset (ImageNet-AO). This work is complementary to ours. Their study explores correlations between disentanglement and compositional generalization by probing CLIP embeddings with respect to the adjective and noun components present in the inputs. For instance, they estimate "attribute" and "object" subspaces by feeding isolated adjectives or nouns into the text encoder, or by generating isolated attributes/objects via a text-to-image model and embedding them with CLIP. However, this approach assumes that CLIP's embedding space is additively decomposed with respect to individual words, an assumption that is not guaranteed to hold. Indeed, Yamada et al. (2024) show that word embeddings in language models are often highly entangled with associated concepts. In contrast, our necessary condition does not rely on word-level decomposition. We posit that models achieving perfect downstream compositional performance must possess linearly factorized representations that separate per-concept components, independent of how an encoder processes individual words. In short, our work provides principled motivation for analyses of representational decomposition, whereas Abbasi et al. (2024) offer an empirical correlation study based on CLIP's emergent disentanglement.

Lippl & Stachenfeld (2025) investigate when a particular form of compositionally structured representations, specifically representations whose similarity depends only on how many underlying components two inputs share, supports downstream compositional generalization. Using kernel theory, they characterize exactly which tasks linear readouts on top of such representations can solve, showing that these models are fundamentally restricted to conjunction-wise additive functions. In contrast, we focus on a specific subclass of compositional tasks: identifying factors of inputs that never co-occur during training. While Lippl & Stachenfeld (2025) characterize what kinds of generalization are possible under a compositional representational structure, we ask the complementary question: given perfect downstream performance on such a task, what representational structure must the model necessarily possess under the desiderata we specify?

# C ADDITIONAL INFORMATION

In this section we expand on linear factorization, make a note on the non-triviality of linear factorization, and expand on the reasoning of using whitening in measuring linear factorization. In Appendix C.1 we summary the overall procedure of measuring linear factorization. In Appendix C.3 we provide an intuition of linear factorization through a simple example. In Appendix C.4 we show that linear factorization is not a trivial property of linearly compositional models, and illustrate a few cases where the representation cannot be decomposed into a sum of per-concept components even under perfect classification.

In contrast to these works, our work is motivated by the same goal of developing systems that exhibit transfer. However, we differ in two key aspects: (1) we do not assume any potential useful structure of the representations for the downstream tasks; instead, our desiderata are strictly based on the downstream performance of the models when learning under a subset of the data space, and (2) we study general NN-based models, which most often include a linear layer, like CLIP and SigLIP, and ask what properties *must* arise if transfer under a subset of the data is possible.

## C.1 TESTING LINEAR FACTORIZATION

Large pre-trained models may encode information beyond the specific concepts in our dataset. To isolate the conceptual structure, we train per-concept linear probes. For each concept $i \in [k]$ and value $j$, we learn a linear probe $\boldsymbol{w}_{i,j}$, form the probe matrix $\boldsymbol{W} \in \mathbb{R}^{m \times d}$, where $m$ is the number of values across all concepts, and project embeddings onto the joint probe span. We do this by first computing the projection matrix $P_{\boldsymbol{W}}$ and then projecting the embeddings onto the joint probe span.

We report *Projected* $R^2$ after projecting embeddings onto the probe span. To prevent trivial high scores from dominant directions, we whiten the embeddings by applying PCA and normalizing to unit covariance. We compute metrics on $P_{\boldsymbol{W}} \boldsymbol{x}$ after PCA-whitening, applying the same transform to data and reconstructions. We elaborate on this below.

## C.2 WHITENING IN MEASURING LINEAR FACTORIZATION

We need to be cautious when assessing the degree of linearity in the representations, otherwise, we may mistake high $R^2$ scores for linear factorization when in fact the representation is not linearly factored. For example, if certain concept values dominate the variance in the representation, the $R^2$ may be inflated. To address this, in the main experiments in Section 5.1 we whiten the representations by applying PCA and normalizing to unit covariance. This ensures that a few dominant directions do not dominate the variance in the representation. If the representations are already linearly factored, this will not affect the $R^2$ score.

We illustrate this through three examples in a hypothetical two-dimensional representation space with two concepts in Figure 12. In the first case (**(a)**) the representation is already linearly factored: each embedding is written as a sum of two concept components without noise. This yields an $R^2$ score of 1; whitening does not change the score.

In the second case (**(b)**) the representation is partly linear, but the noise $\boldsymbol{\epsilon}_{ij}$, independent of the concept values, dominates the overall variance. Since the scale of the noise is generally lower than the scale of the first concept component, the $R^2$ score is high at $0.813$. Whitening, however, removes the dominant direction, and the $R^2$ score drops to $0.509$.

Lastly, in the third case (**(k)**) the representation does not express any information about the second concept, yet the $R^2$ score is still high at $0.991$. Again, whitening reveals the underlying issue and changes the score to $0.564$ due to the noise in the embeddings.

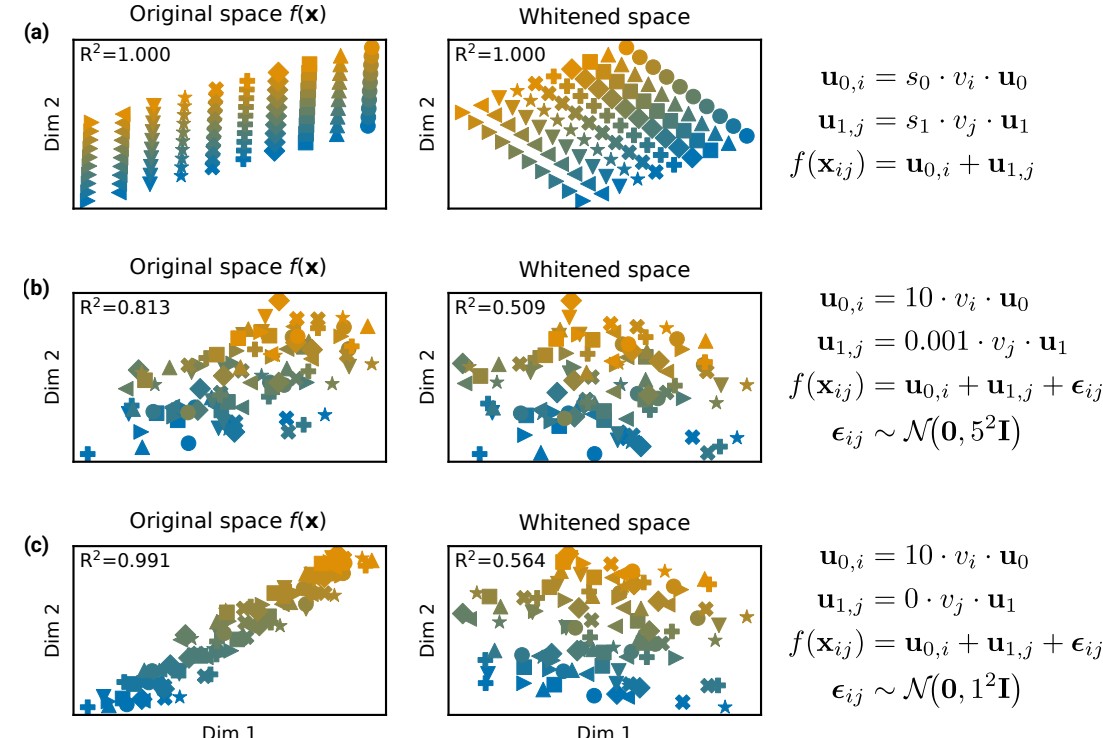

Figure 12: **Whitening in measuring linear factorization**. The representation is not linearly factored, but the $R^2$ score is high due to the dominance of the dominant direction.

### C.3 INTUITION OF LINEAR FACTORIZATION

We measure the extent of linearity present in the embeddings through the $R^2$ score. Intuitively, the score quantifies how well the representation can be decomposed into a sum of per-concept components. Recall from Definition 1 that we assume a presence of $k$ concepts, each of which can take any of the $n$ values. A value of $R^2 = 1$ indicates that the representation can be perfectly decomposed into a sum of per-concept components.

We illustrate a few examples to give intuition. We consider a two-dimensional representation space with two concepts ($k = 2$). In the first case, we consider a case of 24 values per concept ($n = 24$). In the second case, we consider a case of 6 values per concept ($n = 6$). In both cases the reported $R^2$ are w.r.t. the whitened space.

The first case (Figure 13, **(a)**) exhbits perfect linearity in the embeddings with $R^2 = 1$. In this case, the $n^2 = 24^2 = 576$ can be perfectly generated using only $2 \cdot 24 = 48$ vectors in $\mathbb{R}^2$. The second and third columns of the plot show the approximations of the underlying factors $\boldsymbol{u}_{0,i}, \boldsymbol{u}_{1,j}, i, j \in [n]$. As expected, using these approximate factors allow us to perfectly reconstruct the representation, shown in the fourth column.

The second case (Figure 13, **(b)**) exhibits lower degree of linearity with $R^2 = 0.53$. As such, we cannot perfectly reconstruct the representation using only the approximate factors, as shown in the last column of the plot.

### C.4 NON-TRIVIALITY OF LINEAR FACTORIZATION OF LINEARLY COMPOSITIONAL MODELS

Recall that linearly compositional models (though not necessarily generalizable ones), as defined in Definition 3, admit a set of probes that can perfectly classify all inputs in the grid $\mathcal{C}$. Proposition 1 shows that linearly compositional models must exhibit linear factorization. This naturally raises the converse question: does the mere existence of a set of perfect linear classifiers imply linear factorization? We answer in the negative.

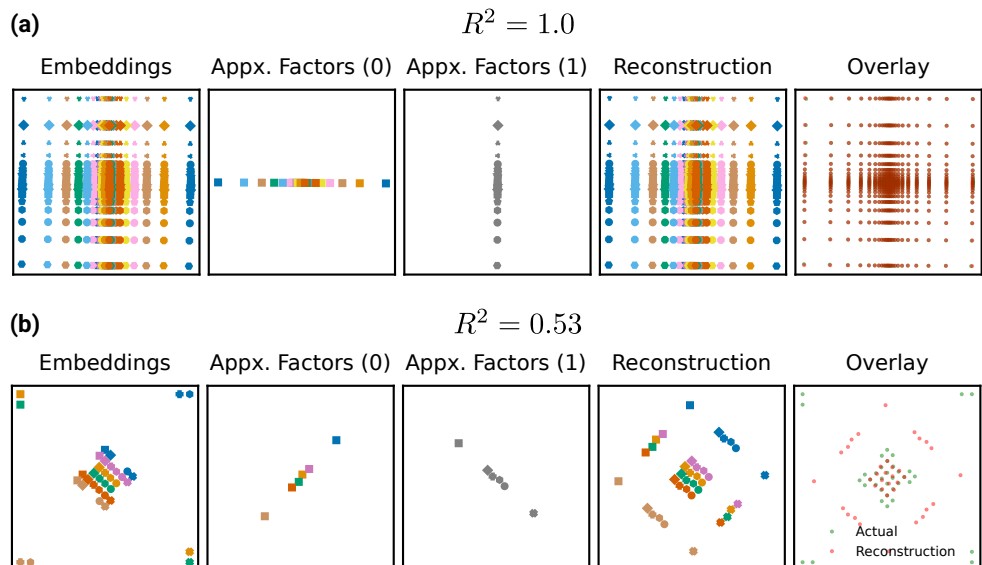

Figure 13: **Intuition of linear factorization**. In **(a)** the representations can be perfectly reconstructed by a set of per-concept components, while in **(b)** they are insufficient to reconstruct the representation. Refer to the text for more details.

The intuition is as follows. As per Desideratum 1, linearly compositional models need to divide the representation space into all possible combinations of concept values, $n^k$ of them. Each region within the $n^k$ partitions must contain the corresponding combination of concept values. Under linear factorization, the degrees of freedom of the embeddings within each cell are low, yielding an $R^2$ score of 1. However, even if linear factorization initially holds, the embeddings can generally be perturbed to violate the linear factorization constraint while still being contained within the correct cell.

To illustrate this point, we consider two general cases: (i) the number of concepts is equal to the dimension of the embeddings ($k = d$), and (ii) the number of concepts is less than the dimension of the embeddings ($k < d$). As detailed in Section 4.2, case (i) is tight (the dimension cannot be further reduced), while case (ii) is not. In both cases we assume two concepts and an embedding space that admits two linear probes, one for each concept. Additionally, in both cases we illustrate separately the argmax regions where a certain concept value is predicted ($\mathcal{R}_{i,j}, i \in [2], j \in [n]$), and the region where a certain combination of concept values is predicted ($\mathcal{R}_{0,j} \cap \mathcal{R}_{1,k}, j, k \in [n]$), as per Desideratum 1).

The first concept values' regions in the embedding space are shown in blue, while the second concept values' regions are shown in orange.

**Case (i):** $k = d$. In Figure 14, **(a), (b)** we show two cases that exhibit perfect linear classification. In **(a)** a few outliers violate the linearity of the representation, which is also reflected in the $R^2 = 0.53 < 1$. In **(b)** the argmax regions are highly irregular, and the majority of the embeddings are almost intersecting the decision boundaries, resembling an extremely brittle embedding space susceptible to adversarial attacks, though the classification accuracy is still 100%.

**Case (ii):** $k < d$. In Figure 14, **(k), (d), (e)** we show three cases that exhibit perfect linear classification, but with linearity scores ranging from $R^2 = 0.32$ to $R^2 = 0.83$. Because of the higher degrees of freedom, the embeddings enjoy even more space to be perturbed while still exhibiting perfect linear classification.

Overall, these points illustrate that linear factorization is not a trivial property of linearly compositional models, even when perfect classification holds.

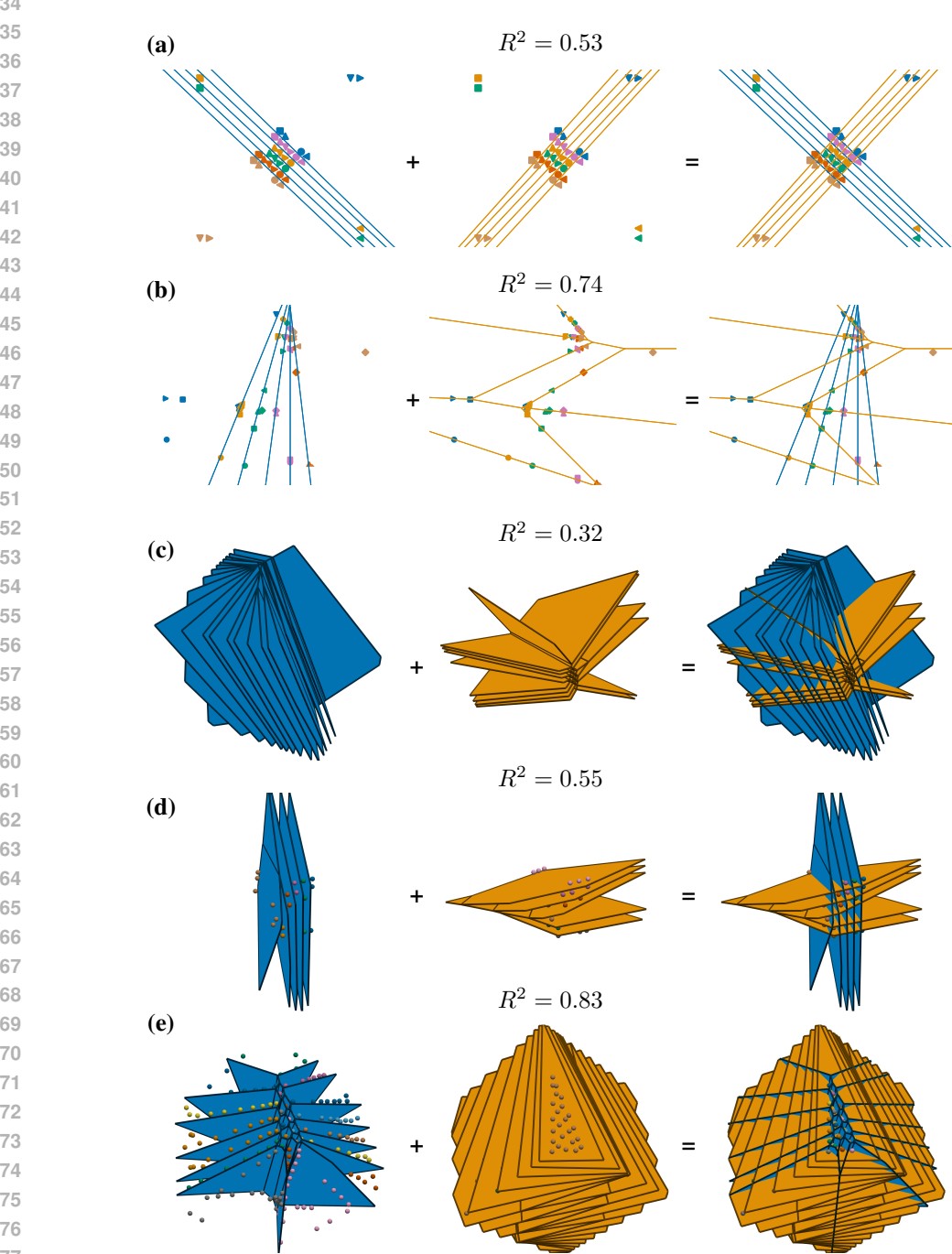

Figure 14: **Counterexamples of linear factorization under perfect classification**. The two concepts are linearly separable, but the representation cannot be decomposed into a sum of two concept components. Each subfigure shows the embedding space overlaid with three columns of argmax regions: the first column shows $\mathcal{R}_{0,i}, i \in [n]$ (shown in blue), the regions where the first concept values are predicted; the second column shows $\mathcal{R}_{1,j}, j \in [n]$ (shown in orange), the regions where the second concept values are predicted; and the third column shows $\mathcal{R}_{0,i} \cap \mathcal{R}_{1,j}, i, j \in [n]$, the joint argmax regions where specific combinations of concept values are predicted. **(a), (b)** show embeddings for two concepts (color and shape) in $\mathbb{R}^2$ ($k = d = 2$). **(d), (e)** show embedding points colored by the first concept value, all for two concepts in $\mathbb{R}^3$ ($k = 2, d = 3$). See text for details.

# D  ADDITIONAL EXPERIMENTAL RESULTS

In this section we provide additional experimental results discussed in the main text.

## D.1  ORTHOGONALITY OF FACTORS

**Setup.** For each dataset/model, we extract image embeddings $x_c$ and restrict analysis to the probe-usable subspace by projecting as in Section 5.1, that is, for each dataset, we compute $\hat{x}_c := P_W x_c$. For concept pair $i, j \in [k]$ with value sets $\mathcal{C}_i, \mathcal{C}_j$, we estimate per-concept difference vectors by averaging differences across concept factors. Concretely, for any pair $(v, v') \in \mathcal{C}_i \times \mathcal{C}_j$, we define

$$d_{i,j,(v,v')} := u_{i,v} - u_{j,v'}, \quad \tilde{d}_{i,j,(v,v')} := \frac{d_{i,j,(v,v')}}{\|d_{i,j,(v,v')}\|}. \tag{7}$$

We measure orthogonality via absolute cosine between difference vectors (lower $|\cos| \Rightarrow$ greater orthogonality). For any concepts $i \neq j$, we define

$$\mathrm{Orth}(i,j) := \frac{1}{|\mathcal{C}_i||\mathcal{C}_j|} \sum_{a \in \mathcal{C}_i} \sum_{b \in \mathcal{C}_j} |\langle \tilde{d}_{i,a}, \tilde{d}_{j,b}\rangle| \quad \text{and} \quad \mathrm{Orth}(i,i) := \frac{1}{|\mathcal{C}_i|(|\mathcal{C}_i| - 1)} \sum_{\substack{a,b \in \mathcal{C}_i \\ a \neq b}} |\langle \tilde{d}_{i,a}, \tilde{d}_{i,b}\rangle|$$

We report $\mathrm{Orth}(i,i)$ as *within-concept direction similarity* and $\mathrm{Orth}(i,j)$ for $i \neq j$ as *across-concept orthogonality*.

We present the complete experimental results here.

In Figure 15, we show the orthogonality of the factors for four models, including a randomly-initialized model, and three datasets.

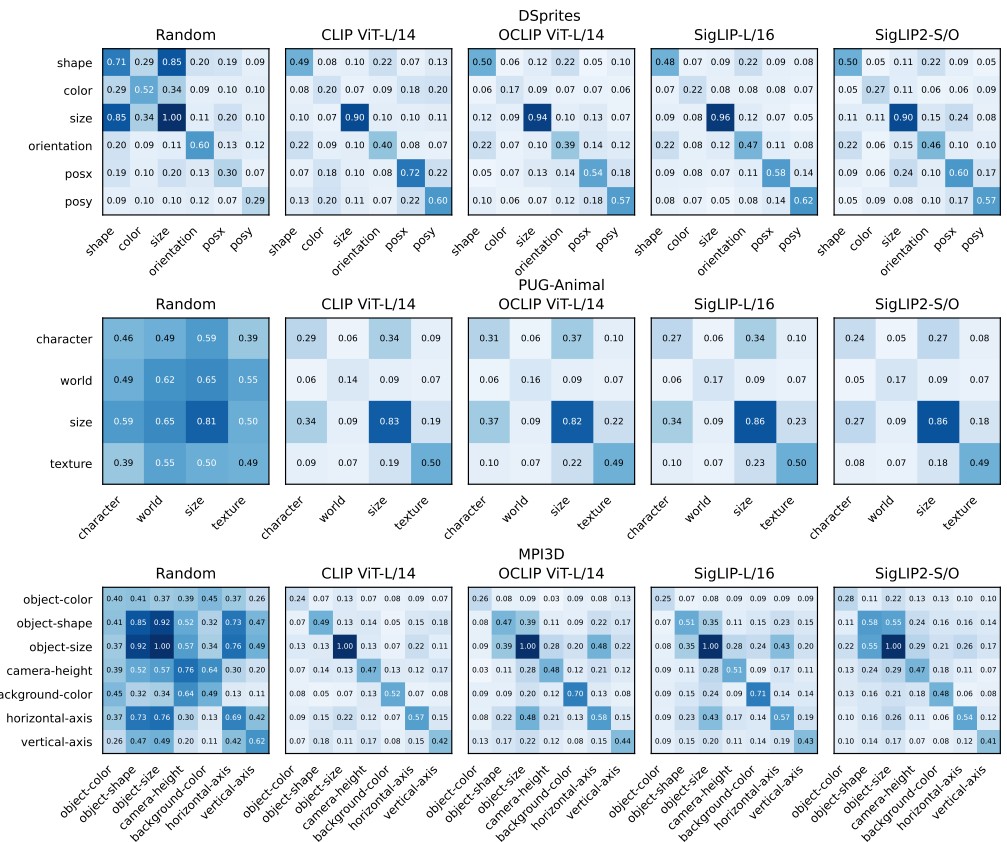

Figure 15: **Orthogonality of factors.** We shot the orthogonality of the factors for four models, including a randomly-initialized model, and three datasets.

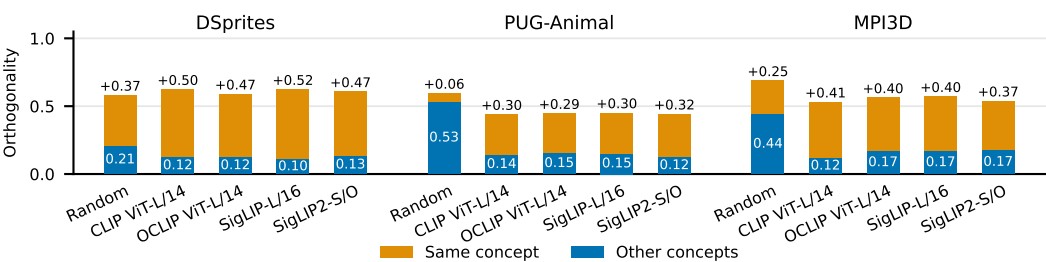

Figure 16: **Orthogonality between factors.**

We show an aggregate view of this result when comparing orthogonality between values of the same and different concepts in Figure 16.

### D.2 DIMENSIONALITY OF FACTORS

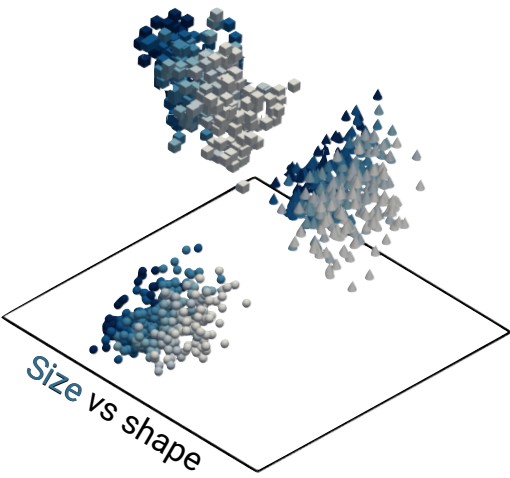

Figure 17: **Geometry of *datapoints* in OpenCLIP ViT-L/14.** We show the span of the joint features of OpenCLIP ViT-L/14.

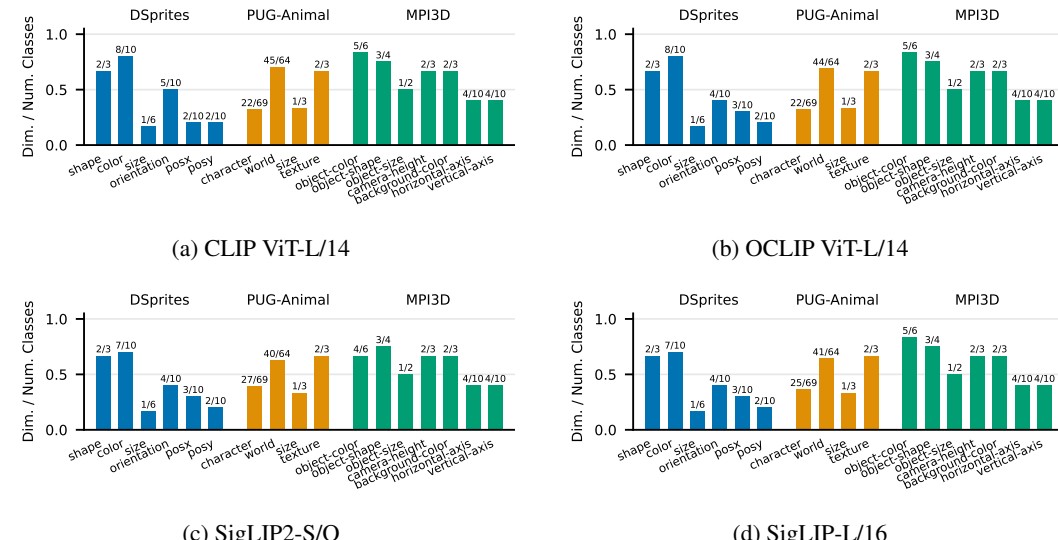

(a) CLIP ViT-L/14

(b) OCLIP ViT-L/14

(c) SigLIP2-S/O

(d) SigLIP-L/16

Figure 18: Dimensionality results computed as the number of SVD factors required to reach 95% explained variance, per dataset.

### D.3 EXPERIMENTS USING TEXT ENCODERS AS PROBES

In the main text (Section 5.1), we analyzed the factors of the models by training linear probes on the image embeddings using gradient descent with cross-entropy. This was done for two reasons: (1) to handle concepts that are difficult to express as text prompts (e.g., visually complex backgrounds or continuous attributes like size or orientation), and (2) to avoid potential misalignment between the text and vision modalities, where the text encoder must accommodate many visual categories, potentially leading to suboptimal performance for certain domains. Here, we ask what happens when we do not take into account these problems and instead rely on the linear probes that the text encoder already produces.

In this section, we provide analogous analyses to those in the main text, but using the text encoder as probes instead of external linear probes for two datasets: PUG-Animal and ImageNet-AO. We use these datasets for two reasons: (1) their concepts and values map naturally to text prompts, and (2) the datasets were released after the CLIP models and exhibit many unnatural concept combinations unlikely to have appeared in text captions during pre-training, and not present in the visual training data.

#### D.3.1 EXPERIMENTS ON PUG-ANIMAL

**Setup.** Four concepts are exposed: character, background, scale, and texture. For each character we parse the character name into a set of words and use prompts of the form "A picture of a <character>". For each background, we use prompts of the form "A picture of a <background>" (detailed in Table 2).

We map numeric scale values and texture labels to descriptive prompt templates for evaluating the models. Specifically, for scale, we use:

- $0.7 \rightarrow$ "A picture of a small object"
- $1.0 \rightarrow$ "A picture of a medium-sized object"
- $1.3 \rightarrow$ "A picture of a large object"

For textures, we use the following mappings:

- "Sky" $\rightarrow$ "A picture of an object in sky texture"
- "Grass" $\rightarrow$ "A picture of an object in grass texture"
- "Asphalt" $\rightarrow$ "A picture of an object in asphalt texture"

Table 2: **Mapping from class names to clean prompt names for PUG-Animal experiments.**

| Original Name | Prompt Name |
|---|---|
| Desert | a desert |
| Tableland | a tableland |
| EuropeanStreet | a European street |
| OceanFloor | the ocean floor |
| Racetrack | a racetrack |
| Ruins | ancient ruins |
| TrainStation | a train station |
| BusStationInterior | the interior of a bus station |
| BusStationExterior | the exterior of a bus station |
| IndoorStairs | indoor stairs |
| Circus | a circus |
| BoxingRing | a boxing ring |
| Mansion | a mansion |
| ShoppingMall | a shopping mall |
| ConferenceRoom | a conference room |
| VillageOutskirt | a village outskirt |
| VillageSquare | a village square |
| Courtyard | a courtyard |
| Forge | a forge |
| Library | a library |
| Museum | a museum |
| Gallery | an art gallery |
| Opera | an opera house |
| Restaurant | a restaurant |
| RuralAustralia | rural Australia |
| AustraliaRoad | a road in Australia |
| ShadyRoad | a shady road |
| SaltFlats | salt flats |
| Castle | a castle |
| Temple | a temple |
| Snow | a snowy landscape |
| Grass | a grassy field |
| DryGrass | a dry grassland |
| Forest | a forest |

These prompt templates are used to generate the corresponding text enbeddings for each concept, matching exactly with the setup of the experiments in the main text.

Concretely, for each concept value $j \in [n]$, we pass the prompt template through the text encoder $g$ to obtain a ($\ell_2$-normalized) probe vector $\boldsymbol{w}_{i,j} = g(p_{i,j}) \in \mathcal{Z}$, as detailed in Section 3.4.

**Linearity of factors and generalization.** We show the projected $R^2$ and average accuracy on all concept combinations on PUG-Animal across models in Figure 19 when using the text encoder as probes. Models exhibiting higher linearity of representations generally exhibit higher accuracy on the full dataset. This coincides with the observations in the main text (Section 5.1); random baseline achieves low projected $R^2$ and accuracy.

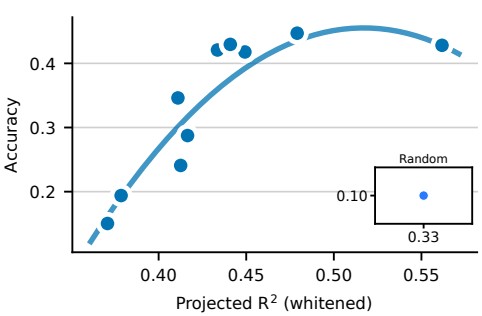

Figure 19: **Projected $R^2$ vs accuracy on PUG-Animal across models.** Higher projected $R^2$ coincides with higher accuracy on the full dataset. The probes are extracted from the text encoder.

**Orthogonality of the factors.** For each of the concepts, we compute the linear factors as detailed in the main text (Section 5.1) with the text encoder as probes. We compute the within- and across-concept orthogonality as detailed in Appendix D.1 and illustrate the results in Figure 20 for each of the models.

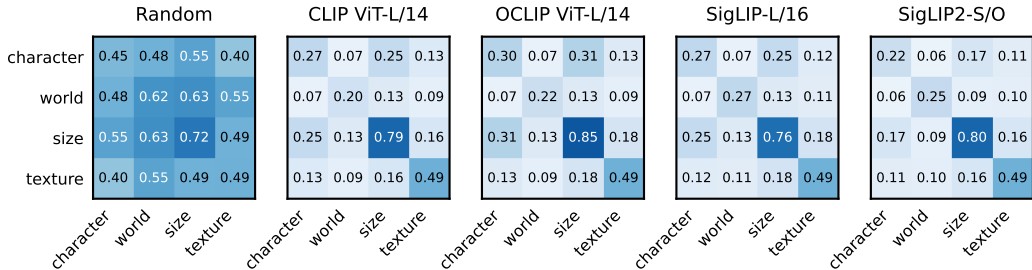

Figure 20: **Orthogonality of the factors on PUG-Animal.** Heatmaps show pairwise cosine similarity between factors for the four PUG-Animal concepts (character, world, size, texture) across multiple models. The factors are more orthogonal across concepts (off-diagonal) than within concepts (diagonal). The random baseline does not generally show this pattern.

For all evaluated models, we observe the same orthogonality pattern: the factors are more orthogonal across concepts (off-diagonal) than within concepts (diagonal). The average cosine similarity for the random baseline is higher (around 0.5) both within and across concepts.

We also note the qualitative similarity between the factors to the case when probes were trained on 90% of the concept combinations (Figure 15, second row).

**Qualitative examples.** We illustrate some of the highest- and lowest-scoring samples in terms of $R^2$ for the SigLIP2 model in Figure 21. We note that high-scoring samples generally depict clean scenes where the character and its size and texture are easier to discern compared to the lower-scoring samples.

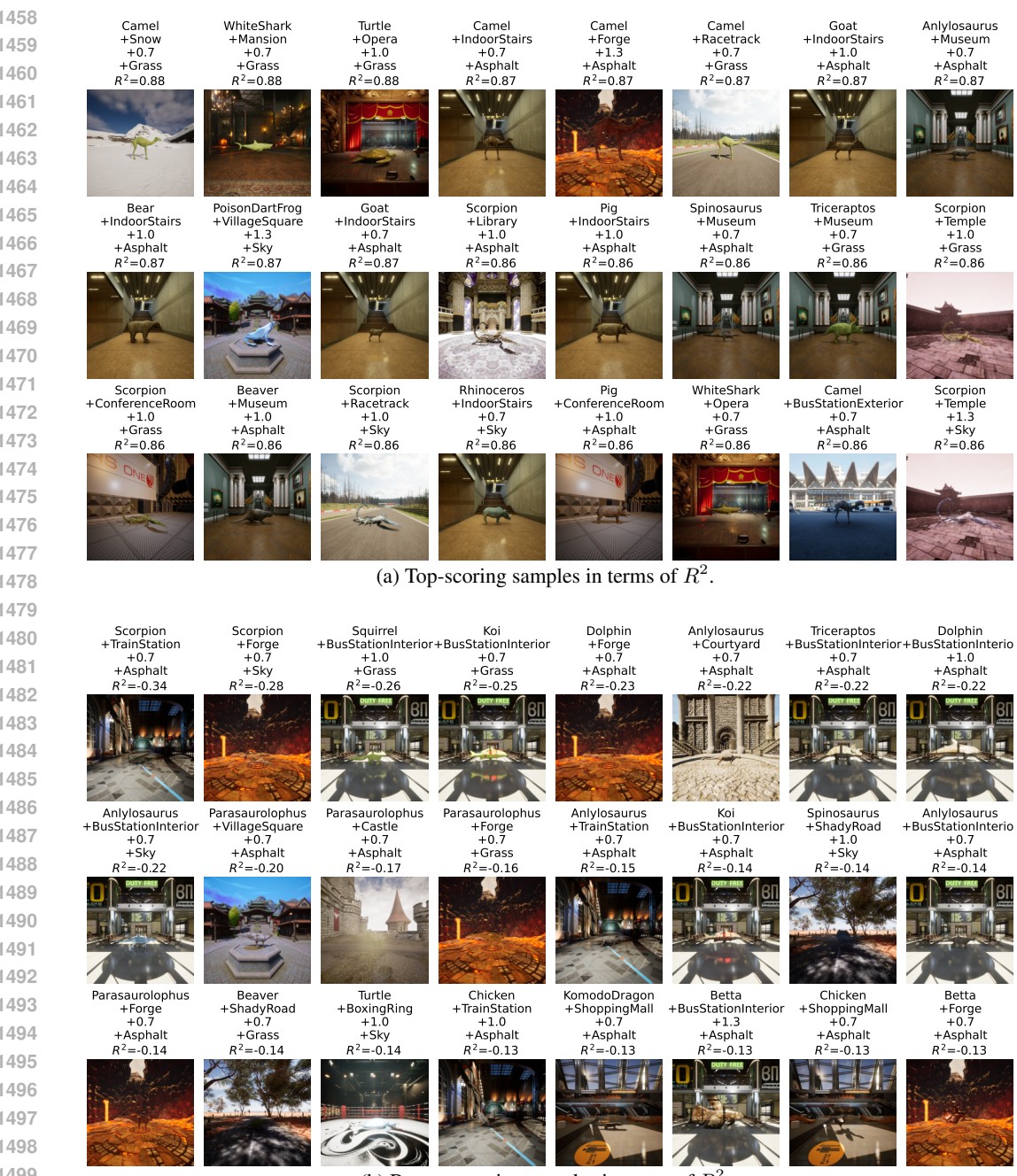

(a) Top-scoring samples in terms of $R^2$.

(b) Bottom-scoring samples in terms of $R^2$.

Figure 21: **Qualitative examples of the top- and lowest-scoring samples in PUG-Animal for the SigLIP2 model.** Each sample shows its character name, world name, size value (0.7 corresponds to "small", 1.0 corresponds to "medium", 1.3 corresponds to "large"), texture name, and its $R^2$ score.

### D.3.2 EXPERIMENTS ON IMAGENET-AO

We additionally perform experiments on a coarse-captioned dataset ImageNet-AO Abbasi et al. (2024), where each image sample has an associated caption composed of an adjective and a noun.

The experiments here are slightly disimilar from the main experiments in Section 5.1, for a few reasons: (1) scracity of per-combination data, (2) inability to train linear probes, (3) noisy/ambigious data, and (4) coarse categories. Regardless, our framework still applies.

**Dataset description.** The dataset contains images described by an adjective and a noun. There are around 80 unique adjectives and over 600 unique nouns. To make the analysis balanced, we work with the dataset restricted to the most common 80 nouns and adjectives. Each potential combination of adjective and noun may have between 0 and 6 images. The dataset is thus sparse, and many of the potential combinations are not observed in the dataset. This results in a total of 3243 datapoints. We illustrate the sparsity and the pairs we work with in Figure 22.

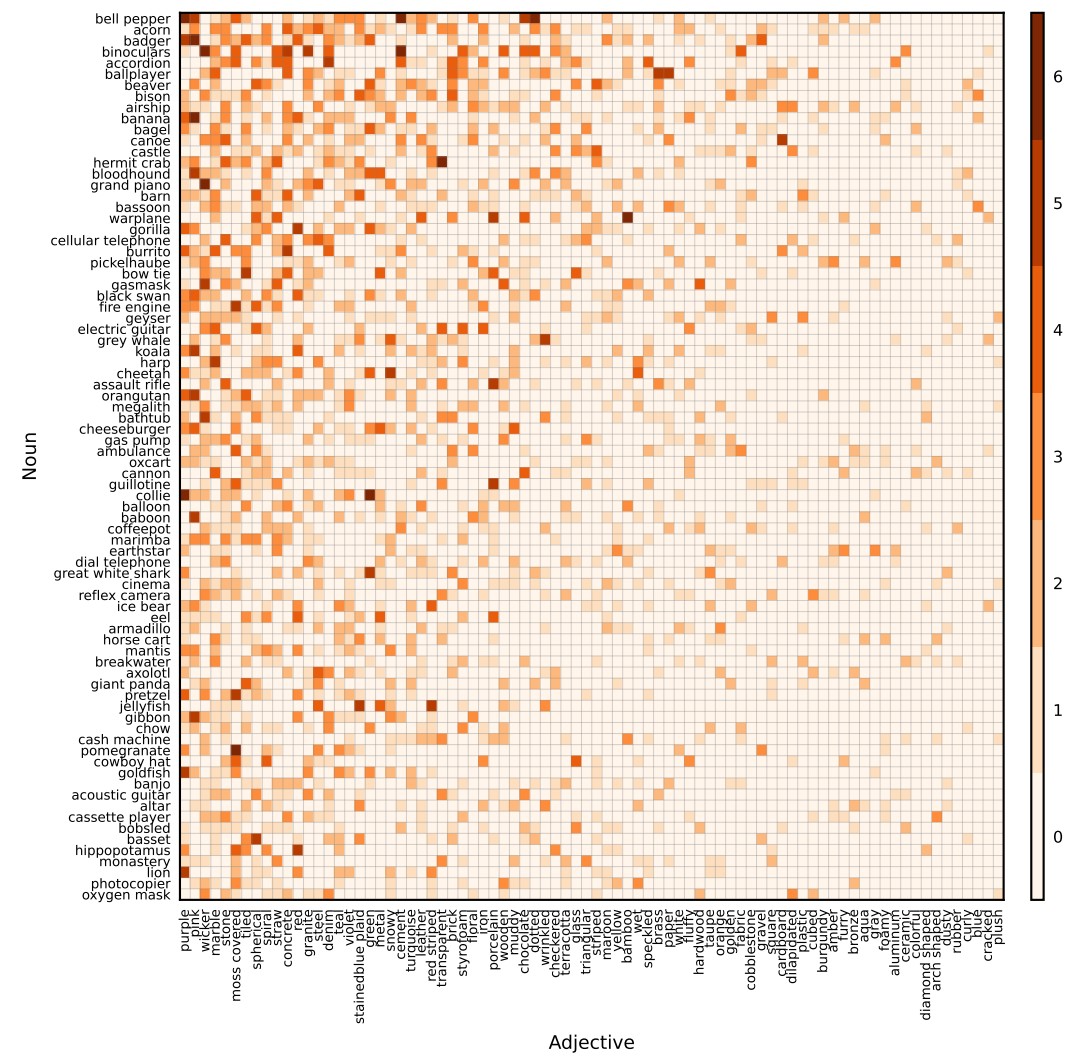

Figure 22: **Adjective-noun count matrix for ImageNet-AO (Abbasi et al., 2024) of the top 80 adjectives and nouns.** The the adjective-noun pairs are sparse, and many of them are not observed in the dataset.

**General setup.** Due to limited availabilty of the data samples, we *do not* train linear probes. Because each sample is associated with a (noun, adjective) combiantion, we instead use the probes from the text encoder to assess the performance of the models (as detailed in the main text in Section 3.4). Concretely, we pass captions in the style of "A picture of <noun>" in the case of noun, and "A picture showing <adjective>" in the case of adjective, through the text encoder.

Because of imbalance and sparsity, we cannot rely on averaging to extract the factors as done in Section 5.1. Instead, we follow Uselis et al. (2025) and solve a linear system of equations to recover the factors. Concretely, we construct a design matrix $A \in \{0, 1\}^{3243 \times 80 \cdot 2}$ where each row corresponds to a sample, and each column corresponds to a either the presenece of a noun (if the column index $< 80$) or the presenece of an adjective (if the column index $\geq 80$). The matrix was of full rank

$2 \cdot 80 - 1$. Then, we solve the linear system $\boldsymbol{A} \begin{bmatrix} \boldsymbol{u}_{\text{noun}} \\ \boldsymbol{u}_{\text{adj}} \end{bmatrix} = \boldsymbol{X}$ to recover the factors $\boldsymbol{u}_{\text{noun}} \in \mathbb{R}^{80 \times d}$

and $\boldsymbol{u}_{\text{adj}} \in \mathbb{R}^{80 \times d}$, where $d$ is the dimension of the representation space, and $\boldsymbol{X} \in \mathbb{R}^{3243 \times d}$ is the cenetered image embeddings. We show the whitened $R^2$ scores. The remaining procedure in the analysis follows Section 5.1.

**Linearity of factors and generalization.** We show the projected $R^2$ vs accuracy on ImageNet-AO across models in Figure 23. As seen in the main text (Section 5.1), higher projected $R^2$ coincides with higher accuracy on the full dataset. Importantly, the random baseline achieves substantially lower projected $R^2$ (less than 0.1) compared to the other models.

**Orthogonality of the factors.** To substantiate the claims of orthogonality of factors across concepts, we extract the factors for all the models as detailed in the setup above. Concretely, for each of the attribute factor $\boldsymbol{u}_i, i \in [80]$ and noun factor $\boldsymbol{u}_j, j \in [80]$, within- and across-concept orthogonality as detailed in Section 5.1.

We illustrate the results in Figure 24. For all of the evaluated models the same pattern of orthogonality is observed: the factors are more orthogonal across concepts than they are within concepts. For example, for the CLIP ViT-L/14 model, the within-concept similarity on average is 0.10 between nouns, and 0.14 between adjectives, while the average cosine similarity across concepts is 0.07. The random baseline on average yields 0.49 cosine similarity both across and within concepts.

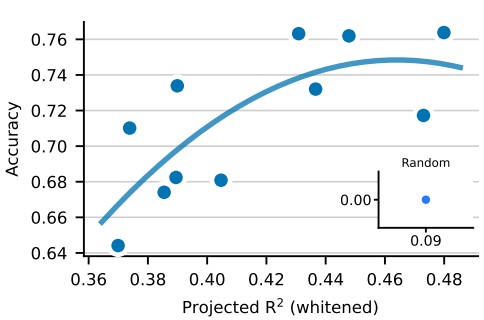

Figure 23: **Projected $R^2$ vs accuracy on ImageNet-AO across models.** Higher projected $R^2$ coincides with higher accuracy on the full dataset. Linear probes were not trained here, and the results are computed using the text encoder.

Interestingly, all of the non-random models exhibit surprising degree of similarity in terms of the cosine similarities. For example, CLIP ViT-L/14 and OpenCLIP ViT-L/14 on average exhibit almost the same cosine similarity within and across concepts, differing only in the noun-noun cosine similarity (0.10 vs 0.11, respectively). These results support the notions of universality between models as argued by the Platonic Representation Hypothesis (Huh et al., 2024), and empirically observed in Universal Sparse Autoencoders (Thasarathan et al., 2025).

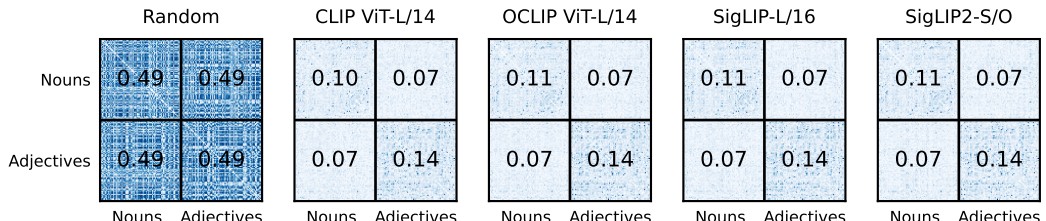

Figure 24: **Orthogonality of the factors on ImageNet-AO.** We show the cosine similarity of the factors for the SigLIP2 model on ImageNet-AO; we separate the first concept (nouns) from the second concept (adjectives) and show average similarity across each $2 \times 2$ block. The factors are more orthogonal across concepts than they are within concepts. The random baseline does not show this pattern.

**Qualitative examples.** To understand the results deeper, we show the qualitative examples of the top- and lowest-scoring samples in ImageNet-AO for the SigLIP2 model in Figure 25. The top-scoring samples show high degree of projected $R^2$ scores (generally $> 0.75$), and correctly depict the adjective and noun of the sample. Even there, however, some samples are incorrectly predicted by the model, suggesting a potential lack of alignment between the image and text encoders[1].

---

[1]This was less of an issue in the main experiments because the image embeddings were analysed using linear probes.

The lowest-scoring samples show low degree of projected $R^2$ scores (generally $< 0.10$), and are often incorrectly predicted by the model. Few of the samples appear to be incorrectly labeled (e.g. first image depicting a orangutan as a gorilla), while some are correctly classified by the model but show a lack of factorization.

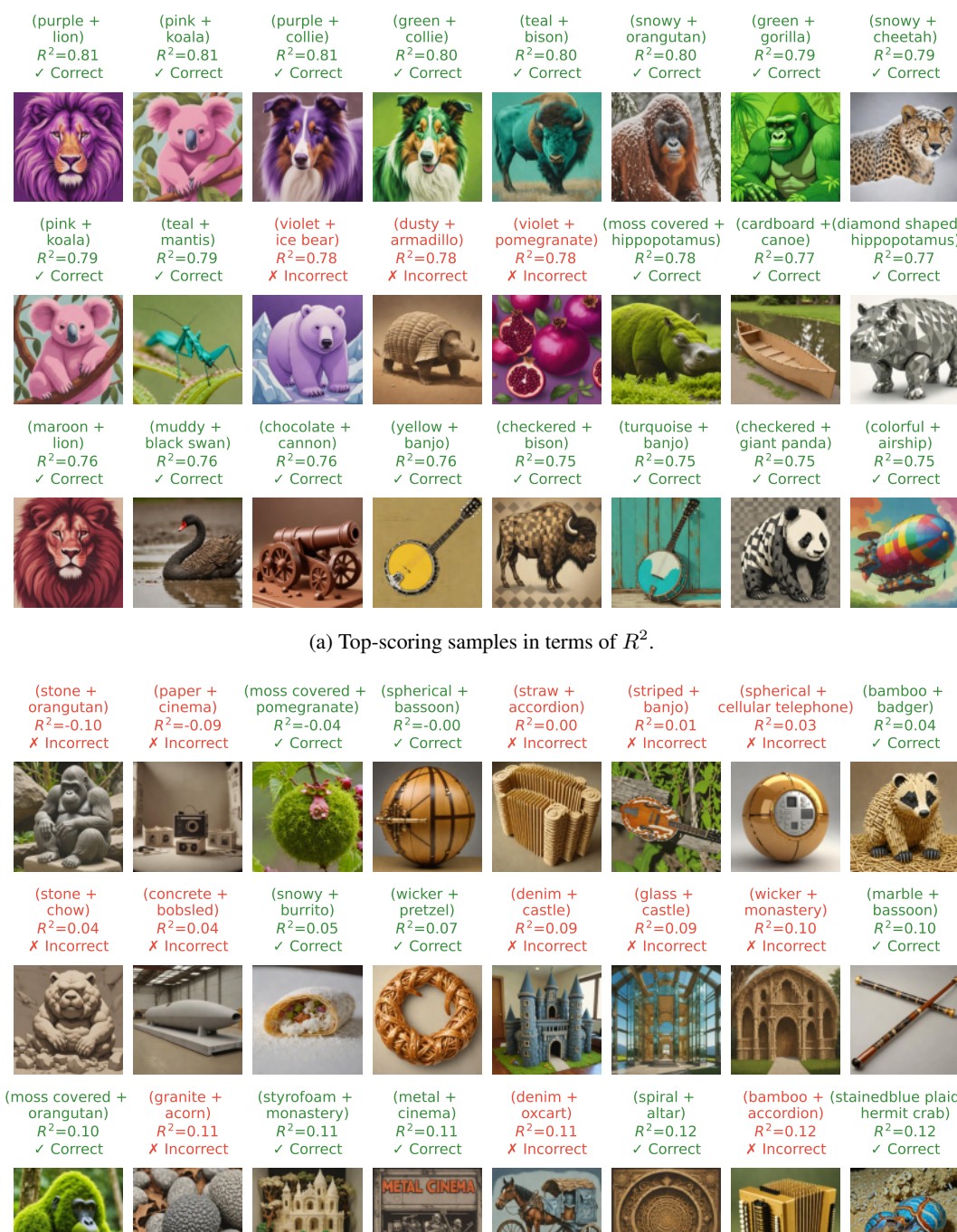

(a) Top-scoring samples in terms of $R^2$.

(b) Bottom-scoring samples in terms of $R^2$.

Figure 25: **Qualitative examples of the top- and lowest-scoring samples in ImageNet-AO for the SigLIP2 model.** Each sample shows its adjective and noun, its $R^2$ score, and whether it was correctly classified by the model. Note that both top- and lowest-scoring samples may be either correctly or incorrectly classified by the model.

# E  SUFFICIENCY OF LINEAR FACTORIZATION FOR COMPOSITIONALLY GENER-ALIZATION

A complementary analysis we provided is on the sufficient conditions for generalizing compositionally. Here, we detail the key results for recovering the factors $\boldsymbol{u}$ from representations that already possess linear factorization.

We first note the minimal dataset setting using the notion of a cross dataset, defined below.

**Definition 5** (Cross dataset at $\boldsymbol{c}$). Given a concept space $\mathcal{C} = \mathcal{C}_1 \times \cdots \times \mathcal{C}_k$, we say that a dataset $\mathcal{D}^{\boldsymbol{c}}$ is a cross-dataset at $\boldsymbol{c} \in [n]^k$ if:

1. It contains only samples that vary one concept at a time around the center $\boldsymbol{c}$:
$$\mathcal{D}^{\boldsymbol{c}} = \big\{ (c_1', c_2, \ldots, c_c) : c_1' \in [n] \big\} \cup \cdots \cup \big\{ (c_1, c_2, \ldots, c_c') : c_c' \in [n] \big\}.$$

2. Its size is $1 + k(n-1)$,

3. It satisfies the diversity condition: $\operatorname{rank}(A^{\mathcal{D}^{\boldsymbol{c}}}) = 1 + k(n-1)$.

**Proposition 3** (Uniqueness up to concept-wise shifts). Let the concept space be $\mathcal{C} = \mathcal{C}_1 \times \cdots \times \mathcal{C}_c$ and assume *linear factorisation* holds, i.e. for every full combination $(v_1, \ldots, v_c) \in \mathcal{C}$ we observe an embedding

$$f(v_1, \ldots, v_c) = \sum_{i=1}^{k} \boldsymbol{u}_{i, v_i},$$

where $\boldsymbol{u}_{i,v} \in \mathbb{R}^d$ is the (unknown) vector for value $v \in \mathcal{C}_i$.

Suppose $\{\boldsymbol{a}_{i,v}\}$ and $\{\boldsymbol{b}_{i,v}\}$ are *any two* families of vectors that satisfy the same equations:

$$\sum_{i=1}^{k} \boldsymbol{a}_{i,v_i} = \sum_{i=1}^{k} \boldsymbol{b}_{i,v_i}, \quad \text{for every } (v_1, \ldots, v_c) \in \mathcal{C}.$$

Then there exist vectors $\boldsymbol{s}_1, \ldots, \boldsymbol{s}_c \in \mathbb{R}^d$ with the single constraint $\sum_{i=1}^{k} \boldsymbol{s}_i = \boldsymbol{0}$ such that

$$\boldsymbol{b}_{i,v} = \boldsymbol{a}_{i,v} + \boldsymbol{s}_i \quad \text{for all } i \in \{1, \ldots, k\},\ v \in \mathcal{C}_i.$$

Hence the solution space of the factorisation equations is $(k-1)d$-dimensional: one free shift vector $\boldsymbol{s}_i$ per concept, minus one global zero-sum constraint.

*Proof.* Let $\delta_{i,v} := \boldsymbol{b}_{i,v} - \boldsymbol{a}_{i,v}$. Subtracting the two versions of the factorisation identity gives

$$\sum_{i=1}^{k} \delta_{i,v_i} = \boldsymbol{0} \qquad \text{for every } (v_1, \ldots, v_c) \in \mathcal{C}.$$

Fix any reference value $v_i^0 \in \mathcal{C}_i$ for each concept and set $\boldsymbol{s}_i := \delta_{i,v_i^0}$. Evaluating the previous display at the reference combination $(v_1^0, \ldots, v_c^0)$ yields

$$\sum_{i=1}^{k} \boldsymbol{s}_i = \sum_{i=1}^{k} \delta_{i,v_i^0} = \boldsymbol{0}.$$

Now fix an index $j \in \{1, \ldots, k\}$ and choose an arbitrary value $v \in \mathcal{C}_j$. Evaluate the identity $\sum_{i=1}^{k} \delta_{i,v_i} = \boldsymbol{0}$ at the combination $(v_1^0, \ldots, v_{j-1}^0, v, v_{j+1}^0, \ldots, v_c^0)$. Then

$$\boldsymbol{0} = \sum_{i=1}^{k} \delta_{i,v_i} = \delta_{j,v} + \sum_{i \neq j} \delta_{i,v_i^0} = \delta_{j,v} + \sum_{i \neq j} \boldsymbol{s}_i.$$

Using $\sum_{i=1}^{k} \boldsymbol{s}_i = \boldsymbol{0}$, we obtain

$$\delta_{j,v} = -\sum_{i \neq j} \boldsymbol{s}_i = \boldsymbol{s}_j.$$

Since $j$ and $v \in \mathcal{C}_j$ were arbitrary, we have shown that $\delta_{i,v} \equiv s_i$ for all $i$ and all $v \in \mathcal{C}_i$. Equivalently, $b_{i,v} = a_{i,v} + s_i$ with $\sum_i s_i = 0$.

Conversely, given any $s_1, \ldots, s_c \in \mathbb{R}^d$ with $\sum_{i=1}^k s_i = 0$, define $b_{i,v} := a_{i,v} + s_i$. Then for every $(v_1, \ldots, v_c) \in \mathcal{C}$,

$$\sum_{i=1}^k b_{i,v_i} = \sum_{i=1}^k a_{i,v_i} + \sum_{i=1}^k s_i = \sum_{i=1}^k a_{i,v_i},$$

so $\{b_{i,v}\}$ also satisfies the factorisation equations. Therefore the set of all solutions is the affine subspace

$$\{a_{i,v}\} + \{(s_1, \ldots, s_c) \in (\mathbb{R}^d)^k : \sum_{i=1}^k s_i = 0\}.$$

$\square$

We illustrate this proposition graphically in Figure 26.

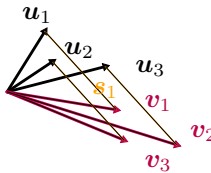

Figure 26: **Illustration of the shift ambiguity in the factorisation equations.**

A neat consequence of this result is that the centered embeddings $u'_t$ are uniquely determined: any factorization we acquire from the embeddings, when centered, will correspond exactly to the true centered factorization.

**Corollary 1** (Uniqueness of the centered factorization). Assume the setting of Proposition 3. For each concept $i$, let

$$\bar{a}_i := \frac{1}{|\mathcal{C}_i|} \sum_{v \in \mathcal{C}_i} a_{i,v}, \qquad \bar{b}_i := \frac{1}{|\mathcal{C}_i|} \sum_{v \in \mathcal{C}_i} b_{i,v},$$

and define the centered factors $a'_{i,v} := a_{i,v} - \bar{a}_i$ and $b'_{i,v} := b_{i,v} - \bar{b}_i$. Then $a'_{i,v} = b'_{i,v}$ for all $i$ and all $v \in \mathcal{C}_i$. Equivalently, for every $(v_1, \ldots, v_c) \in \mathcal{C}$,

$$\sum_{i=1}^k a'_{i,v_i} = \sum_{i=1}^k b'_{i,v_i},$$

so the centered embeddings are uniquely determined by the data. In particular, if $\{u_{i,v}\}$ is the ground-truth factorization and $u'_{i,v} := u_{i,v} - \frac{1}{|\mathcal{C}_i|} \sum_{w \in \mathcal{C}_i} u_{i,w}$, then the centered version of any recovered factorization coincides with $\{u'_{i,v}\}$.

*Proof.* By Proposition 3, there exist $s_1, \ldots, s_c$ with $\sum_i s_i = 0$ such that $b_{i,v} = a_{i,v} + s_i$ for all $i, v$. Averaging over $v \in \mathcal{C}_i$ yields $\bar{b}_i = \bar{a}_i + s_i$. Thus,

$$b'_{i,v} = b_{i,v} - \bar{b}_i = (a_{i,v} + s_i) - (\bar{a}_i + s_i) = a_{i,v} - \bar{a}_i = a'_{i,v},$$

as claimed. Taking $a_{i,v} = u_{i,v}$ gives the final statement. $\square$

First, we consider the general case where the concept values' directions are not necessarily linearly independent. However, suppose the inputs $x_c$ are linearly separable for any $i \in [k], j \in [n]$. In that case, if we can recover all $k \cdot n$ factors, we can reconstruct any $x_c = \sum_{i=1}^k u_{i,c_i}$ as a linear combination of the recovered factors. Due to linear separability, we can then train the linear probes to classify the inputs into the correct concept values.

While such an approach is in principle possible, it is not practical. The reason is that the number of factors to recover is $k \cdot n$, which is exponential in the number of concepts.

To uncover the factors we only need to establish the rank of the design matrix - this then indicates how many datapoints need to be observed to recover the factors. Additionally, this dictates how the samples need to be collected.

**Proposition 4** (Rank of the full–factorial one–hot design)**.** Let $\boldsymbol{X} \in \{0,1\}^{n^k \times cn}$ be the design matrix whose $cn$ columns are $\{x_{j,k} : j = 1, \ldots, k, \ k = 1, \ldots, n\}$, arranged in $k$ blocks of size $n$, with all $n^k$ treatment combinations as rows and each row having exactly one $1$ in each block. Then,

$$\mathrm{rank}(X) = 1 + k(n-1).$$

*Proof.* We show this for the column space of the design matrix $\boldsymbol{X}$. We show that a set of $1 + k(n-1)$ columns span the column space.

Let $\boldsymbol{u} := \boldsymbol{1} \in \mathbb{R}^{n^k}$ and, for each block $j$ and each $k = 2, \ldots, n$, define $\boldsymbol{v}_{j,k} := x_{j,k} - x_{j,1}$. Let

$$\mathcal{B} := \{\boldsymbol{u}\} \ \cup \ \{\boldsymbol{v}_{j,k} : 1 \leq j \leq k, \ 2 \leq k \leq n\}, \quad \text{so} \quad |\mathcal{B}| = 1 + k(n-1).$$

For every block $j$, $\sum_{k=1}^{n} x_{j,k} = u$, hence

$$\sum_{k=2}^{n} \boldsymbol{v}_{j,k} = \boldsymbol{u} - nx_{j,1} \ \Rightarrow \ x_{j,1} = \tfrac{1}{n}\Big(\boldsymbol{u} - \sum_{k=2}^{n} \boldsymbol{v}_{j,k}\Big), \quad x_{j,k} = x_{j,1} + v_{j,k} \ (k \geq 2).$$

Thus every original column $x_{j,k}$ lies in $\mathrm{span}\,\mathcal{B}$, and since $\mathcal{B} \subseteq \mathrm{col}(X)$ we have $\mathrm{col}(X) = \mathrm{span}\,\mathcal{B}$.

Independence of $\mathcal{B}$ can be shown by contradiction. □

Clearly, when the design matrix has full rank $\mathrm{rank}(A) = 1 + k(n-1)$, the linear system $V = AU$ becomes well-determined with a unique solution for the centred per-value vectors $\{\boldsymbol{u}'_v\}$. This ensures that the linear factorization is uniquely identifiable, meaning there is exactly one way to decompose the observed representations into their constituent concept factors. From that, one could recover the full grid of representations over $\mathcal{C}$ and fit linear classifiers on top of them. As long as the original space is linearly separable, a linearly compositional model follows (as defined in Definition 3).

We illustrate some configurations of this in Figure 27 over the case of three concepts with top two rows indicating solvable systems, and the bottom row indicating unsolvable ones due to violating rank constraint.

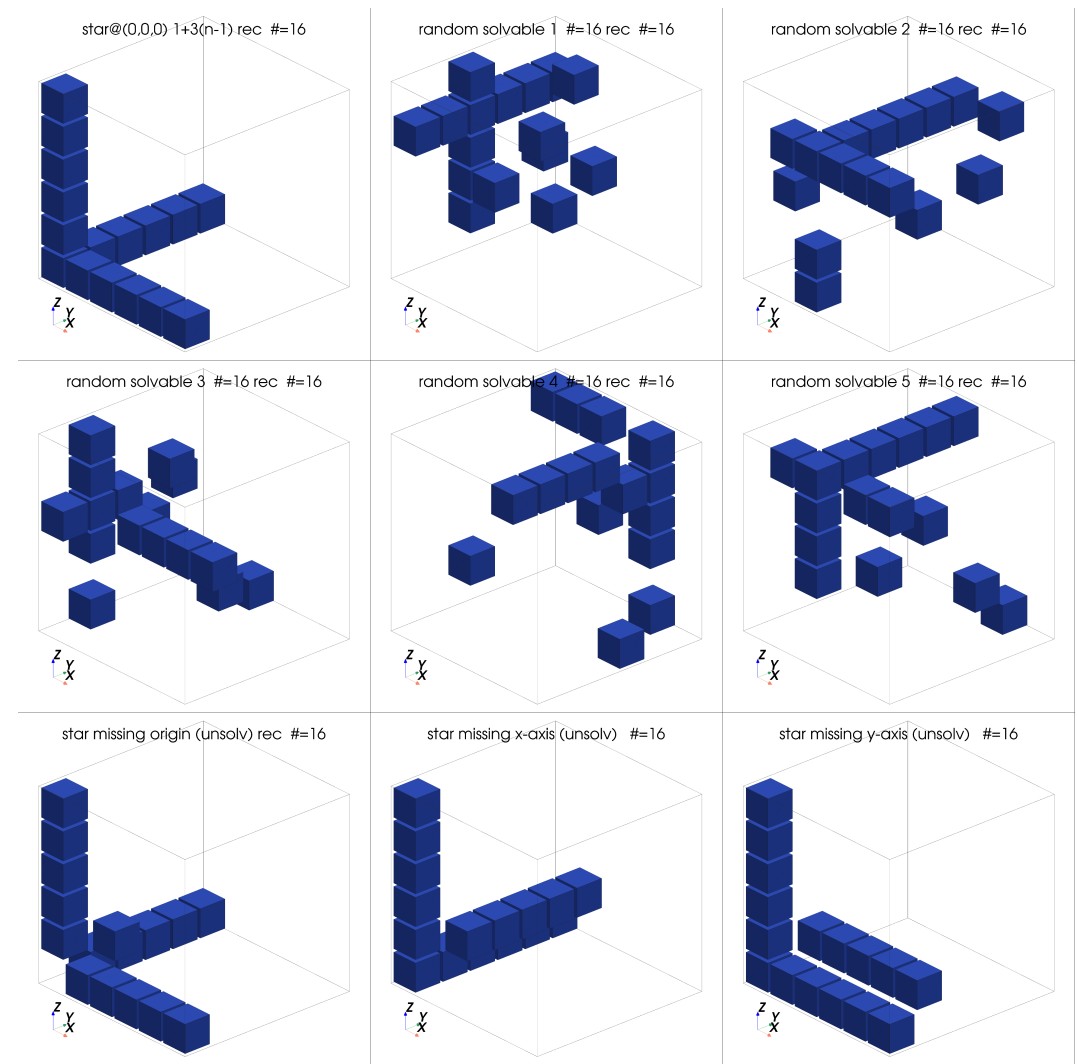

Figure 27: **Examples of one-hot design matrices for recovery of linear factors.** We show sparse grid patterns from the full space $A \in \{0,1\}^{n^k \times \Pi_{i=1}^{k} n_i}$, where each row corresponds to a training tuple and each column to a concept value. The matrices demonstrate how different sampling strategies affect rank and identifiability of the linear factorization. Refer to Definition 5 for the definition of a cross-dataset.

## F  PACKING AND MINIMUM DIMENSION

For a dimension $d \geq 1$. We specify two types of hyperplanes (Ziegler, 1995)

- A central (or *linear*) hyperplane is the zero–set of a non-zero normal vector $w \in \mathbb{R}^d$:

$$H_w = \{x \in \mathbb{R}^d : \langle w, x \rangle = 0\},$$

so it always passes through the origin.

- Allowing an affine bias $b \in \mathbb{R}$ translates the supporting flat:

$$H_{w,b} = \{x \in \mathbb{R}^d : \langle w, x \rangle + b = 0\}.$$

Such hyperplanes need not contain the origin and are sometimes called *offset* or *biased*.

An *arrangement* $\mathcal{H} = \{H_1, \ldots, H_m\}$ is a finite family of hyperplanes. It is said to be in general position when no more than $d$ hyperplanes meet at a single point. This condition prevents degeneracies and maximises the number of connected regions that the arrangement carves out of $\mathbb{R}^d$.

**Theorem 1** (Zaslavsky's region bounds in general position Ziegler (1995)). Let $\mathcal{H}$ be an arrangement of $m$ hyperplanes in $\mathbb{R}^d$ that is in general position. Then, the number of connected regions $R(\mathcal{H})$ is given by:

(a) **Affine (biased) case.** If the hyperplanes may carry arbitrary offsets $b_i$ (so $\mathcal{H}$ is not required to be central), then

$$R(\mathcal{H}) = R_{\text{aff}}(m, d) := \sum_{k=0}^{d} \binom{m}{k}.$$

(b) **Central case.** If every hyperplane passes through the origin,

$$R(\mathcal{H}) = R_{\text{lin}}(m, d) := 2 \sum_{k=0}^{d-1} \binom{m-1}{k}.$$

For $d < k$ one has $R(k, d) = 2^k - \sum_{k=d+1}^{k} \binom{k}{k} < 2^k$, which is the key inequality we will need.

We now exploit Theorem 1 to prove the lower bound on probe dimension; first for the binary case, then for general $n$.

<div>
3 concepts, 2 values per concept, $2D$ space.

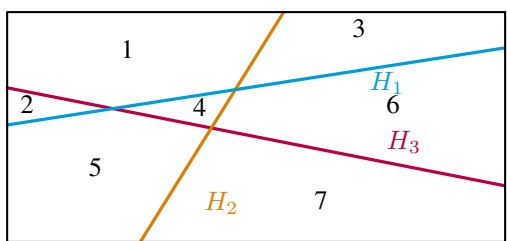

Reduction to two values

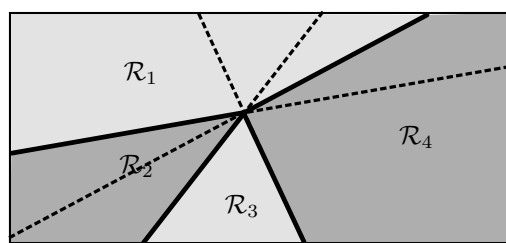
</div>

Figure 28: **Illustration of the probe dimension lower bound.** Schematic showing the arrangement of probe hyperplanes and the resulting partitioning of the embedding space.

**Proposition 5** (Minimum dimension for linear probes). Fix integers $k \geq 1$ (number of concepts) and $n \geq 2$ (values per concept). Suppose:

(a) The feature extractor is $f : \mathcal{X} \to \mathbb{R}^d$, $\boldsymbol{z} := f(x)$

(b) For each $(i, j) \in [k] \times [n]$ there exists a probe $(\boldsymbol{p}_{i,j}, b_{i,j})$ with $\boldsymbol{p}_{i,j} \in \mathbb{R}^d$ and $b_{i,j} \in \mathbb{R}$ used to compute the logit

$$s_{i,j}(\boldsymbol{z}) := \langle \boldsymbol{p}_{i,j}, \boldsymbol{z} \rangle + b_{i,j}, \tag{8}$$

and there are label functions $v_1, \ldots, v_c : \mathcal{X} \to [n]$ such that for every $\boldsymbol{x} \in \mathcal{X}$,

$$\arg\max_{j \in [n]} s_{i,j}(f(\boldsymbol{x})) = v_i(\boldsymbol{x}), \quad \forall i \in [k]. \tag{9}$$

Assume also that every label combination occurs: for every $\boldsymbol{v} = (v_1, \ldots, v_c) \in [n]^k$, there exists $x_{\boldsymbol{v}} \in \mathcal{X}$ such that $v_i(x_{\boldsymbol{v}}) = v_i$ for all $i$. Then necessarily

$$d \ \geq \ k, \tag{10}$$

and this bound is tight: one can construct probe and representation families that achieve perfect prediction in dimension $d = k$.

*Proof sketch.* Reduce to the binary case by fixing two values per concept and restricting to the resulting $2^c$ combinations. Each concept induces one affine separating hyperplane. To realize all binary labelings, the arrangement must carve at least $2^c$ regions. By Theorem 1, when $d < c$ we have $\sum_{k=0}^{d} \binom{c}{k} < 2^c$, so $2^c$ regions are impossible. Hence $d \geq c$. Tightness follows by a $d = c$ construction (coordinates per concept with suitable affine offsets). See Figure 6.

*Proof. Binary case ($n = 2$).* We can take one affine binary classifier per concept:

$$h_i(\boldsymbol{z}) := s_{i,1}(\boldsymbol{z}) - s_{i,2}(\boldsymbol{z}) = \langle \boldsymbol{p}_{i,1} - \boldsymbol{p}_{i,2}, \boldsymbol{z} \rangle + (b_{i,1} - b_{i,2}). \tag{11}$$

By letting $\boldsymbol{w}_i := \boldsymbol{p}_{i,1} - \boldsymbol{p}_{i,2}$, $b_i := b_{i,1} - b_{i,2}$. Each $h_i$ defines an affine hyperplane

$$H_i := \{\boldsymbol{z} \in \mathbb{R}^d \mid \langle \boldsymbol{w}_i, \boldsymbol{z} \rangle + b_i = 0\}. \tag{12}$$

Since all $2^k$ binary label configurations occur, the $k$ affine hyperplanes $H_1, \ldots, H_c$ must jointly separate $\mathbb{R}^d$ into at least $2^k$ distinct regions.

But the number of regions formed by $k$ affine hyperplanes in $\mathbb{R}^d$ is at most

$$\sum_{k=0}^{d} \binom{k}{k} < 2^k \quad \text{whenever } d < k \quad \text{(by Theorem 1)}. \tag{13}$$

Thus, we must have $d \geq k$.

Construction is simple: assume parallel planes in their own dimensions. Let $d = k$, and embed

$$f(x_{\boldsymbol{v}}) := (v_1, \ldots, v_c) \in \mathbb{R}^k. \tag{14}$$

We define probe vectors as

$$\boldsymbol{p}_{i,j} := \boldsymbol{e}_i \quad \text{and} \quad b_{i,j} := -j. \tag{15}$$

Then

$$s_{i,j}(f(x_{\boldsymbol{v}})) = \langle \boldsymbol{e}_i, \boldsymbol{v} \rangle - j = v_i - j. \tag{16}$$

Thus, the correct label is recovered for all $i$, and $d = k$ suffices.

In general for $n > 2$, we can repeat the same computation for colinear weights per concepts and values. This reduces the general $n$ case to the binary case above, and the same lower bound $d \geq k$ follows. $\qquad\square$

## G  PROOFS

We write $\mathcal{D}$ for the full dataset of all $n^k$ combinations and $\mathcal{D}^{\boldsymbol{c}}$ for a cross-dataset as in Definition 5. Any learned quantity carries a superscript indicating the training set, e.g., $\{\boldsymbol{w}_{i,j}^{(\mathcal{D})}\}$ or $\{\boldsymbol{w}_{i,j}^{(\mathcal{D}^{\boldsymbol{c}})}\}$ with logits $\ell_{i,j}^{(S)}(\boldsymbol{x}) := (\boldsymbol{w}_{i,j}^{(S)})^{\top}\boldsymbol{x}$ and probabilities $p_{i,j}^{(S)}(\boldsymbol{x}) := \exp(\ell_{i,j}^{(S)}(\boldsymbol{x}))/\sum_k \exp(\ell_{i,k}^{(S)}(\boldsymbol{x}))$ for a training set $S$.

**Definition 6** (Dataset index set and marginal counts). For any dataset $S \subseteq \{(\boldsymbol{x}_{\boldsymbol{c}'}) : \boldsymbol{c}' \in [n]^k\}$ (e.g., $S = \mathcal{D}$ or $S = \mathcal{D}^{\boldsymbol{c}}$), define the index set $I(S) := \{\boldsymbol{c}' : (\boldsymbol{x}_{\boldsymbol{c}'}) \in S\}$. For concept $i \in [k]$ and value $j \in [n]$, the marginal count of value $j$ in $S$ is

$$N_{i,j}(S) := \big|\{\, \boldsymbol{c}' \in I(S) : k_i' = j \,\}\big|.$$

When $S$ is clear, we abbreviate $N_{i,j} := N_{i,j}(S)$.

**Remark 1** (Marginal counts: full vs cross-datasets). For the full dataset $\mathcal{D}$, the marginal counts are balanced:

$$N_{i,j}(\mathcal{D}) = n^{k-1} \quad \text{for all } i \in [k],\ j \in [n].$$

For a cross-dataset $\mathcal{D}^{\boldsymbol{c}}$ as in Definition 5, the marginal counts satisfy

$$N_{i,c_i}(\mathcal{D}^{\boldsymbol{c}}) = 1 + (k-1)(n-1), \qquad N_{i,j}(\mathcal{D}^{\boldsymbol{c}}) = 1 \text{ for all } j \neq c_i.$$

*Proof.* In $\mathcal{D}$ fixing $v_i = j$ leaves $n^{k-1}$ free coordinates. In $\mathcal{D}^{\boldsymbol{c}}$: varying concept $i$ contributes one point for each $j \neq c_i$; the center contributes one more with $v_i = c_i$; varying any other concept $k \neq i$ adds $(n-1)$ points with $v_i = c_i$, across $(k-1)$ such concepts, totaling $(k-1)(n-1)$. $\square$

**Definition 7** (Intervention on a concept value). For any concept index $i \in [k]$, target value $j \in [n]$, and concept vector $\boldsymbol{c} \in [n]^k$, define the intervened index and representation

$$\boldsymbol{c}(i \to j) := (c_1, \ldots, c_{i-1}, j, c_{i+1}, \ldots, c_c), \quad \boldsymbol{x}_{\boldsymbol{c}(i \to j)} := \boldsymbol{x}_{\boldsymbol{c}} \text{ with concept } i \text{ set to } j.$$

We also write $\boldsymbol{c}^{(i \to j)}$ as an alias for $\boldsymbol{c}(i \to j)$ when convenient. Multiple interventions compose componentwise.

**Definition 8** (Binary complement notation). In the binary case ($\mathcal{C}_i = \{0, 1\}$), we write $\bar{c}_i := 1 - c_i$ for the complement value of concept $i$. As shorthand for an intervention to the complement, we write $\boldsymbol{c}^{(\bar{c}_i)} := \boldsymbol{c}^{(i \leftarrow \bar{c}_i)}$.

**Definition 9** (Per-concept differences). For each concept $i \in [k]$, fix a reference class $r_i \in [n]$ and define the per-concept difference parameters

$$\tilde{\boldsymbol{w}}_{i,j} := \boldsymbol{w}_{i,j} - \boldsymbol{w}_{i,r_i}, \qquad \tilde{b}_{i,j} := b_{i,j} - b_{i,r_i}.$$

Softmax probabilities for concept $i$ are invariant under adding a constant vector and bias shared across classes. Thus only differences $\Delta \boldsymbol{w}_{i,j\ell} := \boldsymbol{w}_{i,j} - \boldsymbol{w}_{i,\ell}$ and $\Delta b_{i,j\ell} := b_{i,j} - b_{i,\ell}$ are identifiable; $\tilde{\boldsymbol{w}}$ and $\tilde{b}$ provide a concrete representative.

For making use of the stability condition we note the degree of freedom in (arg/soft)max.

**Lemma 1** (Equal probabilities imply equal weights up to a shift per concept). For any concept index $i$, and for each class $j \in [n]$, let $\boldsymbol{f}_{i,j} \in \mathbb{R}^d$ and $\boldsymbol{f}_{i,j}' \in \mathbb{R}^d$. Assume that for every input $\boldsymbol{x} \in \mathbb{R}^d$,

$$\frac{\exp(\boldsymbol{f}_{i,j} \cdot \boldsymbol{x})}{\sum_{k=1}^n \exp(\boldsymbol{f}_{i,k} \cdot \boldsymbol{x})} = \frac{\exp(\boldsymbol{f}_{i,j}' \cdot \boldsymbol{x})}{\sum_{k=1}^n \exp(\boldsymbol{f}_{i,k}' \cdot \boldsymbol{x})} \quad \text{for all } j \in [n].$$

Then there exists a vector $\boldsymbol{u}_i \in \mathbb{R}^d$ (independent of $j$) such that

$$\boldsymbol{f}_{i,j} = \boldsymbol{f}_{i,j}' + \boldsymbol{u}_i \quad \text{for all } j \in [n].$$

*Proof.* Fix $i$ and an arbitrary $\boldsymbol{x} \in \mathbb{R}^d$. Define

$$Z_i(\boldsymbol{x}) = \log\Big(\sum_{k=1}^n e^{\boldsymbol{f}_{i,k} \cdot \boldsymbol{x}}\Big), \qquad Z_i'(\boldsymbol{x}) = \log\Big(\sum_{k=1}^n e^{\boldsymbol{f}_{i,k}' \cdot \boldsymbol{x}}\Big).$$

Let

$$p_{i,j}(\boldsymbol{x}) = \frac{e^{\boldsymbol{f}_{i,j}\cdot\boldsymbol{x}}}{\sum_k e^{\boldsymbol{f}_{i,k}\cdot\boldsymbol{x}}}, \qquad p'_{i,j}(\boldsymbol{x}) = \frac{e^{\boldsymbol{f}'_{i,j}\cdot\boldsymbol{x}}}{\sum_k e^{\boldsymbol{f}'_{i,k}\cdot\boldsymbol{x}}}.$$

By assumption $p_{i,j}(\boldsymbol{x}) = p'_{i,j}(\boldsymbol{x})$ for all $j$. Taking logs gives

$$\log p_{i,j}(\boldsymbol{x}) = \log p'_{i,j}(\boldsymbol{x}) \implies \boldsymbol{f}_{i,j}\cdot\boldsymbol{x} - Z_i(\boldsymbol{x}) = \boldsymbol{f}'_{i,j}\cdot\boldsymbol{x} - Z'_i(\boldsymbol{x}) \quad \forall j.$$

Thus for this $\boldsymbol{x}$ there exists a scalar $b_i(\boldsymbol{x}) := Z_i(\boldsymbol{x}) - Z'_i(\boldsymbol{x})$ with

$$\boldsymbol{f}_{i,j}\cdot\boldsymbol{x} = \boldsymbol{f}'_{i,j}\cdot\boldsymbol{x} + b_i(\boldsymbol{x}) \quad \forall j.$$

For classes $j$ and $\ell$, by subtracting, gives:

$$(\boldsymbol{f}_{i,j} - \boldsymbol{f}_{i,\ell})\cdot\boldsymbol{x} = (\boldsymbol{f}'_{i,j} - \boldsymbol{f}'_{i,\ell})\cdot\boldsymbol{x} \quad \forall\boldsymbol{x} \in \mathbb{R}^d.$$

Since this defines a hyperplane on which all $\boldsymbol{x}$ need to lie, the weight differences need to be equal:

$$\boldsymbol{f}_{i,j} - \boldsymbol{f}_{i,\ell} = \boldsymbol{f}'_{i,j} - \boldsymbol{f}'_{i,\ell} \quad \forall j, \ell.$$

Fixing any reference class $\ell$ and setting $\boldsymbol{u}_i := \boldsymbol{f}_{i,\ell} - \boldsymbol{f}'_{i,\ell}$, yields:

$$\boldsymbol{f}_{i,j} = \boldsymbol{f}'_{i,j} + \boldsymbol{u}_i.$$

$\square$

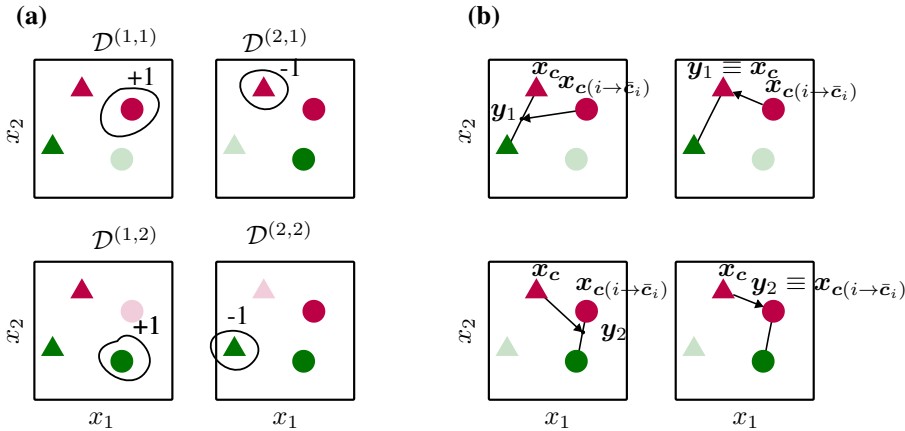

Figure 29: **Illustration of the invariance lemma (left) and the main proposition (right). (a)** The invariance lemma: we can always find a dataset for which a single point is a support vector, leading to invariance. **(b)** The main proposition: any point is projected onto the other class' convex hull by a single concept value flip.

**Lemma 2** (Bi-directional tight support vectors in binary concepts). For binary concepts $\mathcal{C}_i = \{0, 1\}$, consider any cross-dataset $\mathcal{D}^{\boldsymbol{c}}$ and the corresponding SVM solution $\{\boldsymbol{w}_{i,j}^{(\mathcal{D}^{\boldsymbol{c}})}, b_{i,j}^{(\mathcal{D}^{\boldsymbol{c}})}\}$. Because $N_{i,0}(\mathcal{D}^{\boldsymbol{c}}) = N_{i,1}(\mathcal{D}^{\boldsymbol{c}}) = 1$, there exist support vectors $\boldsymbol{x}_{\boldsymbol{c}^0}, \boldsymbol{x}_{\boldsymbol{c}^1} \in \mathcal{D}^{\boldsymbol{c}}$ with $v_i^0 = 0$ and $v_i^1 = 1$ such that both are tight with respect to their class boundaries:

$$(\boldsymbol{w}_{i,0}^{(\mathcal{D}^{\boldsymbol{c}})})^\top \boldsymbol{x}_{\boldsymbol{c}^0} + b_{i,0}^{(\mathcal{D}^{\boldsymbol{c}})} = (\boldsymbol{w}_{i,1}^{(\mathcal{D}^{\boldsymbol{c}})})^\top \boldsymbol{x}_{\boldsymbol{c}^0} + b_{i,1}^{(\mathcal{D}^{\boldsymbol{c}})} + 1 \tag{17}$$

$$(\boldsymbol{w}_{i,1}^{(\mathcal{D}^{\boldsymbol{c}})})^\top \boldsymbol{x}_{\boldsymbol{c}^1} + b_{i,1}^{(\mathcal{D}^{\boldsymbol{c}})} = (\boldsymbol{w}_{i,0}^{(\mathcal{D}^{\boldsymbol{c}})})^\top \boldsymbol{x}_{\boldsymbol{c}^1} + b_{i,0}^{(\mathcal{D}^{\boldsymbol{c}})} + 1 \tag{18}$$

*Proof.* This follows from standard hard-margin SVM theory: each class has at least one support vector achieving equality at the margin (Cortes & Vapnik, 1995). $\square$

**Lemma 3** (Invariance to irrelevant concepts, binary case). Assume each concept is binary, $\mathcal{C}_i = \{0, 1\}$ for all $i \in [k]$, and write $\bar{v} := 1 - v$. For any $i \in [k]$ and any $\boldsymbol{c}, \boldsymbol{c}' \in [2]^k$ with $c_i = c'_i =: v$,

$$P(C_i = v \mid \boldsymbol{x}_{\boldsymbol{c}}) = P(C_i = v \mid \boldsymbol{x}_{\boldsymbol{c}'}). \tag{19}$$

*Proof.* We encode the $i$-label by $y_i(\boldsymbol{x}) \in \{+1, -1\}$ with $y_i(\boldsymbol{x}) = +1$ iff $C_i(\boldsymbol{x}) = 1$ and $-1$ otherwise. Let

$$g_i(\boldsymbol{x}) := (\boldsymbol{w}_{i,1} - \boldsymbol{w}_{i,0})^\top \boldsymbol{x} + (b_{i,1} - b_{i,0}) \tag{20}$$

By Lemma 1, the pair $(\Delta \boldsymbol{w}_i, \Delta b_i) := (\boldsymbol{w}_{i,1} - \boldsymbol{w}_{i,0}, b_{i,1} - b_{i,0})$ is the same no matter which cross-dataset we train on.

Let $\mathcal{I} = [2]^{k-1}$ be assignments of all concepts except $i$. For each $\boldsymbol{u} \in \mathcal{I}$ there are two cross-datasets: $\mathcal{D}^{(\boldsymbol{u},0)}$ and $\mathcal{D}^{(\boldsymbol{u},1)}$. In the binary hard-margin setting, each such training has exactly one minority (support) example w.r.t. concept $i$, and for that example the signed margin is tight:

$$y_i(\boldsymbol{x}) \, g_i(\boldsymbol{x}) = 1 \qquad \text{(for the unique support example of that training).} \tag{21}$$

- In $\mathcal{D}^{(\boldsymbol{u},0)}$, the unique minority is $\boldsymbol{x}_{\boldsymbol{u},1}$, so $y_i(\boldsymbol{x}_{\boldsymbol{u},1}) = +1$ and tightness gives

$$g_i(\boldsymbol{x}_{\boldsymbol{u},1}) = +1. \quad (A_{\boldsymbol{u}}) \tag{22}$$

- In $\mathcal{D}^{(\boldsymbol{u},1)}$, the unique minority is $\boldsymbol{x}_{\boldsymbol{u},0}$, so $y_i(\boldsymbol{x}_{\boldsymbol{u},0}) = -1$ and tightness gives

$$g_i(\boldsymbol{x}_{\boldsymbol{u},0}) = -1. \quad (B_{\boldsymbol{u}}) \tag{23}$$

The *same* $g_i$ (same $\Delta \boldsymbol{w}_i, \Delta b_i$) appears in $(A_{\boldsymbol{u}})$ and $(B_{\boldsymbol{u}})$ for every $\boldsymbol{u}$, by Desideratum 3.

As $\boldsymbol{u}$ ranges over $\mathcal{I}$, the equations $(A_{\boldsymbol{u}})$ cover *every* point with $C_i = 1$, and the equations $(B_{\boldsymbol{u}})$ cover *every* point with $C_i = 0$. Therefore

$$g_i(\boldsymbol{x}) = \begin{cases} +1, & \text{if } C_i(\boldsymbol{x}) = 1, \\ -1, & \text{if } C_i(\boldsymbol{x}) = 0, \end{cases} \quad \text{on the whole grid } \{\boldsymbol{x}_{\boldsymbol{c}} : \boldsymbol{c} \in [2]^k\}.$$

Hence $g_i(\boldsymbol{x})$ depends only on $C_i(\boldsymbol{x})$ and not on the other concepts. Since in the binary model $P(C_i = 1 \mid \boldsymbol{x}) = \sigma(g_i(\boldsymbol{x})) = \dfrac{1}{1 + e^{-g_i(\boldsymbol{x})}}$ (and $P(C_i = 0 \mid \boldsymbol{x}) = 1 - P(C_i = 1 \mid \boldsymbol{x})$), the conditional probability $P(C_i = v \mid \boldsymbol{x}_{\boldsymbol{c}})$ is constant over all $\boldsymbol{c}$ with $c_i = v$. In particular, for any $\boldsymbol{c}, \boldsymbol{c}'$ with $c_i = c_i'$,

$$P(C_i = c_i \mid \boldsymbol{x}_{\boldsymbol{c}}) = P(C_i = c_i \mid \boldsymbol{x}_{\boldsymbol{c}'}).$$

$\square$

Next, we establish an important property of SVMs on two separable sets, one of which is a singleton.

**Lemma 4** (SVM geometry for separable sets). Given a set of points $\mathcal{Y} := \{\boldsymbol{y}_i\}_i^N (\boldsymbol{y}_i \in \mathbb{R}^d)$ and a point $\boldsymbol{x} \in \mathbb{R}^d$ with an optimal linearly separable hyperplane $\mathcal{H}_{\boldsymbol{w},b} = \{\boldsymbol{x} \mid \boldsymbol{w}^\top \boldsymbol{x} + b = 0\}$ under SVM, the following hold:

1. The weight vector $\boldsymbol{w}$ separates convex combinations such that they are support vectors, that is, for some $\{\lambda_i\}_{i=1}^N$ it holds:

$$\boldsymbol{w}^\top \left( \sum_i \lambda_i \boldsymbol{y}_i \right) + b = -1 \quad \text{for } \lambda_i \geq 0, \sum_i \lambda_i = 1 \tag{24}$$

$$\boldsymbol{w}^\top \boldsymbol{x} + b = +1 \tag{25}$$

2. The weight vector $\boldsymbol{w}$ equals the shortest distance between the sets:

$$\frac{2}{||\boldsymbol{w}||^2} \boldsymbol{w} = \left( \boldsymbol{x} - \sum_i \lambda_i \boldsymbol{y}_i \right) \tag{26}$$

*Proof.* These conditions are implied by a standard fact in SVMs: the weight vector $\boldsymbol{w}$ is parallel to the shortest line connecting the two sets (Bennett & Bredensteiner, 2000). By noting that $\alpha \boldsymbol{w} = (\boldsymbol{x} - \sum_i \lambda_i \boldsymbol{y}_i)$, we can derive the proportionality constant as $\alpha = \frac{2}{||\boldsymbol{w}||^2}$. $\square$

We now establish the main result of the resulting geometry of linearly generalizable compositional models.

**Proposition 1** (Binary case: compositional generalization implies linear factorization). Let $\Pi = (f, \mathcal{H}, A, \mathcal{T})$ be the tuple instantiated in Section 3.4, with linear heads $\mathcal{H}$ and $A$ given by GD+CE. Suppose that the training sets follow random sampling with validity rule $R(T) = 1$ if $|T| = 2^{k-1} + 1$. Assume Desiderata 1–3 are satisfied. Then under the binary grid $\mathcal{C}_i = \{0, 1\}$ with $\mathcal{X} = \{\boldsymbol{x_c} : \boldsymbol{c} \in [2]^k\} \subset \mathbb{R}^d$, there exist $\{\boldsymbol{u}_{i,0}, \boldsymbol{u}_{i,1} \in \mathbb{R}^d\}_{i=1}^k$ such that for every $\boldsymbol{c} \in [2]^k$ the following holds:

1. *(Linearity)* $\boldsymbol{x_c} = \sum_{i=1}^k \boldsymbol{u}_{i,c_i}$.

2. *(Cross-concept orthogonality)* $(\boldsymbol{u}_{i,1} - \boldsymbol{u}_{i,0}) \perp (\boldsymbol{u}_{j,1} - \boldsymbol{u}_{j,0})$ for all $i, j \in [k]$ with $(i \neq j)$.

*Proof.* First, note that the fact that any training set $T \in \mathcal{T}$ has $2^{n-1} + 1$ points implies that for any concept and its value, we can always choose a dataset which has only a single point over that concept's value. Because of this, the proof reduces to the case of working with a "cross-like" datasets. We thus work within this simplified setting to avoid technical clutter, but the key idea remains the same.

**Linearity**.

The idea is to show that for a pair of cross-datasets that share the datapoints in negative class, the shortest distance from a single point in the positive class to the convex set of the positive points is achieved by considering a flip in one of the concepts. We make this concrete below.

Consider any datapoint $\boldsymbol{x_c}$ and its corresponding cross dataset centered at this point $\mathcal{D}^{(\boldsymbol{c})}$. Additionally, for any concept $i \in [k]$ consider a "counterfactual" datapoint $\boldsymbol{x}_{\boldsymbol{c}(i \to \bar{c}_i)}$ that flips the value of concept $i$ to $\bar{c}_i$, and consider its corresponding cross-dataset $\mathcal{D}^{(\boldsymbol{c}(i \to \bar{c}_i))}$.

Note that for the concept $i$ it holds that:

1. Under $\mathcal{D}_{\boldsymbol{c}} = \{\boldsymbol{x_c}\} \cup \{\boldsymbol{x}_{\boldsymbol{c}(i \to \bar{c}_i)} : i \in [k]\}$. For each concept $i$, the marginal counts are

$$N_{i,c_i}(\mathcal{D}^{\boldsymbol{c}}) = k, \qquad N_{i,\bar{c}_i}(\mathcal{D}_{\boldsymbol{c}}) = 1 \tag{27}$$

(by Remark 1). Thus $\boldsymbol{x}_{\boldsymbol{c}(i \to \bar{c}_i)}$ is the unique minority example for concept $i$ (label $\bar{c}_i$), and

$$\mathcal{Y}_1 := \mathcal{D}^{\boldsymbol{c}} \setminus \{\boldsymbol{x}_{\boldsymbol{c}(i \to \bar{c}_i)}\} \tag{28}$$

is the set of $k$ majority examples (label $c_i$).

2. Note $\mathcal{D}^{\boldsymbol{c}(i \to \bar{c}_i)} := \{\boldsymbol{x}_{\boldsymbol{c}(i \to \bar{c}_i)}\} \cup \{\boldsymbol{x}_{\boldsymbol{c}(k \to \bar{c}_k)} : k \in [k]\}$.

   For $k \neq i$ the counts are unchanged: $N_{k,c_k}(\mathcal{D}^{\boldsymbol{c}(i \to \bar{c}_i)}) = k$ and $N_{k,\bar{c}_k}(\mathcal{D}^{\boldsymbol{c}(i \to \bar{c}_i)}) = 1$, but for concept $i$ they swap: $N_{i,\bar{c}_i}(\mathcal{D}^{\boldsymbol{c}(i \to \bar{c}_i)}) = k$ and $N_{i,c_i}(\mathcal{D}^{\boldsymbol{c}(i \to \bar{c}_i)}) = 1$. Thus $\boldsymbol{x_c}$ is now the unique minority example for concept $i$ (label $c_i$). Let $\mathcal{Y}_2 = \mathcal{D}^{\boldsymbol{c}(i \to \bar{c}_i)} \setminus \{\boldsymbol{x_c}\}$ be the majority examples for concept $i$.

Let the majority support vectors for $\mathcal{D}^{\boldsymbol{c}}$ and $\mathcal{D}^{\boldsymbol{c}(i \to \bar{c}_i)}$ be $\boldsymbol{y}_1$ and $\boldsymbol{y}_2$ respectively. By Lemma 4, we can write

$$\boldsymbol{y}_1 = \lambda_i \boldsymbol{x_c} + \sum_{j \in [k] \setminus \{i\}} \lambda_j \boldsymbol{x}_{\boldsymbol{c}(j \to \bar{c}_j)} \quad \text{and} \quad \boldsymbol{y}_2 = \gamma_i \boldsymbol{x}_{\boldsymbol{c}(i \to \bar{c}_i)} + \sum_{j \in [k] \setminus \{i\}} \gamma_j \boldsymbol{x}_{\boldsymbol{c}(j \to \bar{c}_j)} \tag{29}$$

for some convex combinations $\lambda_j \geq 0$ with $\sum_i^k \lambda_j = 1$ and $\gamma_j \geq 0$ with $\sum_i^k \gamma_j = 1$.

Additionally, note that by Lemma 3 it holds that for any point $\boldsymbol{x}_{\boldsymbol{c}'}$ it holds that

$$\boldsymbol{w}_j^\top \boldsymbol{x}_{\boldsymbol{c}'} + b_j = y_i(\boldsymbol{c}'), \tag{30}$$

where we use a shorthand $y_i(\boldsymbol{c}') = 1$ if $j = c_i$ and $y_i(\boldsymbol{c}') = -1$ otherwise.

Then, by Lemma 4 it holds that the support vectors are aligned with the shortest segment between the convex sets (pairs of $\boldsymbol{x}_{\boldsymbol{c}(i \to \bar{c}_i)}$ and $\boldsymbol{y}_1$, and $\boldsymbol{x_c}$ and $\boldsymbol{y}_2$)

$$\boldsymbol{x}_{\boldsymbol{c}(i \to \bar{c}_i)} + y_i(\boldsymbol{c}) \frac{2}{||\boldsymbol{w}_i||^2} \boldsymbol{w}_i = \boldsymbol{y}_1 \quad \text{and} \quad \boldsymbol{x_c} - y_i(\boldsymbol{c}) \frac{2}{||\boldsymbol{w}_i||^2} \boldsymbol{w}_i = \boldsymbol{y}_2, \tag{31}$$

where clearly $y_i(\boldsymbol{c}(i \to \bar{c}_i)) = -y_i(\boldsymbol{c})$. From this, it follows that

$$\boldsymbol{y}_1 - \boldsymbol{x}_{\boldsymbol{c}(i \to \bar{c}_i)} = \boldsymbol{x}_{\boldsymbol{c}} - \boldsymbol{y}_2. \tag{32}$$

Now, for any $k \neq i$, evaluate:

$$
\begin{aligned}
\boldsymbol{w}_k^\top \boldsymbol{y}_1 + b_k &= \boldsymbol{w}_k^\top \left( \lambda_i \boldsymbol{x}_{\boldsymbol{c}} + \sum_{j \in [k] \setminus \{i\}} \lambda_j \boldsymbol{x}_{\boldsymbol{c}(j \to \bar{c}_j)} \right) + b_k \\
&= \lambda_i \boldsymbol{w}_k^\top \boldsymbol{x}_{\boldsymbol{c}} + \sum_{j \in [k] \setminus \{i\}} \lambda_j \boldsymbol{w}_k^\top \boldsymbol{x}_{\boldsymbol{c}(j \to \bar{c}_j)} + \sum_i^k \lambda_i b_k \\
&= \lambda_i (\boldsymbol{w}_k^\top \boldsymbol{x}_{\boldsymbol{c}} + b_k) + \sum_{j \in [k] \setminus \{i\}} \lambda_j (\boldsymbol{w}_k^\top \boldsymbol{x}_{\boldsymbol{c}(j \to \bar{c}_j)} + b_k) \\
&= \lambda_i y_k(\boldsymbol{c}) + \sum_{j \in [k] \setminus \{i,k\}} \lambda_j y_k(\boldsymbol{c}(j \to \bar{c}_j)) + \lambda_k y_k(\boldsymbol{c}(k \to \bar{c}_k)) \\
&= \lambda_i y_k(\boldsymbol{c}) + \left( \sum_{j \in [k] \setminus \{i,k\}} \lambda_j \right) y_k(\boldsymbol{c}) - \lambda_k y_k(\boldsymbol{c}) \\
&= (1 - \lambda_k) y_k(\boldsymbol{c}) - \lambda_k y_k(\boldsymbol{c}) = (1 - 2\lambda_k) y_k(\boldsymbol{c}),
\end{aligned}
\tag{33}
$$

where we used the fact that $\lambda$ are convex combinations in the second equality, and the fact that in the paired dataset $k$-concept values remain the same when flipping any other concept than $k$.

By repeating the same calculation as (33) for $\boldsymbol{y}_2$, we get:

$$\boldsymbol{w}_k^\top \boldsymbol{y}_2 + b_k = (1 - 2\gamma_k) y_k(\boldsymbol{c}). \tag{34}$$

By (32) it follows that

$$
\begin{aligned}
& \boldsymbol{w}_k^\top (\boldsymbol{y}_1 - \boldsymbol{x}_{\boldsymbol{c}(i \to \bar{c}_i)}) = \boldsymbol{w}_k^\top (\boldsymbol{x}_{\boldsymbol{c}} - \boldsymbol{y}_2) \\
\Rightarrow \quad & \boldsymbol{w}_k^\top \boldsymbol{y}_1 + b_k - \boldsymbol{w}_k^\top \boldsymbol{x}_{\boldsymbol{c}(i \to \bar{c}_i)} - b_k = \boldsymbol{w}_k^\top \boldsymbol{x}_{\boldsymbol{c}} + b_k - \boldsymbol{w}_k^\top \boldsymbol{y}_2 - b_k \\
\Rightarrow \quad & (1 - 2\lambda_k) y_k(\boldsymbol{c}) - y_k(\boldsymbol{c}) = y_k(\boldsymbol{c}) - (1 - 2\gamma_k) y_k(\boldsymbol{c}) \\
\Rightarrow \quad & 1 - 2\lambda_k - 1 = 1 - 1 + 2\gamma_k \\
\Rightarrow \quad & \lambda_k + \gamma_k = 0.
\end{aligned}
\tag{35}
$$

Clearly, since $\lambda_k$ and $\gamma_k$ are convex combinations and thus non-negative, (35) implies that $\lambda_k = \gamma_k = 0$.

By repeating this process for all $k \neq i$, we get that $\lambda_k = \gamma_k = 0$ for all $k \neq i$, and therefore $\lambda_i = \gamma_i = 1$. From this, it follows that $\boldsymbol{y}_1 = \boldsymbol{x}_{\boldsymbol{c}}$ and $\boldsymbol{y}_2 = \boldsymbol{x}_{\boldsymbol{c}(i \to \bar{c}_i)}$. This means that

$$\boldsymbol{x}_{\boldsymbol{c}(i \to \bar{c}_i)} + y_i(\boldsymbol{c}) \frac{2}{||\boldsymbol{w}_i||^2} \boldsymbol{w}_i = \boldsymbol{x}_{\boldsymbol{c}} \quad \text{and} \quad \boldsymbol{x}_{\boldsymbol{c}} - y_i(\boldsymbol{c}) \frac{2}{||\boldsymbol{w}_i||^2} \boldsymbol{w}_i = \boldsymbol{x}_{\boldsymbol{c}(i \to \bar{c}_i)}, \tag{36}$$

and therefore the differences between $\boldsymbol{x}_{\boldsymbol{c}} - \boldsymbol{x}_{\boldsymbol{c}(i \to \bar{c}_i)}$ are independent of other concept variations. Because of that, we can write any datapoint $\boldsymbol{x}_{\boldsymbol{c}}$ as a sum of concept-specific values $\boldsymbol{u}_{i,c_i}(c_i \in [2])$. For insstance, if we fix $\boldsymbol{c}_0 = (0, \ldots, 0) \in [2]^k$, and let $\boldsymbol{c}_k = (0, \ldots, 0, 1, 0, \ldots, 0) \in [2]^k$ be a vector with 1 in the $k$-th position, we can express $\boldsymbol{x}_{\boldsymbol{c}}$ as, for example (up to a global linear shift per concept)

$$
\begin{aligned}
\boldsymbol{u}_{i,0} &= \boldsymbol{x}_{\boldsymbol{c}_0}/k, \quad \boldsymbol{u}_{i,1} = \boldsymbol{x}_{\boldsymbol{c}_0}/k + \frac{2}{||\boldsymbol{w}_i||^2} \boldsymbol{w}_i, \\
\boldsymbol{x}_{\boldsymbol{c}} &= \sum_{i=1}^k \boldsymbol{u}_{i,c_i},
\end{aligned}
\tag{37}
$$

which establishes linearity.

**Orthogonality.** First, note that by invariance (Lemma 3) it holds that for any concept $i$, changes in concept values other than $i$ do not affect the prediction of concept $i$. Therefore, it holds that for any concept $j \neq i$, it holds that

$$\boldsymbol{w}_i^\top \boldsymbol{x_c} + b_i = \boldsymbol{w}_i^\top \boldsymbol{x}_{\boldsymbol{c}(j \to \bar{c}_j)} + b_i \tag{38}$$

But by linear factorization (37) it follows that

$$
\begin{aligned}
& \boldsymbol{w}_i^\top \boldsymbol{x_c} + b_i = \boldsymbol{w}_i^\top \boldsymbol{x}_{\boldsymbol{c}(j \to \bar{c}_j)} + b_i \\
\Rightarrow\ & \boldsymbol{w}_i^\top (\boldsymbol{x_c} - \boldsymbol{x}_{\boldsymbol{c}(j \to \bar{c}_j)}) = 0 \\
\Rightarrow\ & \boldsymbol{w}_i^\top \left( \boldsymbol{u}_{j,c_j} - \boldsymbol{u}_{j,\bar{c}_j} \right) = 0 \\
\Rightarrow\ & \boldsymbol{w}_i^\top \left( \frac{2}{||\boldsymbol{w}_j||^2} \boldsymbol{w}_j \right) = 0 \\
\Rightarrow\ & \boldsymbol{w}_i^\top \boldsymbol{w}_j = 0.
\end{aligned} \tag{39}
$$

Then,

$$(\boldsymbol{u}_{i,c_i} - \boldsymbol{u}_{i,\bar{c}_i})^\top (\boldsymbol{u}_{j,c_j} - \boldsymbol{u}_{j,\bar{c}_j}) \propto \boldsymbol{w}_i^\top \boldsymbol{w}_j = 0. \tag{40}$$

More generally, orthogonality of one concept holds against the span of other concepts as well. For $\{\alpha_j \in \mathbb{R}\}_{j \neq i}$ it follows that

$$(\boldsymbol{u}_{i,c_i} - \boldsymbol{u}_{i,\bar{c}_i})^\top \left( \sum_{j \neq i} \alpha_j (\boldsymbol{u}_{j,c_j} - \boldsymbol{u}_{j,\bar{c}_j}) \right) \propto \boldsymbol{w}_i^\top \left( \sum_{j \neq i} \alpha_j \boldsymbol{w}_j \right) = 0, \tag{41}$$

and therefore orthogonality holds against the span of other concepts differences. $\square$

# H EXAMPLES OF COMPOSITIONALLY GENERALIZABLE REPRESENTATIONS

We give a few instantiations of the linearly-factored representation families: one, where the representations follow a "tight" LRH, and one, in a sense opposite case: where they follow linear independence.

## H.1 CASE 1: MINIMAL DIMENSIONALITY PROBING

To gain intuition into the geometry of the linear probes, let's analyze a more constrained and idealized version of the problem. Instead of a complex joint optimization, we assume the representations are already given and possess a highly regular structure according to the Linear Representation Hypothesis (LRH).

Specifically, we make the following assumptions:

(1) The representation for any input $\boldsymbol{x_v}$ corresponding to a concept value combination $\boldsymbol{v} = (v_1, \ldots, v_c)$ is given by

$$f(\boldsymbol{x_v}) = \sum_{i=1}^{k} \alpha_i(v_i)\boldsymbol{b}_i \tag{42}$$

(2) The concept direction vectors $\{\boldsymbol{b}_i\}_{i=1}^{k} \subset \mathbb{R}^d$ are known, fixed, and linearly independent (implying $d \geq k$). They can be thought of as forming an orthonormal basis for a $k$-dimensional subspace.

(3) For each concept $i$, its $n$ values correspond to a known, ordered set of scalar coefficients. For instance, the values for concept $i$ are mapped to $n$ equally spaced coefficients in an interval, such as $\alpha_i(v_{i,j}) = 0.1 + (j-1)\frac{0.9}{n-1}$ for $j = 1, \ldots, n$.

Under these assumptions, the set of all $n^k$ representation points $\{f(\boldsymbol{x_v})\}$ is fixed and forms a regular grid or lattice within the subspace spanned by $\{\boldsymbol{b}_i\}$. The optimization problem is no longer a search for representations, but simplifies to finding the optimal set of linear probes $\{\boldsymbol{p}_{i,j}\}$ that can correctly classify these points.

The problem becomes:

$$\min_{\{\boldsymbol{p}_{i,j}\}} \quad \sum_{\boldsymbol{v}} \sum_{i=1}^{k} \mathcal{L}_i \left(\{\boldsymbol{p}_{i,j}^{\top} f(\boldsymbol{x_v})\}_{j=1}^{n}, v_i\right) \tag{43}$$

where the representations $f(\boldsymbol{x_v})$ are fixed as defined above. This is a much simpler problem; for standard losses like cross-entropy or hinge loss, this is a convex optimization problem for each set of probes $\{\boldsymbol{p}_{i,j}\}_{j=1}^{n}$ and can be solved efficiently. The key question then becomes understanding the geometric structure of the resulting optimal probes.

Suppose the concept direction vectors $\{\boldsymbol{b}_i\}_{i=1}^{k}$ are linearly independent. In this case, we can write down an explicit analytical solution for the optimal probes. Let $V = \text{span}(\{\boldsymbol{b}_i\}_{i=1}^{k})$ be the subspace spanned by the concept vectors. For each $k \in [k]$, there exists a unique vector $\boldsymbol{w}_k \in V$ such that

$$\boldsymbol{w}_k^{\top} \boldsymbol{b}_i = \delta_{ki} \tag{44}$$

for all $i \in [k]$. In other words, $\boldsymbol{w}_k$ is the unique linear functional that extracts the coefficient of $\boldsymbol{b}_k$ from any vector in $V$ expressed as a linear combination of the $\boldsymbol{b}_i$. This property allows us to construct probes that are perfectly "decoupled" or "disentangled": the classification of one concept is completely unaffected by the values of any other concepts. The vector $\boldsymbol{w}_k$ is the natural choice for isolating the $k$-th concept from the representation.

The optimal affine probes that achieve perfect classification on the given grid of points are, for each concept $k$ and each of its possible values $v_{k,j}$ (for $j = 1, \ldots, n$): (1) **Linear part:** $\boldsymbol{p}_{k,j} = 2\alpha_k(v_{k,j})\boldsymbol{b}_k$, (2) **Bias term:** $b_{k,j} = -(\alpha_k(v_{k,j}))^2$ If the original concept vectors $\{\boldsymbol{b}_i\}$ are orthonormal, then $\boldsymbol{b}_k = \boldsymbol{b}_k$, and this solution reduces to the orthonormal case discussed in the next section.

This construction is optimal because it achieves perfect classification and does so by maximizing the classification margin, making it the solution for max-margin losses (such as those used in SVMs) and for simpler error-counting losses.

Let us verify the score function. The score for the $j$-th probe of concept $k$ on an input $\boldsymbol{x_v}$ (where the true value for concept $k$ is $v_k$) is:

$$S_{k,j}(\boldsymbol{v}) = \boldsymbol{p}_{k,j}^\top f(\boldsymbol{x_v}) + b_{k,j} = (2\alpha_k(v_{k,j})\boldsymbol{b}_k)^\top \left( \sum_{i=1}^{k} \alpha_i(v_i)\boldsymbol{b}_i \right) - (\alpha_k(v_{k,j}))^2$$

$$= 2\alpha_k(v_{k,j})\alpha_k(v_k) - (\alpha_k(v_{k,j}))^2 \quad \text{(since only the } i = k \text{ term survives)}$$

$$= -(\alpha_k(v_k) - \alpha_k(v_{k,j}))^2$$

This score is maximized when $v_k = v_{k,j}$, so the classifier chooses $\arg\max_j S_{k,j}(\boldsymbol{v}) = \arg\min_j (\alpha_k(v_k) - \alpha_k(v_{k,j}))^2$. This is a nearest-neighbor rule that is guaranteed to be correct, thus minimizing the zero-one loss.

The region where class $m$ is predicted is where its coefficient $\alpha_k(v_{k,m})$ is the closest prototype. The decision boundary between any two adjacent classes, $m$ and $m+1$, is the set of points in the 1D space where a point is equidistant to both prototypes:

$$|\alpha_k - \alpha_k(v_{k,m})| = |\alpha_k - \alpha_k(v_{k,m+1})| \tag{45}$$

Given the ordering, this simplifies to $\alpha_k - \alpha_k(v_{k,m}) = -(\alpha_k - \alpha_k(v_{k,m+1}))$, which yields the decision boundary at their exact midpoint:

$$\alpha_k^{DB} = \frac{\alpha_k(v_{k,m}) + \alpha_k(v_{k,m+1})}{2} \tag{46}$$

The margin for separating this pair of classes is the distance from either class's coefficient to this decision boundary, which is $\frac{1}{2}(\alpha_k(v_{k,m+1}) - \alpha_k(v_{k,m}))$. Since our solution places the decision boundary at the midpoint for every adjacent pair, it maximizes the margin for each pair-wise separation. Therefore, it is the optimal max-margin classifier for this 1D problem. The overall margin for concept $k$ is determined by the smallest gap between any two adjacent alpha values.

### H.2 CASE 2: MAXIMUM DIMENSIONALITY PROBING OF CLIP-LIKE MODELS

We now consider the setting where representations are normalized to lie on the unit sphere, as in CLIP-style models that use cosine similarity for classification. Here, both the representation vectors $\boldsymbol{x}$ and the probe vectors $\boldsymbol{p}_{i,j}$ are constrained to have unit $\ell_2$ norm, i.e., $\|\boldsymbol{x}\|_2 = 1$ and $\|\boldsymbol{p}_{i,j}\|_2 = 1$. The geometry of the decision regions is determined by spherical caps rather than half-spaces. For a cosine similarity classifier, the decision region for class $(i, j)$ is given by

$$\mathcal{C}_{i,j} := \left\{ \boldsymbol{x} \in \mathbb{S}^{d-1} : \boldsymbol{p}_{i,j}^\top \boldsymbol{x} > \boldsymbol{p}_{i,k}^\top \boldsymbol{x} \ \forall k \neq j \right\}. \tag{47}$$

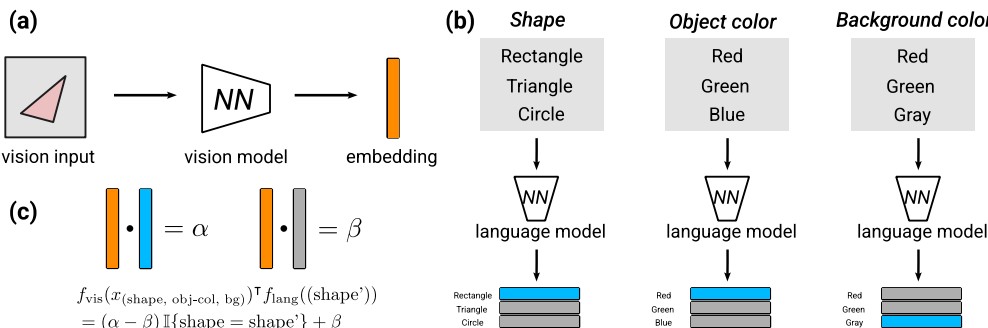

**Figure 30: Illustration of the "on-off concept classifier" mechanism. (a)** A vision input is processed by a neural network to produce an embedding. **(b)** Each concept (e.g., shape, object color, background color) is probed independently using a set of language model probes, one per possible value. **(k)** The probe for a given concept yields a high score $\alpha$ if the concept matches and a lower score $\beta$ otherwise, as formalized in the logit equation at bottom.

That is, for each concept $i$ and classes $j, k$, the cosine similarity satisfies

$$\langle \boldsymbol{x_c}, \boldsymbol{p}_{i,k} \rangle = \begin{cases} 1 & \text{if } j = c_i \\ \beta & \text{if } j \neq c_i \end{cases} \tag{48}$$

for some constant $\beta \in [-1, 1)$.

Under such strict condition, the dimensionality of the representation space must satisfy "all independent" condition. We show this below.

For a probe index $(i, j) \in [k] \times [n]$ we write

$$\boldsymbol{e}_{i,j} \in \mathbb{R}^{cn} \quad \text{for the } (i-1)n + j \text{ standard basis vector, i.e.} \quad (\boldsymbol{e}_{i,j})_{(k,\ell)} = \begin{cases} 1, & k = i, \ \ell = j, \\ 0, & \text{otherwise.} \end{cases}$$

In words, $e_{i,j}$ has a single 1 in the row corresponding to probe $(i, j)$ and 0 elsewhere.

**Proposition 6** (Minimal dimensionality from fixed dot-products). Fix integers $k \geq 1$ (number of concepts) and $n \geq 2$ (values per concept). For each concept $i \in [k]$ and value $j \in [n]$ let

$$\boldsymbol{p}_{i,j} \in \mathbb{R}^d, \qquad \|\boldsymbol{p}_{i,j}\|_2 = 1,$$

be unit *probe* vectors, and for each complete concept tuple $\boldsymbol{v} = (v_1, \ldots, v_c) \in [n]^k$ let

$$\boldsymbol{x_v} \in \mathbb{R}^d, \qquad \|\boldsymbol{x_v}\|_2 = 1,$$

be unit *representations*. Assume there exist constants $\alpha, \beta \in [-1, 1]$ with $\alpha \neq \beta$ such that the fixed logit pattern

$$\boldsymbol{p}_{i,j}^\top \boldsymbol{x_v} = \begin{cases} \alpha, & j = v_i, \\ \beta, & j \neq v_i, \end{cases} \qquad \text{for } all \ i, j, \boldsymbol{v}, \tag{49}$$

holds.

Then the ambient dimension $d$ must satisfy

$$d \geq 1 + k(n - 1). \tag{50}$$

Moreover, this bound is tight: for any valid $(\alpha, \beta)$ with $|\alpha| \leq 1$, $|\beta| \leq 1$ there exist explicit probe/representation families that realise (49) in dimension $d = 1 + k(n - 1)$.

*Proof.* We stack the probes as rows of the matrix

$$P = \begin{bmatrix} \boldsymbol{p}_{1,1}^\top \\ \vdots \\ \boldsymbol{p}_{k,n}^\top \end{bmatrix} \in \mathbb{R}^{cn \times d}, \qquad (\text{row } (i-1)n + j = \boldsymbol{p}_{i,j}^\top). \tag{51}$$

Stack the representations as columns of

$$X = \begin{bmatrix} \boldsymbol{x}_{\boldsymbol{v}_1} & \cdots & \boldsymbol{x}_{\boldsymbol{v}_{n^k}} \end{bmatrix} \in \mathbb{R}^{d \times n^k}. \tag{52}$$

The logit constraints (49) read as

$$Y = PX \in \mathbb{R}^{cn \times n^k}, \tag{53}$$

where $Y \in \mathbb{R}^{cn \times n^k}$ has entries

$$Y_{(i-1)n+j, \, \boldsymbol{v}} = \begin{cases} \alpha, & j = v_i, \\ \beta, & j \neq v_i. \end{cases} \tag{54}$$

For one concept $k = 1$, (when $Y \in \mathbb{R}^{n \times n}$), the single block is

$$(\alpha - \beta)I_n + \beta \mathbf{1}_n \mathbf{1}_n^\top \tag{55}$$

which has full rank $n$ because $\alpha \neq \beta$. Its row–space is therefore spanned by

$$\underbrace{\{\mathbf{1}_{n^k}\}}_{\text{global offset}} \cup \underbrace{\{\mathbf{1}\{v_i = j\} - \mathbf{1}\{v_i = 1\} \mid i \in [k], \ j = 2, \ldots, n\}}_{k(n-1) \text{ zero–sum contrast vectors}}.$$

The contrast vectors all have coordinate-sum $0$, whereas $\mathbf{1}_{n^k}$ has sum $n^k$; hence $\mathbf{1}_{n^k} \notin \mathrm{span}\{\text{contrasts}\}$. The total of $1 + k(n-1)$ vectors is therefore linearly independent, giving

$$\mathrm{rank}(Y) = 1 + k(n-1). \tag{56}$$

Because $Y = PX$,

$$1 + k(n-1) = \mathrm{rank}(Y) \leq \mathrm{rank}(P) \leq d. \tag{57}$$

This proves (50).

Construction follows by placing the probes and representations on the unit sphere in independent directions. $\qquad\square$

Below, we provide a numerical example to illustrate the form of the logit matrix $Y$ for the case of two concepts, three values each.

**Example 1** (Two concepts, three values each: $k = 2$, $n = 3$). Set $(\alpha, \beta) = (1, 0.2)$. The row indices are $(i, j) \in \{1, 2\} \times \{1, 2, 3\}$, the column indices are the $3^2 = 9$ tuples $(v_1, v_2) \in \{1, 2, 3\}^2$:

$$
Y = 
\begin{array}{c}
\\
(1,1) \\
(1,2) \\
(1,3) \\
(2,1) \\
(2,2) \\
(2,3)
\end{array}
\begin{array}{ccccccccc}
11 & 12 & 13 & 21 & 22 & 23 & 31 & 32 & 33 \\
\left(\begin{array}{c}1\end{array}\right. & 1 & 1 & 0.2 & 0.2 & 0.2 & 0.2 & 0.2 & \left.\begin{array}{c}0.2\end{array}\right) \\
0.2 & 0.2 & 0.2 & 1 & 1 & 1 & 0.2 & 0.2 & 0.2 \\
0.2 & 0.2 & 0.2 & 0.2 & 0.2 & 0.2 & 1 & 1 & 1 \\
1 & 0.2 & 0.2 & 1 & 0.2 & 0.2 & 1 & 0.2 & 0.2 \\
0.2 & 1 & 0.2 & 0.2 & 1 & 0.2 & 0.2 & 1 & 0.2 \\
0.2 & 0.2 & 1 & 0.2 & 0.2 & 1 & 0.2 & 0.2 & 1
\end{array}
\tag{58}
$$

**Row-space decomposition.** Each row has the form

$$\beta \mathbf{1}_9 + (\alpha - \beta)\mathbf{1}\{v_i = j\}, \tag{59}$$

so every row is in the span of

$$\mathbf{1}_9, \quad \underbrace{\mathbf{1}\{v_1 = 2\} - \mathbf{1}\{v_1 = 1\},\ \mathbf{1}\{v_1 = 3\} - \mathbf{1}\{v_1 = 1\}}_{n-1 \text{ contrasts for concept 1}}, \quad \underbrace{\mathbf{1}\{v_2 = 2\} - \mathbf{1}\{v_2 = 1\},\ \mathbf{1}\{v_2 = 3\} - \mathbf{1}\{v_2 = 1\}}_{n-1 \text{ contrasts for concept 2}}. \tag{60}$$

That is a set of $1 + 2(3-1) = 5$ linearly independent vectors, hence $\mathrm{rank}(Y) = 5 = 1 + k(n-1)$.

Under such a design, linear factorization holds immediately.

**Proposition 7** (Additive factorisation from the on–off pattern). Let $k \geq 1$ (concepts) and $n \geq 2$ (values per concept). Assume there are unit vectors

$$\boldsymbol{p}_{i,j} \in \mathbb{R}^d, \quad i \in [k],\ j \in [n], \qquad \boldsymbol{x}_{\boldsymbol{v}} \in \mathbb{R}^d, \quad \boldsymbol{v} = (v_1, \dots, v_c) \in [n]^k,$$

and two real numbers $\alpha \neq \beta$ in $(-1, 1)$ such that

$$\langle \boldsymbol{p}_{i,j}, \boldsymbol{x}_{\boldsymbol{v}} \rangle = \begin{cases} \alpha & \text{if } j = v_i, \\ \beta & \text{if } j \neq v_i, \end{cases} \qquad \forall\, i, j, \boldsymbol{v}. \tag{61}$$

Define the global mean, conditional means, and shift vectors from $\{\boldsymbol{x}_{\boldsymbol{v}}\}$ as:

$$g := \frac{1}{n^k} \sum_{\boldsymbol{w} \in [n]^k} \boldsymbol{x}_{\boldsymbol{w}}, \qquad A_{i,j} := \frac{1}{n^{k-1}} \sum_{\boldsymbol{w}:\, w_i = j} \boldsymbol{x}_{\boldsymbol{w}}, \qquad u_{i,j} := A_{i,j} - g.$$

Now, for each class $\boldsymbol{v} = (v_1, \dots, v_c)$, define the reconstructed vector

$$\tilde{\boldsymbol{x}}_{\boldsymbol{v}} := g + \sum_{k=1}^{k} u_{k, v_k}. \tag{62}$$

Then:

1. This reconstructed vector $\tilde{\boldsymbol{x}}_{\boldsymbol{v}}$ satisfies the original on–off pattern. That is, for every probe $\boldsymbol{p}_{i,j}$ and every class $\boldsymbol{v}$,

$$\langle \boldsymbol{p}_{i,j}, \tilde{\boldsymbol{x}}_{\boldsymbol{v}} \rangle = \langle \boldsymbol{p}_{i,j}, \boldsymbol{x}_{\boldsymbol{v}} \rangle = \begin{cases} \alpha & \text{if } j = v_i, \\ \beta & \text{if } j \neq v_i. \end{cases} \tag{63}$$

This means $\tilde{\boldsymbol{x}}_{\boldsymbol{v}}$ is indistinguishable from $\boldsymbol{x}_{\boldsymbol{v}}$ by the probes and is sufficient for any classification task based on these dot products.

2. Moreover, the set of vectors $\{\tilde{\boldsymbol{x}}_{\boldsymbol{v}}\}$ lies in an affine subspace of dimension exactly $1+k(n-1)$. So:

$$\dim(\operatorname{span}\{\tilde{\boldsymbol{x}}_{\boldsymbol{v}}\}) = 1 + k(n-1). \tag{64}$$

*Proof.* Fix $(i,j)$. Averaging (61) over all $n^k$ classes $\boldsymbol{w}$ gives

$$\langle \boldsymbol{p}_{i,j}, \boldsymbol{g} \rangle = \frac{1}{n^k}\left(n^{k-1}\alpha + (n^k - n^{k-1})\beta\right) = \frac{\alpha + (n-1)\beta}{n} =: d. \tag{65}$$

independent of $(i,j)$.

Then, compute $\langle \boldsymbol{p}_{i',k}, A_{i,j} \rangle$ by expanding the definition of $A_{i,j}$:

$$\langle \boldsymbol{p}_{i',k}, A_{i,j} \rangle = \frac{1}{n^{k-1}} \sum_{\boldsymbol{w}: w_i = j} \langle \boldsymbol{p}_{i',k}, \boldsymbol{x}_{\boldsymbol{w}} \rangle. \tag{66}$$

We consider two cases for the probe index $i'$.

**Case 1: $i' = i$ (probe and condition on the same concept).** The sum is over $\boldsymbol{w}$ where $w_i = j$.

- If $k = j$, the probe is $\boldsymbol{p}_{i,j}$. For every term in the sum, $w_i = j$, so $\langle \boldsymbol{p}_{i,j}, \boldsymbol{x}_{\boldsymbol{w}} \rangle = \alpha$. There are $n^{k-1}$ such terms, so the sum is $n^{k-1}\alpha$. The average is $\alpha$.

- If $k \neq j$, the probe is $\boldsymbol{p}_{i,k}$. For every term, $w_i = j \neq k$, so $\langle \boldsymbol{p}_{i,k}, \boldsymbol{x}_{\boldsymbol{w}} \rangle = \beta$. The sum is $n^{k-1}\beta$. The average is $\beta$.

**Case 2: $i' \neq i$ (probe and condition on different concepts).** The sum is still over all $n^{k-1}$ vectors $\boldsymbol{w}$ where $w_i = j$. For a given probe $\boldsymbol{p}_{i',k}$, the value of $\langle \boldsymbol{p}_{i',k}, \boldsymbol{x}_{\boldsymbol{w}} \rangle$ depends on whether $w_{i'} = k$ or $w_{i'} \neq k$. Since $i' \neq i$, the condition $w_i = j$ does not fix the value of $w_{i'}$.

- The number of vectors $\boldsymbol{w}$ with $w_i = j$ *and* $w_{i'} = k$ is $n^{k-2}$ (since two components are fixed, and $k - 2$ are free). For these terms, $\langle \boldsymbol{p}_{i',k}, \boldsymbol{x}_{\boldsymbol{w}} \rangle = \alpha$.

- The number of vectors $\boldsymbol{w}$ with $w_i = j$ *and* $w_{i'} \neq k$ is $(n-1)n^{k-2}$ (one component fixed, one has $n - 1$ choices, $k - 2$ are free). For these terms, $\langle \boldsymbol{p}_{i',k}, \boldsymbol{x}_{\boldsymbol{w}} \rangle = \beta$.

The sum (66) is therefore (when $i' \neq i$)

$$n^{k-2}\alpha + (n-1)n^{k-2}\beta. \tag{67}$$

The average is:

$$\langle \boldsymbol{p}_{i',k}, A_{i,j} \rangle = \frac{n^{k-2}\alpha + (n-1)n^{k-2}\beta}{n^{k-1}} = \frac{\alpha + (n-1)\beta}{n} = d. \tag{68}$$

Combining these cases, we have:

$$\langle \boldsymbol{p}_{i',k}, A_{i,j} \rangle = \begin{cases} \alpha, & i' = i, \ k = j, \\ \beta, & i' = i, \ k \neq j, \\ d, & i' \neq i. \end{cases}$$

By linearity, $\langle \boldsymbol{p}_{i',k}, u_{i,j} \rangle = \langle \boldsymbol{p}_{i',k}, A_{i,j} \rangle - \langle \boldsymbol{p}_{i',k}, g \rangle$. The results from steps 1 and 2 give:

$$\langle \boldsymbol{p}_{i',k}, u_{i,j} \rangle = \begin{cases} \alpha - d, & i' = i, \; k = j, \\ \beta - d, & i' = i, \; k \neq k, \\ 0, & i' \neq i. \end{cases} \tag{69}$$

Finally, by evaluation, it follows that $\tilde{\boldsymbol{x}}_{\boldsymbol{v}} = g + \sum_{k=1}^{k} u_{k,v_k}$ satisfies the on-off pattern:

$$\begin{aligned} \langle \boldsymbol{p}_{i,j}, \tilde{\boldsymbol{x}}_{\boldsymbol{v}} \rangle &= \langle \boldsymbol{p}_{i,j}, g \rangle + \sum_{k=1}^{k} \langle \boldsymbol{p}_{i,j}, u_{k,v_k} \rangle \\ &= d + \langle \boldsymbol{p}_{i,j}, u_{i,v_i} \rangle + \sum_{k \neq i} \underbrace{\langle \boldsymbol{p}_{i,j}, u_{k,v_k} \rangle}_{=0 \text{ from (69)}} \\ &= d + (\langle \boldsymbol{p}_{i,j}, A_{i,v_i} \rangle - d) = \langle \boldsymbol{p}_{i,j}, A_{i,v_i} \rangle \\ &= \begin{cases} \alpha, & j = v_i, \\ \beta, & j \neq v_i. \end{cases} \end{aligned}$$

This confirms that $\langle \boldsymbol{p}_{i,j}, \tilde{\boldsymbol{x}}_{\boldsymbol{v}} \rangle = \langle \boldsymbol{p}_{i,j}, \boldsymbol{x}_{\boldsymbol{v}} \rangle$ for all probes, and establishes (63).

The reconstructed vectors $\{\tilde{\boldsymbol{x}}_{\boldsymbol{v}}\}$ are all affine combinations of $\{g\} \cup \{\boldsymbol{u}_{i,j}\}$. A basis for this affine space can be formed by $\{g\}$ and the differences $\{\boldsymbol{u}_{i,j} - \boldsymbol{u}_{i,1} \mid i \in [k], \; j = 2, \ldots, n\}$, a set of $1 + k(n-1)$ vectors. These are linearly independent because contrasts from different concepts are orthogonal (with respect to probes), and within a concept, independence follows from $\alpha \neq \beta$. Thus, the set $\{\tilde{\boldsymbol{x}}_{\boldsymbol{v}}\}$ lies in an affine subspace of dimension exactly $1 + k(n-1)$. This establishes (64). $\qquad \square$

# I WHAT IF STABILITY IS NOT REQUIRED?

We detail and discuss the stability axiom in the main text. Suppose it was not true, what other structure does the representation need to have?

## I.1 COUNTEREXAMPLES TO LINEAR FACTORIZATION EVEN AS $n \to \infty$

Suppose that instead of assuming a transferable compositional model, we *only* assume the model supports linear separation. That is, given $n^k$ datapoints in total, let's suppose there exist $n \cdot k$ linear probes that can be used to classify each concept value for any datapoint. (Formally: there are $k$ concepts indexed by $j \in \{1, \ldots, k\}$, each with $n$ values indexed by $k \in \{1, \ldots, n\}$; a datapoint is $t = (k_1, \ldots, k_c) \in \{1, \ldots, n\}^k$; a representation map $f : \mathcal{X} \to \mathbb{R}^d$ yields $z_t := f(x_t)$; and for each concept $j$ there are weights and biases $\{(w_{j,k}, b_{j,k})\}_{k=1}^n$ with $\arg\max_k (w_{j,k}^\top z_t + b_{j,k}) = k_j$.)

Does such a construct imply a certain representational structure? Perhaps—but it is not, in general, linearly factorizable. Concretely, suppose we restrict ourselves to a two-dimensional representation space. Assume it's Euclidean and the linear probes are weight vectors with biases. Additionally, assume there are only two concepts that the data is distributed over. Now, given that there are $n$ values, is there some structure that the representations need to converge to as $n \to \infty$? Not necessarily: even in this $d = 2$, $k = 2$ setting, one can satisfy all the linear separability probes with point clouds $\{z_{k_1, k_2}\} \subset \mathbb{R}^2$ that do *not* admit an additive decomposition of the form $z_{k_1, k_2} = u_0 + u_{1, k_1} + u_{2, k_2}$. This is the sense in which linear separability does not imply linear factorizability.

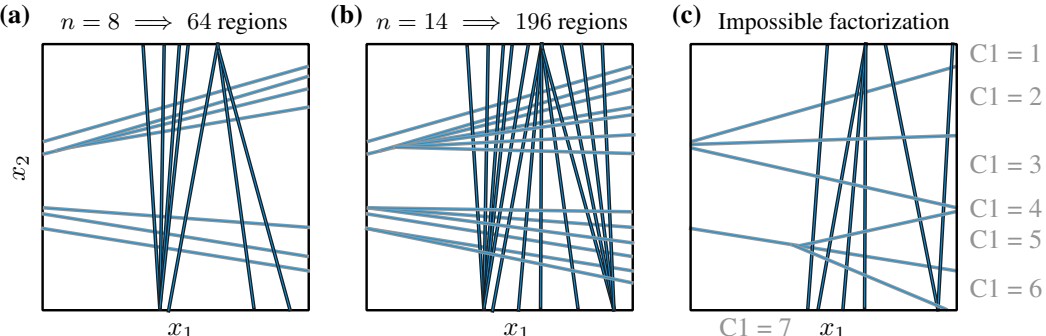

**(a)** $n = 8 \implies 64$ regions    **(b)** $n = 14 \implies 196$ regions    **(c)** Impossible factorization

Figure 31: **Linear separability without linear factorization.** Two families of affine decision boundaries in $\mathbb{R}^2$ (black for concept 1, gray for concept 2) divide the plane into regions, one per pair of concept values. Panels (a,b): with $n = 8$ and $n = 14$ levels per concept the arrangement yields $n^2$ regions (64 and 196). By inserting additional nearly-parallel boundaries, existing regions can be split into smaller and smaller pieces, creating arbitrarily tiny regions while maintaining perfect linear separability. Panel (k): No linear factorization can be achieved: whichever factors we pick, the separability of some datapoints are violated.

From Figure 31: panels (a) and (b) show two interleaved line families whose intersections produce a grid of $n^2$ convex cells, one for each $(k_1, k_2)$. Nothing forces these cells to align with an additive basis; in fact, we can keep adding lines that are $\varepsilon$-perturbations of existing ones to subdivide cells, driving some cell areas to zero as $n$ grows, yet all multiclass linear probes remain valid.

## THE USAGE OF LLMS

In accordance with ICLR 2026 policy, we disclose that large language models were used to assist in text editing and polishing of writing. All research ideas, experiments, and analyses were conducted by the authors.

