# OpenReview forum: "Necessary Conditions for Compositional Generalization in Visual Models"
_ICLR.cc/2026/Conference — Submitted to ICLR 2026_

### Official Review · Reviewer_xU4a · 2025-10-20

**Soundness:** 3
**Presentation:** 2
**Contribution:** 2
**Rating:** 6
**Confidence:** 4

**Summary:**

The paper investigates compositional generalization from a formal perspective. The authors put forward three properties (divisibility, transferability, and stability) which they argue argue should be fulfilled by a system achieving compositional generalization. From these, they derive a set of implications (e.g., additive linear factorization, cross-concept orthogonality)  that the representations of an encoder model need to satisfy to generalize compositionally under a linear readout. Finally, the authors provide some empirical evidence that the representations of CLIP and SigLIP models exhibit linearity and orthogonality (above a random encoder baseline) and that the degree of linearity correlates positively with compositional generalization performance.

**Strengths:**

- The approach of looking at compositional generalization from an idealized perspective (the proposed desiderata) and then formally deriving necessary properties of the corresponding representations is good and distinguishes this work from many other papers that study compositional generalization in existing (imperfect) models which usually do not achieve full transferability.
- Formally deriving the necessity of intuitive properties such as linearity and cross-concept orthogonality in the studied setting (Proposition 1) is a nice result and likely hints why compositional generalization is usually very challenging in practice.
- Finding evidence for the idealized requirements in actual CLIP/SigLIP models significantly strengthens the credibility of the claim and the proposed desiderata. Further, showing a positive correlation between linearity and compositional generalization also supports the theoretic findings.

**Weaknesses:**

- The figures of the paper could be noticeably improved. Most figures (e.g., Figure 1 & 6) are not easily understandable by just looking at them as well as not properly explained in the caption or referencing text. For example, for Figure 6 I assume that it depicts two scenarios where the bound $d \geq c$ is tight for one but not the other but it is not properly explained why this is true. In general, it could help to not only state what the figure does (e.g., Fig. 3/4) but also give an explanation or interpretation in each caption.
- While more realistic datasets with compositional structure often only have few concepts and contain some noisy or wrong concept labels, I think it would significantly strengthen the work to also include one or two more such datasets in the analysis, e.g., DomainNet [1] or ImageNet-AO [2].

### Comments
- It would be nice to provide short proof sketches or intuitions in the main text (similar as for Proposition 1) for the other propositions to give some credibility and/or confidence about the respective statements.
- I think the paper would benefit from a short discussion to better contextualize some of its findings. For example, perfect compositional generalization requiring linear factorized concept embeddings is something many people might have expected to be true, so the finding might not appear as interesting. Further emphasizing the added value of providing a theoretic derivation for this could be useful.

---
[1] Xingchao Peng, Qinxun Bai, Xide Xia, Zijun Huang, Kate Saenko, and Bo Wang. "Moment matching for multi-source domain adaptation." International Conference on Computer Vision. 2019.

[2] Reza Abbasi, Mohammad Hossein Rohban, and Mahdieh Soleymani Baghshah. "Deciphering the role of representation disentanglement: Investigating compositional generalization in CLIP models." European Conference on Computer Vision. 2024.

**Questions:**

- Another work by Abbasi et al. [1] has also investigated compositional generalization and its correlation with representation disentanglement. They find that higher disentanglement also correlates with better compositional generalization. Can the authors either discuss the findings of [1] as related work and/or in context of the proposed desiderata or argue why the work might not be relevant?
- I'm a bit confused by Definition 3. It intuitively states that an encoder $f$ is compositional if there exists **any** function $h$ that correctly classifies all concepts $c_i$ from $f(\mathbf{x_c})$. In the general case (where $h$ can be non-linear), wouldn't any injective encoder $f$ be compositional then (simply define $h : f(\mathbf{x_c}) \mapsto \mathrm{onehot(c)}$)? I understand that such a function would likely not be learned in practice but from mathematical perspective it fulfills the definition. Maybe the definition could either focus only on linear functions $h$ where this cannot happen or be adapted in some other way that accounts for this.

---
[1] Reza Abbasi, Mohammad Hossein Rohban, and Mahdieh Soleymani Baghshah. "Deciphering the role of representation disentanglement: Investigating compositional generalization in CLIP models." European Conference on Computer Vision. 2024.

---

> ### Author Response · Authors · 2025-11-24
>
> We thank you for your thorough review.
>
> ## Figs. 1,6 aren't easy to parse.
> We have updated the figures to be easier to parse and added more explanations in the main text.
>
> ## Realistic/noisy datasets would strengthen the work
> Thank you for the suggestions - ImageNet-AO seemed like an appropriate choice: it's both modern (i.e. post-CLIP) and concept-oriented (nouns & actions). We have added the results to the main text and Appendix D.3 ("Experiments on ImageNet-AO"). In short, the conclusions from the main text hold there as well - better $R^2$ models yield better compositional accuracy. Additionally, cross-concept factors are more orthogonal than within-concept ones.
>
> We will revise the referencing in the main text in the next revision.
>
> ## Intuition for proofs
>
> We have added more intuition for the proofs in the main text (e.g. in Section 4.2).
>
> ## Contextualization of the results would be useful
>
> We have added a discussion of the related work in the main text (Section 2) and Appendix.
>
> ## Previous work on connections between disentanglement and compositional generalization.
>
> We believe the work is very relevant, thank you for pointing it out. We have discussed it in the main text and extended the related work section (Appendix B).
>
> ## Don't compositional models always exist as long as they're injective?
>
> Yes, they do. We have clarified this in the text (Section 3.2).

---

### Official Review · Reviewer_VaZz · 2025-10-30

**Soundness:** 2
**Presentation:** 2
**Contribution:** 1
**Rating:** 2
**Confidence:** 5

**Summary:**

The authors propose a set of necessary conditions that  compositional representations should posses. They call these desiderata and include several sensible notions that seem intuitive: divisibility, transferability and stability. These crystallize the notion that compositional representations should be identifiable (divisibility), that this should hold for unseen combinations (transferability) and that these should remain constant across different spaces which share concepts (stability). They authors emphasize how these conditions lead to representations that show compositional generalization.

**Strengths:**

The authors are very thorough in the way they define their desiderata. Furthermore they explore the implications of said desiderata for the geometry of the learned representations across several models in the Visual Language Model category.

**Weaknesses:**

The main weakness is that similar ideas and considerations have already been explored under a different name.

The idea of representations being divisible, stable and transferable was already explored in the context of disentangled representations, which essentially follow the desiderata proposed here (Higgins et al., and Watters et al and especially, Eastwood and Williams. Note: there are two issues discussed in this literature, how to learn disentangled representations, and whether this lead to compositionally). Incidentally, Watters et al., already explored the implications that this has for compositional generalization, though admittedly this is in their appendix and is not very systematic. A more systematic exploration was done in Montero et al., which showed that even if the equivalent of the desiderata above held, models those models couldn't generalize (see the result for the GT decoder. Note that this is pre LLM, so those weren't considered). I include this because this is very relevant literature which the authors do not reference, or mention yet covers much of the ground that they are retreading.

As for the implications for compositional generalization, we must start by discussing a fundamental misunderstanding about what compositionality is. Briefly, the definition of compositionality states that the meaning of a proposition depends on the semantic meaning of its parts, and the relations between them (i.e how they are bound into some "expression" whatever form this may take). In the case of the generalization part, it states that you should be able to recombine parts in order to perform some task and exhibit two properties systematicity and productivity( though the latter is disputed. See the Fodor and Pylyshyn reference).

In my view, there are thus two issues here: first, the authors are studying pre-trained models which they cannot say have not seen the dataset in question (or similar enough ones) and thus it is not true that they are generalizing. The only claim that they can make is that their (linear) probes, are compositional because they learn to predict specific concept values for unseen combinations. This is fine, but servers more as an exploration of how linear the geometry of these models are, then about compositional generalization more broadly.

Second, simple prediction is really just the bare minimum since it doesn't require models understand how the different concepts relate to each other. If they can read out the information from the representation then they can ignore all other information. This is why the studies above use image generation: it forces the model to learn how the different factors interact i.e. how the semantic meaning of the full proposition depends on the semantics of the parts and how they relate to each other.


References:
Higgins, I., Amos, D., Pfau, D., Racaniere, S., Matthey, L., Rezende, D., & Lerchner, A. (2018). Towards a definition of disentangled representations. arXiv preprint arXiv:1812.02230

Watters, N., Matthey, L., Burgess, C. P., & Lerchner, A. (2019). Spatial broadcast decoder: A simple architecture for learning disentangled representations in vaes. arXiv preprint arXiv:1901.07017.

C. Eastwood and C. K. I. Williams, “A framework for the quantitative evaluation of disentangled representations,”
in International Conference on Learning Representations, 2018.

Montero, M. L., Ludwig, C. J., Costa, R. P., Malhotra, G., & Bowers, J. (2021). The role of disentanglement in generalisation. In International Conference on Learning Representations.

**Questions:**

1. The authors need to explain how their proposals are different from the ones mentioned above
2. Also how it is possible to control for the issue of VLMs having access to the test data.
3. Can they find a harder task that shows stronger compositionally?

---

> ### Author Response · Authors · 2025-11-24
>
> Thank you for your thorough review.
>
> ## The idea of representations being divisible, stable and transferable was already explored in the context of disentangled representations, which essentially follow the desiderata proposed here (Higgins et al., and Watters et al and especially, Eastwood and Williams. Note: there are two issues discussed in this literature, how to learn disentangled representations, and whether this lead to compositionally).
>
> Thank you for the references - we should have discussed them in detail in the paper to make it clear how our work relates to the domain of disentangled representation learning.
>
> The key distinction is one of direction and the role of data sampling in training. Some of the disentanglement literature studies whether learning disentangled representations via architectural priors or training objectives is sufficient for compositional generalization, additionally measuring success using metrics such as DCI (disentanglement, completeness, informativeness). In other words, they impose or encourage structure and ask: does this lead to compositional generalization?
>
> We ask the opposite: if a discriminative model does generalize compositionally when trained on subsets of the combinatorial concept space, what must necessarily be true of its embeddings? Our approach does not impose any constraints on the representations a priori. Instead, we define model-level desiderata - properties of the model's behavior across different training subsets:
>
> - Our desiderata say that for a model to be considered generalizing compositionally, it needs to:
>     - (1) have sufficient capacity to accommodate the combinatorial concept space (of $n^c$ combinations) - Divisibility,
>     - (2) recognize novel compositions of known parts when trained only on a subset of the concept space - Transferability (this is the core data sampling requirement), and,
>     - (3) maintain consistent "understanding" of individual concepts regardless of which particular subset of combinations appears in training - Stability.
>
> From these desiderata, we derive that linear factorization and cross-concept-orthogonality must emerge as necessary consequences under standard training in model types like CLIP that do not enforce any disentanglement priors. The structure emerges from the requirement to generalize across varying data subsets, not from objective constraints.
>
> We have now clarified this distinction in Sec. 3.3 to make it explicit that the desiderata are model- and not representation-level.
>
> Additionally, we discuss the relation to the disentangled representation learning domain more broadly in the related work in the updated PDF.
>
> ##  Incidentally, Watters et al., already explored the implications that this has for compositional generalization, though admittedly this is in their appendix and is not very systematic.
>
> We appreciate their qualitative results showing that VAE-based models sometimes infer "consistent" latents on unseen x-y coordinate combinations. We have included this description in the related work section. Our work differs in that we don't attempt to inject structure into models explicitly (as large scale systems like CLIP almost never do), but instead derive what must emerge if models do generalize.
>
> ## A more systematic exploration was done in Montero et al., which showed that even if the equivalent of the desiderata above held, models those models couldn't generalize (see the result for the GT decoder. Note that this is pre LLM, so those weren't considered).
>
> We study a different question. Montero et al. investigated whether imposing disentanglement on small-scale VAEs at train time leads to compositional generalization on unseen combinations, finding it often doesn't. We instead ask: what geometric properties must representations have IF a model does achieve compositional generalization from a limited subset of data?
>
> Modern large-scale models like CLIP show some compositional generalization capabilities, but as we document in our related work, they still fail on many compositional tasks. Our work identifies necessary geometric structure for reliable compositional generalization. Montero et al. show that certain approaches don't achieve this in practice; we characterize what must be true when it does work.

---

> > ### Author Response · Authors · 2025-11-24
> >
> > ## In the case of the generalization part, it states that you should be able to recombine parts in order to perform some task and exhibit systematicity and productivity
> >
> > We agree that systematicity (the ability to produce/understand novel combinations of known parts) is central to compositionality. However, there are two complementary aspects: (1) combining known parts into novel wholes (generation), and (2) decomposing novel wholes into known parts (recognition/discrimination).
> >
> > We study the latter. The space of concept combinations is vast - training on all combinations is infeasible. Therefore, systems must be able to decompose novel compositions into their constituent concepts to generalize. This is the inverse of the generative process: where generative models learn how concepts compose into observations, discriminative models must learn how to decompose observations into concepts.
> >
> >
> > ## The authors are studying pre-trained models which they cannot say have not seen the dataset in question (or similar enough ones) and thus it is not true that they are generalizing.  / How it is possible to control for the issue of VLMs having access to the test data
> >
> > We appreciate this concern. Our contribution is primarily theoretical: deriving necessary geometric properties from model-level desiderata. The empirical component surveys whether existing models exhibit these predicted properties -- not necessarily to validate that they generalize, but to explore whether models showing some generalization (however achieved) align with our theoretical predictions.
> >
> > Regarding the datasets: some were unlikely to have been seen during pretraining. PUG-Animal was released in 2023 (post-CLIP), and MPI3D is not typically found in web-scraped collections like LAION. Additionally, we have added results on ImageNet-AO (Appendix D.3), a 2024 dataset specifically designed with unlikely combinations of concept combinations. However, we cannot entirely rule out exposure to similar visual distributions, and acknowledge that this is a difficult problem to control for.
> >
> >
> > ## Second, simple prediction is really just the bare minimum since it doesn't require models understand how the different concepts relate to each other. If they can read out the information from the representation then they can ignore all other information. This is why the studies above use image generation: it forces the model to learn how the different factors interact i.e. how the semantic meaning of the full proposition depends on the semantics of the parts and how they relate to each other.
> >
> > The compositional generalization accuracy we measure requires correctly inferring _all_ concepts and their values on novel combinations - the studied datasets expose all factors of variation, so no information is ignored. This evaluation approach is directly analogous to the generative modeling work cited (VAEs with disentanglement objectives), which also evaluates by extracting and analyzing representations. It is also analogous to metrics like DCI, which measure how well embeddings support predicting all factors of variation by training classifiers on representations and analyzing prediction distributions.
> >
> >
> > ## Harder tasks that show stronger compositionally
> >
> > Since there are two ways to construe the questions, we answer both:
> >
> > 1. In case you meant "stronger" compositionality in terms of the semantic composition complexity - e.g. due to relations like binding of attributes to certain objects, or expressing relationships between objects. Then these are out of scope. The compositionality we consider is quite standard, especially in comparison to disentangled representation learning literature (exemplified by datasets like DSprites). We use the similar datasets. Datasets like PUG-Animal, however, express additional complexities in the scene, due to viewpoints, shadows, and object-specific renderings of textures on objects.
> > 2. In case you meant "stronger" compositionally in terms of visual complexity. We have since added a coarse-grained and visually complex dataset used to probe CLIP's discrimiantive compositional understanding dubbed ImageNet-AO [REF]. We have added the results are available in Appendix C.3. In short, all of the conclusions from the main text hold.
> >
> > [a] Reza Abbasi, Mohammad Hossein Rohban, and Mahdieh Soleymani Baghshah. "Deciphering the role of representation disentanglement: Investigating compositional generalization in CLIP models."

---

> > > ### Comment · Reviewer_VaZz · 2025-11-26
> > >
> > > I thank the authors for their replies. However, I remained unconvinced. I realize that my initial review was not clear enough and failed to establish a more productive discussion. I will try to restate my concerns.
> > >
> > > First, I agree with the authors' desiderata as stated in plain words, I just don't think these necessary conditions are novel since 2 and 3 are just the definition of combinatorial generalization. That is, these are not necessary, they are necessary and sufficient and the rest is tautological (a model generalizes if it does everything that is stated in the definition of generalization). Second, and as a corollary, I believe the authors are misusing the concept. Combinatorial generalization and compositionality are not the same, with the later being one mechanism by which you can achieve the former. This leads me to my third concern which is that the authors are studying a particular form of combinatorial generalization: namely the ability to decompose an input into constituent concepts when presented with novel combinations. We could call this "semantic decomposition". How these two mechanisms (compositionality and semantic decomposition) interact when solving a particular task is definitely an interesting question, but not one that is being explored in this work.
> > >
> > > The reason for this conflation is why I brought up the previous work on measuring disentanglement since it follows a similar path as the one taken here: given some embedding from a pre-trained model, measure how disentangled they are by measuring their predictive capability using a (non) linear classification. These classifiers failed at generalization for similar reasons as the ones investigated here: the embedding space they had to predict from was not "linear enough" (in the case of the linear classifiers) and so novel combinations were projected into un-expected areas of the latent space (see Schott and Montero). Except in that case the encoders were also tested for generalization (that is they were presented with held-out data) while here only the probes are tested in this way (since by their own admission the authors cannot guarantee that similar or even the same datasets were not seen during training by CLIP).
> > >
> > > Thus the article is left in a weird position. It claims to propose necessary conditions for compositional generalization while just restating the definition of combinatorial generalization, then it focuses on a particular flavor of the generalization task which may not require compositionality, which further changes from the original framing as it introduces further assumptions during the experimental setup (by focusing on linear probes on pre-trained models). All of which makes the results not as surprising in light of previous studies in the disentanglement literature/combinatorial generalization literature.
> > >
> > > I think the authors can still make this a good article but I would recommend rephrasing it. First, forget about compositionality, since I as I said, I don't believe that this is the experiments study. They can focus directly on the idea of semantic decomposition, framing it as a desirable step on the road to perform compositionality. Finally they can either study how easy it is to generalize from pre-trained (and thus probably in-distribution) models or train new models from scratch to establish if they are learning easy to decode linear representations (which is what they are ultimately trying to show anyway).
> > >
> > >
> > > Some further clarifications/questions:
> > >
> > > "if a discriminative model does generalize compositionally when trained on subsets of the combinatorial concept space, what must necessarily be true of its embeddings?"
> > >
> > > Surely your necessary conditions. But again, this is straightforward since it is exactly the definition of combinatorial generalization.
> > >
> > > "We have now clarified this distinction in Sec. 3.3 to make it explicit that the desiderata are model- and not representation-level."
> > >
> > > But all the experiments are at the representation level, no? You take probes and attempt to decode from the embedding level of CLIP. You are not testing CLIP for compositionality on some task: for example aligning some text embedding with the corresponding image when they are out of distribution, you are just testing if it is easy to decode from them consistently for novel combinations.
> > >
> > > "Modern large-scale models like CLIP show some compositional generalization capabilities, but as we document in our related work, they still fail on many compositional tasks. Our work identifies necessary geometric structure for reliable compositional generalization. "
> > >
> > > But, do you relate this linearity (or lack thereof) to generalization performance on the tasks that CLIP is usually evaluated? As I understand it your experiments focus on establishing how linear the embedding space of CLIP is.
> > >
> > >
> > > "Since there are two ways to construe the questions, we answer both:"
> > >
> > > I meant 1.
> > >
> > > In light of the above I am keeping my score and recommendation.

---

> > > > ### Author Response · Authors · 2025-11-30
> > > >
> > > > Thank you for your comments and suggestions.
> > > >
> > > > ## 1. First, I agree with the authors' desiderata as stated in plain words, I just don't think these necessary conditions are novel since 2 and 3 are just the definition of combinatorial generalization.
> > > >
> > > > We appreciate that the reviewer finds the desiderata reasonable in plain words.
> > > > However, the claim that they are "just the definition" does not match the literature the reviewer cites.
> > > > - The disentanglement papers referenced (Higgins et al.; Watters et al.; Eastwood & Williams) do **not** define model-level criteria for compositional generalization, do **not** specify combinatorial train/test partitions, and do **not** consider a notion of model-level **stability across different training subsets**. These works analyze _representation-level_ properties (e.g., disentanglement).
> > > > - Only Montero et al. consider held-out combinations systematically, but even there no formal desiderata are stated, and no connection is made between such desiderata and representational geometry.
> > > > - The works cited mostly consider generative setting. We ask how it relates to compositional generalization in discriminative settings.
> > > >
> > > > Our desiderata are **model-level conditions across families of training sets**, and it is these model-level conditions, and not the informal notion of "doing well on unseen combos" that allow us to prove nontrivial geometric consequences (factorization, orthogonality). These results do not follow from any definition of compositional/combinatorial generalization in prior work.
> > > >
> > > > ## 2. That is, these are not necessary, they are necessary and sufficient and the rest is tautological (a model generalizes if it does everything that is stated in the definition of generalization).
> > > >
> > > > **This is a misunderstanding of what the desiderata are.**
> > > > - Our desiderata describe **how the model should behave when trained son different subsets of the concept space**.
> > > >     This is **not** part of any standard definition of compositional generalization, and it does **not** appear in the disentanglement papers the reviewer cites - especially in a formal setup.
> > > >
> > > > - We do _not_ prove “generalization $\Rightarrow$ generalization.”
> > > >     - We prove:
> > > >         - **If** a model behaves well across these different training subsets,
> > > >         - **then** its representations must have a very specific structure (additive factors, near-orthogonality, etc.).
> > > >     - In other words, we prove: "generalization $\Rightarrow$ geometric structure"
> > > >
> > > >     This structure does **not** follow from any definition of compositional generalization; it is what the desiderata force under standard training.
> > > >
> > > > To make this more clear, we have modified the introduction to emphasize this.

---

> > > > > ### Author Response · Authors · 2025-11-30
> > > > >
> > > > > ## 3. Second, and as a corollary, I believe the authors are misusing the concept. Combinatorial generalization and compositionality are not the same, with the later being one mechanism by which you can achieve the former. This leads me to my third concern which is that the authors are studying a particular form of combinatorial generalization: namely the ability to decompose an input into constituent concepts when presented with novel combinations. We could call this "semantic decomposition". How these two mechanisms (compositionality and semantic decomposition) interact when solving a particular task is definitely an interesting question, but not one that is being explored in this work.
> > > > >
> > > > > We appreciate the care with definitions. There are two separate issues here: (1) the use of the term _compositional generalization_, and (2) the claim that we are studying a different mechanism ("semantic decomposition").
> > > > >
> > > > > - On terminology, we follow how _compositional generalization_ is used in works on vision and VLMs: generalizing to **unseen combinations of known concepts**, whether in generative or discriminative form [1-5]. Our setting is the standard discriminative one: given an input with a novel combination of familiar concepts, the model should still recover the correct concept values. This matches how it is used in current compositionality and classification work.
> > > > >
> > > > >   We are happy to acknowledge alternative terms (e.g. combinatorial generalization) in the paper used in other domains though.
> > > > > - On mechanisms, the disentanglement papers cited by the reviewer study generative models and representation priors; they do **not** define compositional generalization in terms of train/test splits over the concept grid, nor do they analyze discriminative generalization under such splits. Conceptually, we are doing the same thing those works do, but in the inverse direction: they study how concepts combine to _generate_ inputs, we study how to _recover_ the same concepts from inputs under novel combinations. Both sides are about the same structure; we focus on the discriminative side, which is precisely what our desiderata formalize.
> > > > > - Lastly, our desiderata are behavioral: a model should correctly predict concepts on held-out combinations, stably across training supports. Whether this is achieved via "compositionality" (sometimes defined through properties of representations) or "semantic decomposition", or any other mechanism is outside our scope. We prove that any model satisfying these behavioral conditions must have specific geometric structure. The reviewer's distinction between mechanisms, while conceptually interesting, does not affect our results, which follow from behavior of the model alone.
> > > > >
> > > > >
> > > > > [1] Mahajan, Divyat; Pezeshki, Mohammad; Arnal, Charles; Mitliagkas, Ioannis; Ahuja, Kartik; Vincent, Pascal - Compositional Risk Minimization, ICML 2025
> > > > > [2] Elmoznino, Eric; Jiralerspong, Thomas; Bengio, Yoshua; Lajoie, Guillaume - Towards a Formal Theory of Representational Compositionality, ICML 2025
> > > > > [3] Ren, Yi; Lavoie, Samuel; Galkin, Mikhail; Sutherland, Danica J.; Courville, Aaron - Improving Compositional Generalization Using Iterated Learning and Simplicial Embeddings, NeurIPS 2023
> > > > > [4] Kempf, Elias; Schrodi, Simon; Argus, Max; Brox, Thomas - When and How Does CLIP Enable Domain and Compositional Generalization?, ICML 2025
> > > > > [5] Atzmon, Yuval; Kreuk, Felix; Shalit, Uri; Chechik, Gal - A Causal View of Compositional Zero-Shot Recognition, NeurIPS 2020
> > > > >
> > > > >
> > > > > ## 4. This follows a similar path to disentanglement work: linear classifiers fail because the embedding space is not linear enough (as shown by Schott, Montero, etc.).
> > > > >
> > > > > There are two separate issues here: (1) architecture, and (2) the model regime.
> > > > > - The disentanglement papers cited by the reviewer study relatively small, specialized, and mostly generative models with latent spaces and heads that were _not_ designed for large-scale transfer, and where linear readouts were only one of many possible classifiers. In that setting, failures of linear probes say more about the chosen architecture than about necessary structure for generalization.
> > > > > - In contrast, modern foundation models such as CLIP use a **linear** decision rule: the image encoder and text encoder produce embeddings, and a dot product between them defines the logits.
> > > > > 	- Linear readout is the inference mechanism in CLIP.
> > > > > 	- Our focus on linear probing is therefore aligned with how these models are actually used
> > > > > 	- The papers the reviewer cite do not consider foundation models at all; we study them precisely of the expectation that they _should_ work under unseen data comopositions.

---

> > > > > > ### Author Response · Authors · 2025-11-30
> > > > > >
> > > > > > ## 5. Except in that case the encoders were also tested for generalization (that is they were presented with held-out data) while here only the probes are tested in this way (since by their own admission the authors cannot guarantee that similar or even the same datasets were not seen during training by CLIP).
> > > > > >
> > > > > > We do not think this is an accurate characterization.
> > > > > > - In all our experiments, the _encoders_ are evaluated on held‑out **combinations** of concepts, and in several cases on datasets that CLIP definitely did **not** see during pretraining. In particular, PUG‑Animal and ImageNet-AO are datasets introduced after CLIP.
> > > > > > 	- In these settings, it is not just the probes but also the encoders that are tested on novel images and compositions.
> > > > > > - Our main goal, however, is theoretical: assuming a model _does_ generalize compositionally from limited coverage of the concept grid, we characterize what its representations must look like. The empirical results simply check to which extent the modern foundation models exhibit the predicted structure in realistic regimes with unnatural concept combinations.
> > > > > >
> > > > > > ## 6. Thus the article is left in a weird position. It claims to propose necessary conditions for compositional generalization while just restating the definition of combinatorial generalization
> > > > > >
> > > > > > We hope the earlier responses clarify this (Especially 2).
> > > > > >
> > > > > > ## 7. then it focuses on a particular flavor of the generalization task which may not require compositionality, which further changes from the original framing as it introduces further assumptions during the experimental setup (by focusing on linear probes on pre-trained models).
> > > > > >
> > > > > > We hope (4) clarifies this.
> > > > > >
> > > > > > ## 8. All of which makes the results not as surprising in light of previous studies in the disentanglement literature/combinatorial generalization literature.
> > > > > >
> > > > > > The disentanglement works cited by the reviewer do not establish our results. They:
> > > > > > 	1. focus on generative VAEs and representation-level notions such as disentanglement and completeness,
> > > > > > 	2. do not formalize compositional generalization via combinatorial train/test splits or stability across supports, and
> > > > > > 	3. do not derive necessary geometric structure (linear factorization, orthogonality) for models that _do_ generalize. Our contribution is exactly this implication for discriminative, CLIP-like settings.
> > > > > >
> > > > > > ## 9. I think the authors can still make this a good article but I would recommend rephrasing it. First, forget about compositionality, since I as I said, I don't believe that this is the experiments study. They can focus directly on the idea of semantic decomposition, framing it as a desirable step on the road to perform compositionality. Finally they can either study how easy it is to generalize from pre-trained (and thus probably in-distribution) models or train new models from scratch to establish if they are learning easy to decode linear representations (which is what they are ultimately trying to show anyway).
> > > > > >
> > > > > > We appreciate the suggestions, but we do not believe it reflects our work accurately:
> > > > > >
> > > > > > - Our object of study is compositional generalization in the usual sense used in vision/VLM work: handling **unseen combinations of known concept values**. Recovering the constituent concepts from inputs (what the reviewer calls “semantic decomposition”) is simply the standard **discriminative** way to evaluate this, not a different problem.
> > > > > > - Training new models from scratch would address a different question. We are not trying to show that some particular architecture “learns easy‑to‑decode linear representations”, but to characterize **what must be true** of representations _if_ a discriminative model does generalize compositionally from subsets of the concept grid. We aim to understand where current and future foundation models are headed and what properties they must satisfy once they get there.
> > > > > > - Finally, training from scratch tends to produce linear structure only when the observed data covers a large portion of the underlying concept space, as shown in [6]. As our motivation explains, such data coverage is infeasible in realistic settings, and our goal here is instead to understand and track the emergence of the **implied** representational structure under limited data. In short, training from scratch is orthogonal to our main question and not what we are trying to show.
> > > > > > 		[6] Uselis, Arnas; Dittadi, Andrea; Oh, Seong Joon - Does Data Scaling Lead to Visual Compositional Generalization?, ICML 2025.
> > > > > > - The "ID vs OOD" issue is also separate from this: our desiderata and theoretical results are formulated at the level of concept combinations and do not depend on whether a particular dataset was seen during pretraining (and as explained in (5) some of the datasets are OOD in qualitative sense). We discuss this in the related work.
> > > > > >
> > > > > > ## 10. This is straightforward since it is exactly the definition of combinatorial generalization.
> > > > > >
> > > > > > We believe (2) clarifies this.

---

> > > > > > > ### Author Response · Authors · 2025-11-30
> > > > > > >
> > > > > > > ## 11. All experiments operate only at the representation level; you are not evaluating CLIP on an actual compositional task, only how easily concepts decode linearly.
> > > > > > >
> > > > > > > We are exactly testing how strong the **image encoder** is. We instantiate our framework on top of it: we train linear probes on its embeddings and check whether they satisfy the ideal structure our theory says models need in order to compositionally generalize.
> > > > > > >
> > > > > > > CLIP’s text encoder is doing essentially the same thing: it tries to “just” map prompts to concept values so that a **dot product** with the image embedding implements the classifier (Sec. 2.6). We choose to fit linear probes on the image embeddings instead of going through the text encoder in order to (1) avoid prompt and wording issues where the concept is hard to express in text, and (2) avoid confounds from text-image misalignment. Our experiments are thus directly about whether CLIP’s image representations support compositional generalization in the standard discriminative sense (predicting concepts of unseen combinations).
> > > > > > >
> > > > > > >
> > > > > > > ## 12. But, do you relate this linearity (or lack thereof) to generalization performance on the tasks that CLIP is usually evaluated? As I understand it your experiments focus on establishing how linear the embedding space of CLIP is.
> > > > > > >
> > > > > > > Yes, indirectly. We relate geometric structure (factorization, orthogonality) to generalization by fitting linear probes on the image encoder and measuring whether representations exhibit the structure our theory predicts. We use probes rather than zero-shot text prompts to isolate the image encoder's geometry from confounds introduced by the text encoder (prompt wording, misalignment between text and image modalities, continuous concept values, etc.). This gives cleaner and less biased measurements of whether the predicted structure is present.
> > > > > > >
> > > > > > > That said, to make the analysis more direct, we now provide experiments where the text encoder is directly used zero-shot in Appendix D.3.
> > > > > > > - Instead of fitting the linear probes, we derive them from the text encoder by passing text prompts.
> > > > > > > - We report zero-shot CLIP performance on PUG-Animal and ImageNet-AO and relate it to the same geometric properties.
> > > > > > > - In conclusion though, the same qualitative trend holds: models with representations closer to the predicted linear structure perform better on compositional benchmarks, and the concept factors are more orthogonal across- than within-concept.

---

### Official Review · Reviewer_RSFo · 2025-10-30

**Soundness:** 3
**Presentation:** 3
**Contribution:** 3
**Rating:** 8
**Confidence:** 4

**Summary:**

This manuscript investigates how the format of model representations influences the ability to compositionally generalize, formalizing compositional generalization ability into three desiderata. They prove theoretically that their desiderata are only satisfied by models with a linearly factorized representation immediately before the readout (if those models a trained on crossentropy with gradient descent). Moreover, different concepts should be represented in orthogonal directions whereas, in principle, the representation could collapse different categories for the same concept into the same dimension. They then test their predictions in pretrained vision models and find that they are indeed more linearly factorized after pretraining than with random weights, their degree of linear factorization predicts their model performance, and distinct concepts are indeed represented more orthogonally than distinct categories of the same concept.

**Strengths:**

I liked this paper! It provides a formally principled perspective on compositional generalization and a useful theoretical intuition on the representations we'd expect in such models. It also provides a principled set of experiments to test its theoretical predictions. Below I'm highlighting a few things I particularly liked:

- The paper does a great job at motivating the problem of compositional generalization. It is written very clearly; it is apparent that the authors have thought carefully about how to present their findings in an accessible manner.
- I found proposition 2 particularly thought-provoking and the illustration was helpful in getting an immediate intuition for how to think about this
- The empirical findings are very interesting, in particular the fact that linear factorization predicts compositional generalization and that distinct categories are represented in a less orthogonal manner than distinct concepts

**Weaknesses:**

Currently, the theory in the paper imposes a pretty strong constraint in the form of stability: to leave generalization completely unchanged across different training sets, we need exactly linearly factorized representations. I'm not entirely convinced by the authors' justification for the desideratum of stability; while this may be a relevant starting point, I think it substantially limits the impact of the theory. In particular, the authors find that in practice representations are certainly not perfectly linearly factorized (as their theory would prescribe). In its current shape, we can't use the theory to interpret whether an R^2 of 0.4-0.6 is good or bad. That said, I do appreciate the experimental observation that an increased R^2 yields improved compositional generalization, which helps with bridging this gap. I also still appreciated the theoretical conceptualization.

I also think it would be good to further contextualize this theory in existing work on relating representational geometry and compositional generalization. In particular, [1] and [2] (and [3], which the authors do cite) both tie compositional generalization to linearly factorized representations in the output representational space. Notably, both [2] and [3] formalize this question through a kernel perspective which is highly related to the SVM perspective leveraged in the proof here. On the conceptual end, I think the "related work" section (l. 82-92) mostly talks about how linear subspaces have been identified. However, many papers also emphasize the usefulness of factorized representations or try to identify factorized/disentangled representations [4,5]; I think the current paper goes beyond this work by providing a particular formal perspective on this, but it may be useful to draw a connection.

Some minor notes:
- L. 75-79: I think the verb is missing in this sentence.
- L. 91: I imagine you don’t want to have the “we” in italics?
- In Figure 5 I would make the light orange data points a little more salient, they are a little hard to see right now.
- L. 170: “practise” -> “practice”
- L. 298-299: I think there is a spare “either” there.
- L. 347: “factoried” -> “factorized”
- L. 361: Why is there a c in the superscript, aren’t you counting through i\in[c] in the subscript?
- L. 389 “Generaliaztion” -> “Generalization”

1. https://arxiv.org/abs/2409.14981
2. https://arxiv.org/abs/2501.18797
3. https://arxiv.org/abs/2405.16391
4. https://openreview.net/forum?id=Sy2fzU9gl
5. https://www.cambridge.org/core/journals/behavioral-and-brain-sciences/article/building-machines-that-learn-and-think-like-people/A9535B1D745A0377E16C590E14B94993

**Questions:**

- Could you provide some more context on stability as a relevant desideratum and talk about how the theory could be extended beyond this specific desideratum, e.g. to clarify how different degrees of linear factorization influence compositional generalization? (See notes under "Weaknesses".)
- Could you contextualize the paper a bit further in the prior literature? (See notes under "Weaknesses".)
- Do you have any intuition for how linear the representations observed in these models actually are? I realize that this is a somewhat vague question but I found it a bit difficult to interpret an R^2 between 0.4-0.6. It may be useful to indicate the R^2 of e.g. the kinds of schematic representations you are showing in Fig. 5?
- Doesn't transferability imply divisibility?

---

> ### Author Response · Authors · 2025-11-24
>
> Thank you for a thorough review.
>
> ## Context on stability as a relevant desideratum
>
> We appreciate this concern and question.
>
> We believe that (1) stability as a desideratum, and (2) relation between linearity and compositional generalization are separate.
>
> Regarding stability.
>
> We believe that if a model's understanding of a concept value changes based on which co-occurrences were in a particular training set, it's necessarily biased towards particular combinations - it's learning something about the specific combinations rather than the concepts themselves. Stability captures this requirement: once concept values are covered, their meaning shouldn't depend on which particular combinations achieved that coverage.
>
> Without stability, transferability can become fragile. For example, Figure 14(b) shows a nasty case where the majority of the datapoints lie almost on the boundary of classification decision boundaries. If only a fraction of the subset was shown to the classifier, such configurations wouldn't transfer: the decision boundaries would look vastly different. To avoid such brittleness in transferable-but-not-stable settings, one would need to impose other "robustness" conditions. Because of that, we do think it is a desirable property to have stability.
>
> Regarding relation between linearity and compositional generalization.
>
> This does seem tricky, especially from a theory perspective. Our theory characterizes necessary conditions for perfect compositional generalization. To characterize partial generalization, we would need to relax this assumption and instead consider a model that exhibits some degree of linearity but does not necessarily achieve full compositional transfer. The question would then become: if some portion of variance isn't explained by linear factors, how does this influence classification? To understand this, one would have to consider different potential sources for this lack of linearity: is it unexplained noise? or perhaps it comes from interactions/entanglement between the values of certain concepts in the representations? We do believe that understanding the source of failure cases is an interesting avenue in itself.
>
> From an empirical perspective, qualitative examples showing sample-level linearity scores are somewhat telling (displayed in Figure 21) and could be used as a starting point.
>
>
> ## In particular, the authors find that in practice representations are certainly not perfectly linearly factorized (as their theory would prescribe)
>
> We may have been unclear in our writing. We did not mean to suggest that current models achieve the ideal our theory derives - indeed, the work is motivated by the failure cases they exhibit.
>
> ## Do you have any intuition for how linear the representations observed in these models actually are?
>
> This is challenging to develop intuition for, particularly in high dimensions. We provide qualitative visualizations in Figures 13 and 14, showing configurations in 2D and 3D spaces. When the number of concepts exceeds the ambient dimension (cases c, d, e in Figure 14), interpretation becomes harder. Figure 17 shows actual datapoints in 3D space for a model with $R^2 = 0.42$ for shape and size, where three concept factors collectively span the 3-dimensional space.
>
> ## I also think it would be good to further contextualize this theory in existing work on relating representational geometry and compositional generalization.
>
> We now do so in the updated manuscript, thank you for suggestions and references.
>
> ## Doesn't transferability imply divisibility?
> Yes it does. Since we study the capacity of the representation space, we made Divisibility explicit: before we can consider transfer, we need to ensure there is sufficient capacity to represent all concepts and their values. We have clarified this in the text as well.
>
>
> ## Colors in Figure 5
> Thank you for pointing that out, we will do so in the next revision.
>
> ##  L. 361: Why is there a c in the superscript, aren't you counting through i\\in\[c\] in the subscript?
> Thank you for catching that; it was a typo, now removed.
>
> ## Typos
> Thank you for pointing these out, fixed.

---

> > ### Comment · Reviewer_RSFo · 2025-11-24
> >
> > Thank you for your response! Your further contextualization of stability was helpful. I continue to think that it is quite a strong requirement and that there are many cases that don't satisfy stability that we would count as compositionally generalizing. That said, I agree that there is value in considering idealized theoretical scenarios and leaving the messier cases to future work.
> >
> > I also appreciate the extended discussion of related work; I think it helps in further embedding your paper in the prior literature.
> >
> > Overall, I continue to think that this is a good submission and I would recommend acceptance.

---

### Official Review · Reviewer_Tn7L · 2025-10-31

**Soundness:** 3
**Presentation:** 3
**Contribution:** 3
**Rating:** 6
**Confidence:** 3

**Summary:**

This paper studies the structural properties that representations must have for models to generalize compositionally, in other words, recognize familiar parts within new combinations. The authors argue that since models are trained on a tiny subset of all possible concept combinations, to generalize reliably their embeddings must satisfy certain necessary conditions.

They define three key necessary desiderata for compositional generalization: divisibility, transferability, and stability. From these, they argue that embeddings must be a linear combination of per-concept components and that the components must be nearly orthogonal. Furthermore, the embedding dimension must be at least as large as the number of independent concepts.

Thesee ideas are tested empirically on variants of CLIP and SigLIP across datasets such as PUG-Animal, dSprites, and MPI3D. The authors find that compositional generalization requires this linear, nearly-orthogonal structure. These findings support a practical diagnostic  method for checking whether models can generalize compositionally.

**Strengths:**

The paper identifies a set of simple principles (divisibility, transferability, and stabiliity) as the foundation for compositional generalization, and the math nicely shows that these principles directly lead to linear factorization and orthogonality in embeddings.

The paper shows how $R^2$ and factor orthogonality can measure how compositional a model’s features are, which is an interesting and practical idea.

The text is will structured, with each section including clear takeaways and helpful diagrams supporting the arguments.
The paper gives a useful geometry-based checklist for when models can or cannot generalize compositionally.

**Weaknesses:**

The theory relies on strong assumptions such as linear readouts, binary concepts, and ideal separability. Extending the experiments to incude models with nonlinear heads or continuous attributes would help test how far the results generalize.

Although the concept of "Projected R^2" is used throughout the paper, it is only described in Appendix B. It would help to have at least a brief conceptual explanation of it when it is first introduced (line 368), and the use of whitening.

Even though the paper introduces interesting theoretical and empirical results about linear factorization, orthogonality, and dimensionality, these ideas could be more useful in practice if the authors at some point explicitly framed them as a diagnostic toolkit for model design. For example, they could summarize how to measure each property (Projected R2, concept orthogonality, and factor rank) and explain how those metrics might guide architecture choices for new problems/datasets.

**Questions:**

How much does the Projected R^2 metric depend on whitening?

What counts as a "good" value for R^2?

How are the concepts defined in datasets like PUG-Animal or MPI3D?

---

> ### Author Response · Authors · 2025-11-24
>
> Thank you for a thorough review.
>
> ## Extending the experiments to include models with nonlinear heads or continuous attributes would help test how far the results generalize.
>
> Regarding continuous attributes: While our theoretical framework models concepts as discrete, some of the concepts in our experimental datasets are continuous (as shown e.g. in Figures 10 and 11).
>
> Regarding non-linear heads: we agree that experiments on non-linear readouts would be interesting. We'd like to note that if the readout is a NN, most commonly such a network would produce another representation that would then still be probed linearly (as is done in CLIP with the text encoder producing weights of the probes). This is the reason why we focus on linear readouts. While alternatives exist, these would be interesting to explore in future work.
>
> ## Explanation of $R^2$ score
> Thank you for the suggestion - we now explain it in Section 5.1. We additionally provide intuition for it in Appendix C.3.
>
> ## How are the concepts defined in datasets like PUG-Animal or MPI3D?
>
> These datasets expose the generative factors; for all datasets these are specified in Figure 10. For example, PUG-Animal exposes 4 concepts: character, world, size, texture. Each concept has from 3 to 64 values. We will be sure to provide the full table of all concepts and their values in the next revision.
>
> ## How much does the Projected R^2 metric depend on whitening?
>
> It can depend quite a bit. Generally though, the score post-whitening is almost always lower. We added justification and intuition for it in Appendix C.2, illustrating that removing the dominant directions in the embedding space generally reduces the high-variance directions dominating the raw $R^2$ score.
>
> ## What counts as a "good" value for R^2?
>
> A score of 1.0 is unambiguous: it means the model can perfectly reconstruct $n^c$ datapoints using only $n \cdot c$ components. Lower scores indicate partial factorization.
>
> We interpret scores of 0.4-0.6 as showing some but incomplete linear factorization. Current models exhibit the derived properties partially, suggesting room for improvement.
>
> ## Framing: a diagnostic toolkit for model design
>
> Thank you for this suggestion. We agree, and did intend to present our results more as a comparison of current models the idealized ones derived in our work, than an argument that current systems are already ideal. We revised our takeaways to reflect this.

---

> > ### Comment · Reviewer_Tn7L · 2025-11-27
> >
> > I thank the authors for their responses and for the clarifications.
> >
> > I remain finding this is a good submission but I still think that the limitation of experiments to linear heads restricts the applicability of these results in practice.
> >
> > Finally, reviewer VaZz has pointed out interesting prior work that I did not know. I will need to study it first to be in a better position to evaluate the contributions of this paper.

---

### Official Review · Reviewer_8gtS · 2025-10-31

**Soundness:** 2
**Presentation:** 2
**Contribution:** 1
**Rating:** 2
**Confidence:** 2

**Summary:**

This paper presents theoretical and empirical analysis into the properties needed for visual encoding models to exhibit compositional generalization. The paper states three desiderata for embeddings to be considered to generalize compositionally, and then shows that any embeddings that meet these desiderata must decompose embeddings into the sum of orthogonal concept representations (linear factorization + orthogonal factors). Empirically, experiments on several models and datasets show that vision models show some degree of linear factorization, the degree of factorization is correlated with compositional generalization accuracy (using linear probes), and different concept factors are "nearly orthogonal".

**Strengths:**

- This paper investigates the interesting question of how neural networks trained on limited data are able to generalize compositionally. I think this is a useful topic--understanding compositional generalization could improve our theoretical understanding of neural networks and motivate practical improvements to data or architecture.

- The organization of the paper is clear and the paper is mostly easy to follow.

- The paper conducts extensive empirical experiments to support the predictions, including a variety of image models and datasets.

**Weaknesses:**

- The two conditions identified in this paper (linear factorization + orthogonal factors) might be limited contributions. The linear factorization condition seems like a straightforward consequence of the definition of compositional generalization, which requires that concepts can be classified with a linear probe. The condition of strictly orthogonal factors is likely never actually met in practice--concepts are only approximately orthogonal. (I would think that dimensionality is typically less than the number of concepts.) One way to strengthen the paper could be to discuss in more detail how the desiderata and conclusions change given approximate orthogonality rather than strict orthogonality. See relevant discussion in [1].

- A number of the empirical conclusions are based on thresholds that seem subjective to me, so it is hard to say how much they really substantiate the theoretical claims. Section 5.1/figure 7 states that embeddings exhibit "substantial" linear factorization, but it is not clear to me what $R^2$ should be considered substantial, and what we should conclude from the fact that the $R^2$ scores are still much less than 1.0. The takeaway in section 5.3 is that "per-concept difference vectors are nearly-orthogonal across concepts", but I am not sure what the threshold is for defining "nearly-orthogonal". Similarly, in section 5.4, Figure 10 seems to show that per-concept dimensionality is actually relatively high for categorical features, so it is not clear to me that we can conclude that "overall, semantic factors are low-rank". In general, I think the paper would be stronger if it made more formal, precise predictions.

- The main text of the paper is missing some methodological details, making it difficult to interpret the empirical results. For example, how exactly is compositional accuracy calculated in section 5.2? The paper also does not give a clear definition of Project $R^2$ or explain why it should be considered a good measure of linear factorization (although there is some discussion in the appendix). It would be helpful to include more experimental details in the main text.

- Related work: I am not very familiar with this area and I found it difficult to fully understand the relationship between this paper and the prior work. For example, in section 2, the paper mentions prior work that has "emphasized formal sufficient conditions for generative systems". How does that differ from this work? I appreciated the illustrations of previous work in Figure 2, but these works are not discussed in the text, so it is still hard to understand the relationship with prior work.

- I found the notation to be difficult to follow at times. For example, $c$ is used to denote the number of concepts, but $c_i$ is used to denote the $i^{th}$ entry in the concept tuple $\mathcal{c}$. Please see more detailed comments in the Questions section.


**Summary:** I think the paper presents an interesting analysis of compositional generalization, with a variety of empirical evidence to support the conclusions. However, I feel that presentation issues make it difficult to understand the experiments and the notation; the interpretation of the experiments might be somewhat subjective; and the contributions might have limited significance. I would consider increasing my score if the authors could suggest ways to make the empirical predictions more precise; and address my concerns about clarity; and better contextualize the contributions with respect to prior work.


_References_

[1] Elhage et al., 2022. Toy Models of Superposition.

**Questions:**

- Can you expand on the role of the "validity class" (bottom of page 3)? It seems that this can be any arbitrary rule for defining constraints on the training data. Do the desiderata and conclusions apply to any choice of validity class? If not, how is the validity class identified?

- In Proposition 1 (line 299), what is the justification for the validity rule $|T| = 2^{n-1} + 1$? Previously, $n$ was the number of values each concept could take on. In this binary case, $n = 2$. I would have thought that the validity rule should depend on the number of concepts.

- Why is desideratum 1 (divisibility) necessary? Doesn't desideratum 2 (transferability) imply desideratum 1?

- Desideratum 3: It seems that this implies constraints on the validity rule for determining valid supports.


- In sections 5.1 and 5.2, can you explain what exactly Project $R^2$ is measuring and why this is a measure of linear factorization?

- In line 160, should $\mathcal{T} \subseteq 2^{\mathcal{C}}$ be $\mathcal{T} \subseteq n^c$? Similarly in Definition 2.

- What is $\mathcal{X}'$ in equation 4?

- What is $\mathcal{F}$ in Figure 3?

- In figure 10a, what are the annotations above each of the columns?

- I do not quite understand what Figure 1 is meant to illustrate. It seems that the two concepts are "is cat present" and "is person present", but both the training image and the testing image show the same composition of concepts (cat present = true, person present = true), so it doesn't seem like an illustration of compositionality.

- Section 5.4 mentions that some of the concepts in the experiments are continuous rather than discrete, but the theoretical analysis assumes discrete concepts. Can you comment on how the desiderata and conclusions extend to continuous concepts?

---

> ### Author Response · Authors · 2025-11-24
>
> We thank you for a thorough response.
>
> ## The linear factorization condition seems like a straightforward consequence of the definition of compositional generalization
>
> We would like to note that linear factorization isn't trivial: under linear
> factorization, $n^c$ datapoints can be explained using only $n \cdot c$ factors.
> However, not every linearly compositional model achieves this. To illustrate this furher, we have included a section in Appendix B.4 ("Non-triviality of linear
> factorization of linearly compositional models") - such models linearly clasify all the datapoints. However, their embedding space isn't linearly factorized.
>
> ## I would think that dimensionality is typically less than the number of concepts
>
> We believe this is an important point: we consider a single concept space underlying the world, and instantiate it in experiments through datasets like DSprites and PUG-Animal. In practice, however, datasets are often mixtures of domains: depending on the content of the data (or image), the actual concepts depend on the content of the data. For example, when the data depicts airplanes, the concept space associated with it may be different than when the data depicts 2D shapes. In other words, taken together, yes, the number of concepts is large, but locally within-domain, the required concept space may be smaller.
> We will clarify this in the revised text.
> ## Approximate orthogonality and relation to Toy Models of Superposition
>
> Thank you for raising this point. We have now discussed [a] in the related work.
>
> The key distinction is one of scope and direction: [1] study superposition: how networks can represent more features than dimensions in two-layer auto-encoder models with varying feature importance. They empirically
> observe near-orthogonal encoding as a practical solution to this trade-off. Our work addresses a different question: what geometric structure is *necessary*
> for compositional generalization? We show that under our desiderata and standard
> discriminative training, cross-concept orthogonality must emerge. Also, we note that [1] does not consider generalization settings.
>
> [a] Elhage et al., 2022\. Toy Models of Superposition.
>
>
> ## Section 5.1/figure 7 states that embeddings exhibit "substantial" linear factorization, but it is not clear to me what should be considered substantial, and what we should conclude from the fact that the scores are still much less than 1.0
>
> Thank you for pointing this out. We weren't clear on our aims in the text. By "substantial," we meant that the $R^2$ scores are higher than those of randomly-initialized models with fitted probes. Our intention was to indicate that current models exhibit these properties partially but not fully, suggesting room for improvement. We have revised the takeaways to reflect this more precisely.
>
>
> ## Elaboration on $R^2$ computation and compositional accuracy measurement
>
> We have elaborated how compositional accuracy is measured in Sec. 5.2, and $R^2$ score computation in Sec. 5.1. We now also provide additional results for intuition of linearity and its score in Appendix B.3.
>
>
> ## Difference from the sufficient conditions
>
> Previous works mostly emphasize sufficiency for compositional generalization: e.g., if certain conditions hold (either in the model or the data generative process), then compositional generalization is guaranteed. We instead ask what properties are implied *if* a model transfers from a restricted subset of the data space to the full space under our desiderata - a question of necessary rather than sufficient conditions. We have now clarified this distinction in the related work section.
>
> ## Discussion of works shown in Figure 2
>
> We have added a discussion of the works shown in Figure 2 to the text.

---

> ### Author Response · Authors · 2025-11-24
>
> ## 1. Role of validity classes
>
> In principle, the validity rule can be arbitrary. However, certain validity classes, e.g., of size less than $1 + c(n-1)$, necessarily do not cover at least a single concept value support and thus are clearly unlearnable. The desiderata apply with respect to any choice of validity class. The conclusions hold for all validity classes of size $1 + c \leq |T| \leq 1 + 2^{c-1}$ for $c \geq 2$. We have clarified this in the text.
>
> ## 2. Reasoning behind $|T| = 2^{n-1} + 1$
>
> That was a typo - it was meant to be $2^{c-1} + 1$, thank you for catching it.
>
> ## Doesn't Desideratum 2 imply Desideratum 1?
>
> Yes, it does. Since we study the capacity of the representation space, we made Divisibility explicit: before we can consider transfer, we need to ensure there is sufficient capacity to represent all concepts and their values. We have clarified this in the text as well.
>
> ## 3. Desideratum 3 implies constraints on the validity rules
>
> Perhaps our wording was confusing there: we wrote "posteriors agree across valid supports", where the validity is with respect to the validity class.
>
> ## L160: $\mathcal{T} \subseteq 2^{\mathcal{C}}$ or $\mathcal{T} \subseteq n^c$?
>
> We denote the power set of the concept space by $2^{\mathcal C}$, i.e., the set of all subsets of $\mathcal C$. We have now clarified this in the text.
>
> ## $\mathcal{X}'$ in Eq. 4, $\mathcal{F}$ in Fig. 3
>
> Our apologies for these typos. We have clarified that these should be $\mathcal{Z}$ and $T$, respectively.
>
> ## 4. Meaning of Fig. 1
>
> Thank you for pointing out this lack of clarity. We have clarified the caption and its meaning in the introduction text.
>
> ## Figure 10a: annotations
>
> We have added annotations to the caption. The annotations show the rank of the concept factors and the total number of concept values.
>
> ## Impact of continuous concept values
>
> Thank you for raising this. In theory, continuous concepts (infinite concept values) are compatible with our framework: Proposition 4.2 shows dimensionality depends on the number of concepts, not the number of values per concept.
>
> However, a key requirement is that concept values seen during training adequately cover the test distribution -- our framework operates only on novel compositions of known concepts. If continuous values can vary arbitrarily between train and test, the model could represent them in widely different regions of the embedding space.
>
> In practice, large-scale models like CLIP are supervised through text captions that naturally discretize continuous concepts (e.g., "left side" rather than exact coordinates, "tilted" rather than precise angles), which our framework covers.
>
>
> ## Notation
> Thank you for pointing this out. We have renamed the confusing $c$ to $k$ in the revised manuscript. Note that the answers in this rebuttal use the old notation.

---

### Meta-Review · Area_Chair_zuj9 · 2026-01-06

**Summary:**

The reviewers were divided in this paper, giving scores (2,2,6,6,8).

One concern raised by a couple of reviewers is that it is unclear how to validate whether the experiments support the conclusions. One reviewer (VaZz), who gave a score of 2 with confidence 5, raised many concerns. In particular, they claimed that the experiments provided are not systematic enough, and compared them with previous, more systematic work. The reviewers responded, but the reviewer was still not satisfied with the response and restated their concerns. The authors responded again, but the reviewer could not participate in the discussion anymore. The discussion shows that there is a fundamental disagreement between this reviewer and the authors.

If it were just one reviewer who suggested rejecting, I'd be more comfortable with suggesting accepting despite the disagreement. But given that two reviewers suggested to reject, I'm leaning towards rejection and recommending the authors to resubmit. I must note that reviewer 8gtS, who gave a score of 2 with confidence 2, was willing to increase their score.

A second look by the senior AC would be useful.

**Reviewer Concerns:**

The authors provide a fair summary of the issues brought up by the reviewers and their responses.

**Reviewer Scores:**

Reviewer 8gtS would probably have raised their score after the rebuttal. The rest of the reviewers would not have.

---

### Decision · Program_Chairs · 2026-01-26

Reject